# Transformers as Game Players: Provable In-context Game-playing Capabilities of Pre-trained Models

**Chengshuai Shi**
University of Virginia
cs7ync@virginia.edu

**Kun Yang**
University of Virginia
ky9tc@virginia.edu

**Jing Yang**
The Pennsylvania State University
yangjing@psu.edu

**Cong Shen**
University of Virginia
cong@virginia.edu

## Abstract

The in-context learning (ICL) capability of pre-trained models based on the transformer architecture has received growing interest in recent years. While theoretical understanding has been obtained for ICL in reinforcement learning (RL), the previous results are largely confined to the single-agent setting. This work proposes to further explore the in-context learning capabilities of pre-trained transformer models in competitive multi-agent games, i.e., *in-context game-playing* (ICGP). Focusing on the classical two-player zero-sum games, theoretical guarantees are provided to demonstrate that pre-trained transformers can provably learn to approximate Nash equilibrium in an in-context manner for both decentralized and centralized learning settings. As a key part of the proof, constructional results are established to demonstrate that the transformer architecture is sufficiently rich to realize celebrated multi-agent game-playing algorithms, in particular, decentralized V-learning and centralized VI-ULCB.

## 1  Introduction

Since proposed in Vaswani et al. [57], the transformer architecture has received significant interest. It has powered many recent breakthroughs in artificial intelligence [11, 12, 21, 23], including the extremely powerful large language models such as GPT [42] and Llama [55, 56]. One of the most striking observations from the research of these transformer-powered models is that they demonstrate remarkable *in-context learning* (ICL) capabilities. In particular, after appropriate pre-training, the models can handle new tasks when prompted by a few descriptions or demonstrations without any parameter updates, e.g., Brown et al. [11], Chowdhery et al. [15], Liu et al. [39].

ICL is practically attractive as it provides strong generalization capabilities across different downstream tasks without requiring further training or a large amount of task-specific data. These appealing properties have motivated many empirical studies to better understand ICL [18, 26, 59]; see the survey by Dong et al. [22] for key findings and results. In addition to the empirical investigations, recent years have witnessed growing efforts in gaining deeper theoretical insights into ICL, e.g., Ahn et al. [2], Akyürek et al. [3], Bai et al. [7], Cheng et al. [14], Li et al. [37], Raventós et al. [44], Wu et al. [64], Xie et al. [66], Zhang et al. [71].

Among these empirical and theoretical studies, one emerging direction focuses on the capability of pre-trained transformer models to perform *in-context reinforcement learning* (ICRL) [28, 34, 35, 50, 73]. In particular, the transformer is pre-trained with interaction data from diverse environments, modeling

38th Conference on Neural Information Processing Systems (NeurIPS 2024).

the interaction as a sequential prediction task. During inference, the pre-trained transformer is prompted via the interaction trajectory in the current environment for it to select actions. The work by Lin et al. [38] provides some theoretical understanding of ICRL, including both a general pre-training guarantee and specific constructions of transformers to realize some well-known designs in multi-armed bandits and RL (especially, LinUCB [1], Thompson sampling [54], and UCB-VI [6]). Wang et al. [62] further provides understandings on the capability of transformers learning temporal difference (TD) methods [52] via an in-context fashion. A detailed literature review can be found in Sec. 6.

The insights from Lin et al. [38], Wang et al. [62] are largely confined to the single-agent scenario, i.e., a single-agent multi-armed bandit or Markov decision process (MDP). The power of RL, however, extends to the much broader multi-agent scenario, especially the multi-player competitive games such as GO [49], Starcraft [58], and Dota 2 [10]. To provide a more comprehensive understanding of ICRL, this work targets further studying the *in-context game-playing* (ICGP) capabilities of transformers in multi-agent competitive settings. To the best of our knowledge, *this is the first work providing theoretical analyses and empirical pieces of evidence on the ICGP capabilities of transformers*. The contributions of this work are further summarized as follows.

• A general framework is proposed to model in-context game-playing via transformers, where we focus on the representative two-player zero-sum Markov games and target learning Nash equilibrium (NE). Compared with the single-agent scenario [38], the multi-agent setting considered in this work broadens the ICRL research scope while it is also more complicated due to its game-theoretic nature.

• The challenging decentralized learning setting is first studied, where two distinct transformers are trained to learn NE, one for each player, without observing the opponent's actions. A general realizability-conditioned guarantee is first derived that characterizes the generalization error of the pre-trained transformers. Then, the capability of the transformer architecture is demonstrated by providing a concrete construction so that the famous V-learning algorithm [8] can be exactly realized. Lastly, a finite-sample upper bound on the approximation error of NE is proved to establish the ICGP capability of transformers. As a further implication, the result of realizing V-learning demonstrates the capability of pre-trained transformers to perform model-free RL designs, in addition to the model-based ones (e.g., UCB-VI [6] as studied in Lin et al. [38]).

• To obtain a complete understanding, the centralized learning setting is also investigated, where one transformer is pre-trained to control both players' actions. A similar set of results is provided: a general pre-training guarantee, a constructional result to demonstrate realizability, and a finite-sample upper bound on the approximation error of NE. Distinctly, the transformer construction is presented as a specific parameterization to implement the renowned centralized VI-ULCB algorithm [8].

• Furthermore, experiments are also performed to practically test the ICGP capabilities of the pre-trained transformers. The obtained results not only corroborate the derived theoretical claims, but also empirically motivate this and further studies on the interesting direction of pre-trained models in game-theoretic settings.

## 2 A Theoretical Framework for In-Context Game Playing

### 2.1 The Basic Setup of Environments

To demonstrate the ICGP capability of transformers, we focus on one of the most basic game-theoretic settings: two-player zero-sum Markov games [47], while the discussions provided later conceivably extend to more general games. An illustration of the different settings (i.e., decentralized and centralized) considered in this work (with details explained later) is given in Fig. 1. The overall framework is introduced in the following, which extends Lin et al. [38] from the single-agent decision-making setting to the competitive multi-agent domain.

Considering a set of two-player zero-sum Markov games denoted as $\mathcal{M}$. Each environment $M \in \mathcal{M}$ shares the number of episodes $G$, the number of steps $H$ in each episode, the state space $\mathcal{S}$ (with $|\mathcal{S}| = S$), the action spaces $\{\mathcal{A}, \mathcal{B}\}$ (with $|\mathcal{A}| = A$ and $|\mathcal{B}| = B$), and the reward space $\mathcal{R}$. Here $\mathcal{A}$ and $\mathcal{B}$ denote the action spaces of two players, respectively, which are referred to as the max-player and the min-player for convenience.

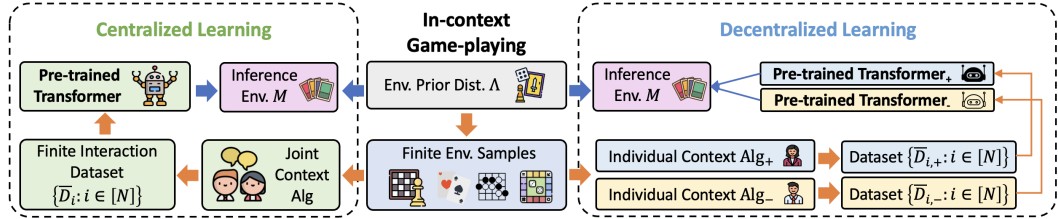

Figure 1: An overall view of the framework, where the in-context game-playing (ICGP) capabilities of transformers are studied in both decentralized and centralized learning settings. The orange arrows denote the supervised pre-training procedure and the blue arrows mark the inference procedure.

Each environment $M = \{\mathbb{T}_M^{h-1}, \mathbb{R}_M^h : h \in [H]\}$ has its transition model $\mathbb{T}_M^h : \mathcal{S} \times \mathcal{A} \times \mathcal{B} \to \Delta(\mathcal{S})$ and reward functions $\mathbb{R}_M^h : \mathcal{A} \times \mathcal{B} \to \Delta(\mathcal{R})$, where $\mathbb{T}_M^0(\cdot)$ denotes the initial state distribution. Particularly, overall $G$ episodes of $H$ steps happen in each environment $M$, with each episode starting at $s^{g,1} \sim \mathbb{T}_M^0(\cdot)$. If the action pair $(a^{g,h}, b^{g,h})$ is taken upon state $s^{g,h}$ at step $h$ in episode $g$, the state is transited to $s^{g,h+1} \sim \mathbb{T}_M^h(\cdot|s^{g,h}, a^{g,h}, b^{g,h})$, and reward $r^{g,h} \sim \mathbb{R}_M^h(s^{g,h}, a^{g,h}, b^{g,h})$ (respectively, $-r^{g,h}$) is collected by the max-player (respectively, the min-player). For simplicity, we assume that the max-player rewards are bounded in $[0, 1]$ and deterministic, i.e., for each $(s, a, b, h)$, there exists $r \in [0, 1]$ such that $\mathbb{R}_M^{g,h}(r|s, a, b) = 1$. Also, the initial state $s^{g,1}$ is assumed to be a fixed one $s^1$, i.e., $\mathbb{T}_M^0(s^1) = 1$.

We further leverage the notation $T := GH$, while using time $t$ and episode-step pair $(g, h)$ in an interleaving manner with $t := (g-1)H + h$. The partial interaction trajectory up to time $t$ is then denoted as $D^t := \{(s^\tau, a^\tau, b^\tau, r^\tau) : \tau \in [t]\}$ and we use the abbreviated notation $D := D^T$. Individually, for the max-player, we denote her observed interaction trajectory up to time $t$ by $D_+^t := \{(s^\tau, a^\tau, r^\tau) : \tau \in [t]\}$ and write $D_+ := D_+^T$ for short. Similarly, for the min-player, we denote $D_-^t := \{(s^\tau, b^\tau, r^\tau) : \tau \in [t]\}$ and $D_- := D_-^T$.

## 2.2 Game-playing Algorithms and Nash Equilibrium

A game-playing algorithm $\texttt{Alg}$ can map a partial trajectory $D^{t-1}$ and state $s^t$ to a distribution over the actions, i.e., $\texttt{Alg}(\cdot, \cdot|D^{t-1}, s^t) \in \Delta(\mathcal{A} \times \mathcal{B})$. If one algorithm $\texttt{Alg}$ is decoupled for the two players (as in the later decentralized setting), we denote it as $\texttt{Alg} = (\texttt{Alg}_+, \texttt{Alg}_-)$, where $\texttt{Alg}_+(\cdot|D_+^{t-1}, s^t) \in \Delta(\mathcal{A})$ and $\texttt{Alg}_-(\cdot|D_-^{t-1}, s^t) \in \Delta(\mathcal{B})$. Given an environment $M$ and an algorithm $\texttt{Alg}$, the distribution over a full trajectory $D$ can be expressed as

$$\mathbb{P}_M^{\texttt{Alg}}(D) = \prod_{t \in [T]} \mathbb{T}_M^{t-1}(s^t|s^{t-1}, a^{t-1}, b^{t-1}) \cdot \texttt{Alg}(a^t, b^t|D^{t-1}, s^t) \cdot \mathbb{R}_M^t(r^t).$$

If further considering an environment prior distribution $\Lambda \in \Delta(\mathcal{M})$ such that $M \sim \Lambda$, the joint distribution of $(M, D)$ is denoted as $\mathbb{P}_\Lambda^{\texttt{Alg}}(D)$, where $M \sim \Lambda(\cdot)$ and $D \sim \mathbb{P}_M^{\texttt{Alg}}(\cdot)$.

For environment $M$ and a game-playing algorithm $\pi$, we define its value function over one episode as $V_M^\pi(s^1) = \mathbb{E}_{D^H \sim \mathbb{P}_M^\pi}[\sum_{t \in [H]} r^t]$. With the marginalized policies of $\pi$ denoted as $(\mu, \nu)$, we define their best responses as

$$\nu_\dagger(\mu) := \arg\min_{\nu'} V_M^{\mu,\nu'}(s^1), \quad \mu_\dagger(\nu) := \arg\max_{\mu'} V_M^{\mu',\nu}(s^1),$$

whose corresponding values are

$$V_M^{\mu,\dagger}(s^1) := V_M^{\mu,\nu_\dagger(\mu)}(s^1), \quad V_M^{\dagger,\nu}(s^1) := V_M^{\mu_\dagger(\nu),\nu}(s^1).$$

With the notion of best responses, the following classical definition of approximate Nash equilibrium (NE) can be introduced [8, 32, 40, 47].

**Definition 2.1** (Approximate Nash equilibrium). *A decoupled policy pair $(\hat{\mu}, \hat{\nu})$ is an $\varepsilon$-approximate Nash equilibrium for environment $M$ if $V_M^{\hat{\mu},\dagger}(s^1) + \varepsilon \geq V_M^{\hat{\mu},\hat{\nu}}(s^1) \geq V_M^{\dagger,\hat{\nu}}(s^1) - \varepsilon$, i.e., the Nash equilibrium gap $V_M^{\dagger,\hat{\nu}}(s^1) - V_M^{\hat{\mu},\dagger}(s^1) \leq 2\varepsilon$.*

For each environment, our learning goal is to approximate its NE policy pair. In other words, we target outputting a policy pair $(\hat{\mu}, \hat{\nu})$ that is $\varepsilon$-approximate NE with an error $\varepsilon$ that is as small as possible, after interacting with the environment for an overall $T$ rounds.

## 2.3 The Transformer Architecture

With the basics of the game-playing environment and the learning target established, we now introduce the transformer architecture [57], which has demonstrated great potential in processing sequential inputs. First, for an input vector $\boldsymbol{x} \in \mathbb{R}^d$, we denote $\sigma_{\mathrm{r}}(\boldsymbol{x}) := \mathrm{ReLU}(\boldsymbol{x}) := \max\{\boldsymbol{x}, 0\} \in \mathbb{R}^d$ as the entry-wise ReLU activation function and $\sigma_{\mathrm{s}}(\boldsymbol{x}) := \mathrm{softmax}(\boldsymbol{x}) \in \mathbb{R}^d$ as the softmax activation function, while using $\sigma(\cdot)$ to refer to a non-specified activation function (i.e., both ReLU and softmax may be used). Then, the masked attention layer and the MLP layer can be defined as follows.

**Definition 2.2** (Masked Attention Layer). *A masked attention layer with $M$ heads is denoted as* $\mathrm{Attn}_{\boldsymbol{\theta}}(\cdot)$ *with parameters* $\boldsymbol{\theta} = \{(\boldsymbol{V}_m, \boldsymbol{Q}_m, \boldsymbol{K}_m)\}_{m \in [M]} \subset \mathbb{R}^{d \times d}$. *On any input sequence* $\boldsymbol{H} = [\boldsymbol{h}_1, \cdots, \boldsymbol{h}_N] \in \mathbb{R}^{d \times N}$, *we have* $\overline{\boldsymbol{H}} = \mathrm{Attn}_{\boldsymbol{\theta}}(\boldsymbol{H}) = [\overline{\boldsymbol{h}}_1, \cdots, \overline{\boldsymbol{h}}_N] \in \mathbb{R}^{d \times N}$, *where*

$$\overline{\boldsymbol{h}}_i = \boldsymbol{h}_i + \sum_{m \in [M]} \frac{1}{i} \sum_{j \in [i]} \sigma_r(\langle \boldsymbol{Q}_m \boldsymbol{h}_i, \boldsymbol{K}_m \boldsymbol{h}_j \rangle) \cdot \boldsymbol{V}_m \boldsymbol{h}_j.$$

**Definition 2.3** (MLP Layer). *An MLP layer with hidden dimension $d'$ is denoted as* $\mathrm{MLP}_{\boldsymbol{\theta}}$ *with parameters* $\boldsymbol{\theta} = (\boldsymbol{W}_1, \boldsymbol{W}_2) \in \mathbb{R}^{d' \times d} \times \mathbb{R}^{d \times d'}$. *On any input sequence* $\boldsymbol{H} = [\boldsymbol{h}_1, \cdots, \boldsymbol{h}_N] \in \mathbb{R}^{d \times N}$, *we have* $\overline{\boldsymbol{H}} = \mathrm{MLP}_{\boldsymbol{\theta}}(\boldsymbol{H}) = [\overline{\boldsymbol{h}}_1, \cdots, \overline{\boldsymbol{h}}_N] \in \mathbb{R}^{d \times N}$, *where*

$$\overline{\boldsymbol{h}}_i = \boldsymbol{h}_i + \boldsymbol{W}_2 \cdot \sigma(\boldsymbol{W}_1 \cdot \boldsymbol{h}_i).$$

The combination of masked attention layers and MLP layers leads to the overall decoder-based transformer architecture studied in this work, as defined in the following.

**Definition 2.4** (Decoder-based Transformer). *An $L$-layer decoder-based transformer, denoted as* $\mathrm{TF}_{\boldsymbol{\theta}}(\cdot)$, *is a composition of $L$ masked attention layers, each followed by an MLP layer and a clip operation:* $\mathrm{TF}_{\boldsymbol{\theta}}(\boldsymbol{H}) = \boldsymbol{H}^{(L)} \in \mathbb{R}^{d \times N}$, *where $\boldsymbol{H}^{(L)}$ is defined iteratively by taking $\boldsymbol{H}^{(0)} = \boldsymbol{H} \in \mathbb{R}^{d \times N}$ and for $l \in [L]$,*

$$\boldsymbol{H}^{(l)} = \mathrm{MLP}_{\boldsymbol{\theta}_{\mathrm{mlp}}^{(l)}} \left( \mathrm{Attn}_{\boldsymbol{\theta}_{\mathrm{mattn}}^{(l)}} \left( \boldsymbol{H}^{(l-1)} \right) \right) \in \mathbb{R}^{d \times N},$$

*where parameter* $\boldsymbol{\theta} = \{(\boldsymbol{\theta}_{\mathrm{mattn}}^{(l)}, \boldsymbol{\theta}_{\mathrm{mlp}}^{(l)}) : l \in [L]\}$ *consists of* $\boldsymbol{\theta}_{\mathrm{mattn}}^{(l)} = \{(\boldsymbol{V}_m^{(l)}, \boldsymbol{Q}_m^{(l)}, \boldsymbol{K}_m^{(l)}) : m \in [M]\} \subset \mathbb{R}^{d \times d}$ *and* $\boldsymbol{\theta}_{\mathrm{mlp}}^{(l)} = (\boldsymbol{W}_1^{(l)}, \boldsymbol{W}_2^{(l)}) \in \mathbb{R}^{d' \times d} \times \mathbb{R}^{d \times d'}$.

We further define the parameter class of transformers as $\Theta_{d,L,M,d',F} := \{\boldsymbol{\theta} = (\boldsymbol{\theta}_{\mathrm{mattn}}^{(1:L)}, \boldsymbol{\theta}_{\mathrm{mlp}}^{(1:L)}) : \|\boldsymbol{\theta}\| \leq F\}$, where the norm of a transformer $\mathrm{TF}_{\boldsymbol{\theta}}$ is denoted as

$$\|\boldsymbol{\theta}\| := \max_{l \in [L]} \left\{ \max_{m \in [M]} \left\{ \|\boldsymbol{Q}_m^{(l)}\|_{\mathrm{op}}, \|\boldsymbol{K}_m^{(l)}\|_{\mathrm{op}} \right\} + \sum_{m \in [M]} \|\boldsymbol{V}_m^{(l)}\|_{\mathrm{op}} + \|\boldsymbol{W}_1^{(l)}\|_{\mathrm{op}} + \|\boldsymbol{W}_2^{(l)}\|_{\mathrm{op}} \right\}.$$

**Other Notations.** The total variation distance between two algorithms $\{\pi, \pi'\}$ upon $D^{t-1} \cup \{s\}$ is denoted as $\mathrm{TV}(\pi, \pi'|D^{t-1}, s) := \mathrm{TV}(\pi(\cdot|D^{t-1}, s), \pi'(\cdot|D^{t-1}, s))$. Also, the notation $x \lesssim y$ indicates that $x$ is a lower or equivalent order term compared with $y$, i.e., $x = \mathcal{O}(y)$, $\tilde{\mathcal{O}}(\cdot)$ hides poly-logarithmic terms in $H, G, S, A, B$, and $\mathrm{poly}(\cdot)$ compactly denotes a polynomial term with respect to the input.

## 3 Decentralized Learning

First, we study the decentralized learning setting, i.e., each player takes actions following her own model independently without observing the opponent's actions, as it better captures the unique game-playing scenario considering in this work. This setting is aligned with the canonical study of normal-form games [20, 53], and has been extended to Markov games in recent years [32, 41, 51]. In the following, we start by introducing the basic setup of supervised pre-training and provide a general performance guarantee relying on a realizability assumption. Then, we provide a constructional result

to demonstrate that the algorithms induced by transformers are rich enough to realize the celebrated V-learning algorithm [32]. With these results, we finally establish that with V-learning providing training data, the pre-trained transformer can effectively approximate NE when interacting with different environments in an in-context fashion.

## 3.1 Supervised Pre-training Results

### 3.1.1 Basic Setups

**Training Dataset.** In the supervised pre-training, we use a context algorithm $\mathtt{Alg}_0$ to collect the offline trajectories. For the decentralized setting, the context algorithm $\mathtt{Alg}_0$ used for data collection is assumed to be consisted of two decoupled algorithms $(\mathtt{Alg}_{+,0}, \mathtt{Alg}_{-,0})$ for the max- and min-players, respectively. With the context algorithm, we consider $N$ i.i.d. offline trajectories $\{\overline{D}_i : i \in [N]\}$ are collected, where $\overline{D}_i := D_i \cup D_i'$ with

$$D_i := \{(s_i^t, a_i^t, b_i^t, r_i^t) : t \in [T]\} \sim \mathbb{P}_\Lambda^{\mathtt{Alg}_0}(\cdot);$$
$$D_i' := \{(a_{i,s}^t, b_{i,s}^t) \sim \mathtt{Alg}_0(\cdot, \cdot | D_i^{t-1}, s) : t \in [T], s \in \mathcal{S}\}.$$

It can be observed that $D_i$ is the commonly considered offline interaction trajectory of $\mathtt{Alg}_0$, while $D_i'$ is the sampled actions of each state $s$ at each step $t$ with $\mathtt{Alg}_0$. Compared with Lin et al. [38], $D_i'$ is an augmented component. We first note that collecting $D_i'$ is relatively easy in practical applications, as it only needs to additionally sample from the distribution $\mathtt{Alg}_0(\cdot, \cdot | D_i^{t-1}, s)$ for each $s \in \mathcal{S}$ (i.e., no additional interactions with the environment). Moreover, the reason to incorporate such an augmentation is to provide additional diverse pre-training data due to the unique game-theoretic environment, with further discussions provided after the later Lemma 3.6. It has also been recognized previously [16, 72] that the data requirement for learning Markov games is typically much stronger than that for single-agent RL.

To facilitate the decentralized training, the overall dataset is further split into two parts: $\{\overline{D}_{+,i} : i \in [N]\}$ and $\{\overline{D}_{-,i} : i \in [N]\}$, where

$$\overline{D}_{+,i} := D_{+,i} \cup D_{+,i}', \quad D_{+,i} := \{(s_i^t, a_i^t, r_i^t) : t \in [T]\}, \quad D_{+,i}' := \{a_{i,s}^t : t \in [T], s \in \mathcal{S}\};$$
$$\overline{D}_{-,i} := D_{-,i} \cup D_{-,i}', \quad D_{-,i} := \{(s_i^t, b_i^t, r_i^t) : t \in [T]\}, \quad D_{-,i}' := \{b_{i,s}^t : t \in [T], s \in \mathcal{S}\}.$$

In other words, $\overline{D}_{i,+}$ denotes the observations of the max-player, while $\overline{D}_{i,-}$ those of the min-layer. Note that neither player can observe the opponent's actions.

**Algorithm Induced by Transformers.** Due to the decentralized nature, two embedding mappings of $d_+$ and $d_-$ dimensions are considered as $\mathtt{h}_+ : \mathcal{S} \cup (\mathcal{A} \times \mathcal{R}) \to \mathbb{R}^{d_+}$ and $\mathtt{h}_- : \mathcal{S} \cup (\mathcal{B} \times \mathcal{R}) \to \mathbb{R}^{d_-}$, together with two transformers $\mathrm{TF}_{\boldsymbol{\theta}_+}$ and $\mathrm{TF}_{\boldsymbol{\theta}_-}$. Taking the max-player's transformer as representative, for trajectory $(D_+^{t-1}, s^t)$, let $\boldsymbol{H}_+ = \mathtt{h}_+(D_+^{t-1}, s^t) = [\mathtt{h}_+(s^1), \mathtt{h}_+(a^1, r^1), \cdots, \mathtt{h}_+(s^t)]$ be the input to $\mathrm{TF}_{\boldsymbol{\theta}_+}$, and the obtained output is $\overline{\boldsymbol{H}}_+ = \mathrm{TF}_{\boldsymbol{\theta}_+}(\boldsymbol{H}_+) = [\overline{\boldsymbol{h}}_{+,1}, \overline{\boldsymbol{h}}_{+,2}, \cdots, \overline{\boldsymbol{h}}_{+,-2}, \overline{\boldsymbol{h}}_{+,-1}]$, which has the same shape as $\boldsymbol{H}_+$. Similarly, the mapping $\mathtt{h}_-$ is used for the min-player's transformer $\mathrm{TF}_{\boldsymbol{\theta}_-}$ to embed trajectory $(D_-^{t-1}, s^t)$.

We further assume that two fixed linear extraction mappings, $\mathtt{A} \in \mathbb{R}^{A \times d_+}$ and $\mathtt{B} \in \mathbb{R}^{B \times d_-}$, are used to induce algorithms $\mathtt{Alg}_{\boldsymbol{\theta}_+}$ and $\mathtt{Alg}_{\boldsymbol{\theta}_-}$ over the action spaces $\mathcal{A}$ and $\mathcal{B}$ of the max- and min-players, respectively, as

$$\mathtt{Alg}_{\boldsymbol{\theta}_+}(\cdot | D_+^{t-1}, s^t) = \mathrm{proj}_\Delta \left( \mathtt{A} \cdot \mathrm{TF}_{\boldsymbol{\theta}_+} \left( \mathtt{h}_+(D_+^{t-1}, s^t) \right)_{-1} \right),$$
$$\mathtt{Alg}_{\boldsymbol{\theta}_-}(\cdot | D_-^{t-1}, s^t) = \mathrm{proj}_\Delta \left( \mathtt{B} \cdot \mathrm{TF}_{\boldsymbol{\theta}_-} \left( \mathtt{h}_-(D_-^{t-1}, s^t) \right)_{-1} \right),$$

$\text{(1)}$

where $\mathrm{proj}_\Delta$ denotes the projection to a probability simplex.

**Training Scheme.** We consider the standard supervised pre-training to maximize the log-likelihood of observing training datasets $\overline{D}_+$ (resp., $\overline{D}_-$) over algorithms $\{\mathtt{Alg}_{\boldsymbol{\theta}_+} : \boldsymbol{\theta}_+ \in \Theta_+\}$ (resp., $\{\mathtt{Alg}_{\boldsymbol{\theta}_-} : \boldsymbol{\theta}_- \in \Theta_-\}$) with $\Theta_+ := \Theta_{d_+, L_+, M_+, d_+', F_+}$ (resp., $\Theta_- := \Theta_{d_-, L_-, M_-, d_-', F_-}$). In particular, the pre-training outputs $\widehat{\boldsymbol{\theta}}_+$ and $\widehat{\boldsymbol{\theta}}_-$ are determined as

$$\widehat{\boldsymbol{\theta}}_+ = \arg\max_{\boldsymbol{\theta}_+ \in \Theta_+} \frac{1}{N} \sum_{i \in [N]} \sum_{t \in [T]} \sum_{s \in \mathcal{S}} \log \left( \mathtt{Alg}_{\boldsymbol{\theta}_+}(a_{i,s}^t | D_{+,i}^{t-1}, s) \right);$$

$$\widehat{\boldsymbol{\theta}}_- = \arg\max_{\boldsymbol{\theta}_- \in \Theta_-} \frac{1}{N} \sum_{i \in [N]} \sum_{t \in [T]} \sum_{s \in \mathcal{S}} \log\left(\mathtt{Alg}_{\boldsymbol{\theta}_-}\left(b_{i,s}^t | D_{-,i}^{t-1}, s\right)\right).$$

### 3.1.2 Theoretical Guarantees

In this section, we provide a generalization guarantee of the algorithms $\mathtt{Alg}_{\widehat{\boldsymbol{\theta}}_+}$ and $\mathtt{Alg}_{\widehat{\boldsymbol{\theta}}_-}$ pre-trained following the scheme introduced above. First, the standard definition regarding the covering number and an assumption of approximate realizability are introduced to facilitate the analysis, which are also leveraged in Lin et al. [38].

**Definition 3.1** (Decentralized Covering Number). *For a class of algorithms* $\{\mathtt{Alg}_{\boldsymbol{\theta}_+} : \boldsymbol{\theta}_+ \in \Theta_+\}$, *we say* $\tilde{\Theta}_+ \subseteq \Theta_+$ *is a* $\rho_+$-*cover of* $\Theta_+$, *if* $\tilde{\Theta}_+$ *is a finite set such that for any* $\boldsymbol{\theta}_+ \in \Theta_+$, *there exists* $\tilde{\boldsymbol{\theta}}_+ \in \tilde{\Theta}_+$ *such that for all* $D_+^{t-1}, s \in \mathcal{S}, t \in [T]$, *it holds that*

$$\left\|\log \mathtt{Alg}_{\tilde{\boldsymbol{\theta}}_+}(\cdot | D_+^{t-1}, s) - \log \mathtt{Alg}_{\boldsymbol{\theta}_+}(\cdot | D_+^{t-1}, s)\right\|_\infty \le \rho_+.$$

*The covering number* $\mathcal{N}_{\Theta_+}(\rho_+)$ *is the minimal cardinality of* $\tilde{\Theta}_+$ *such that* $\tilde{\Theta}_+$ *is a* $\rho_+$-*cover of* $\Theta_+$. *Similarly, we can define the* $\rho_-$-*cover of* $\Theta_-$ *and the covering number* $\mathcal{N}_{\Theta_-}(\rho_-)$.

**Assumption 3.2** (Decentralized Approximate Realizability). *There exist* $\boldsymbol{\theta}_+^* \in \Theta_+$ *and* $\varepsilon_{+,\mathrm{real}} > 0$ *such that for all* $t \in [T], s \in \mathcal{S}, a \in \mathcal{A}$, *it holds that*

$$\log\left(\mathbb{E}_{D \sim \mathbb{P}_\Lambda^{\mathtt{Alg}_0}}\left[\frac{\mathtt{Alg}_{+,0}(a|D_+^{t-1}, s)}{\mathtt{Alg}_{\boldsymbol{\theta}_+^*}(a|D_+^{t-1}, s)}\right]\right) \le \varepsilon_{+,\mathrm{real}}.$$

*We also similarly assume* $\varepsilon_{-,\mathrm{real}}$-*approximate realizability of* $\mathtt{Alg}_{-,0}$ *via* $\mathtt{Alg}_{\boldsymbol{\theta}^*}$ *with* $\boldsymbol{\theta}_-^* \in \Theta_-$.

Then, we can establish the following generalization guarantee on the TV distance between $(\mathtt{Alg}_{\widehat{\boldsymbol{\theta}}_+}, \mathtt{Alg}_{\widehat{\boldsymbol{\theta}}_-})$ and $\mathtt{Alg}_0 = (\mathtt{Alg}_{0,+}, \mathtt{Alg}_{0,-})$, capturing their similarities.

**Theorem 3.3** (Decentralized Pre-training Guarantee). *Let* $\widehat{\boldsymbol{\theta}}_+$ *be the max-player's pre-training output defined in Sec. 3.1.1. Take* $\mathcal{N}_{\Theta_+} = \mathcal{N}_{\Theta_+}(1/N)$ *as in Def. 3.1. Then, under Assumption 3.2, with probability at least* $1 - \delta$, *it holds that*[1]

$$\mathbb{E}_{D \sim \mathbb{P}_\Lambda^{\mathtt{Alg}_0}}\left[\sum_{t \in [T], s \in \mathcal{S}} \mathrm{TV}\left(\mathtt{Alg}_{+,0}, \mathtt{Alg}_{\widehat{\boldsymbol{\theta}}_+} | D_+^{t-1}, s\right)\right] \lesssim TS\sqrt{\varepsilon_{+,\mathrm{real}}} + TS\sqrt{\frac{\log\left(\mathcal{N}_{\Theta_+} TS/\delta\right)}{N}}.$$

*A similar result holds for the min-players' pre-training output* $\widehat{\boldsymbol{\theta}}_-$.

Theorem 3.3 demonstrates that in expectation of the pre-training data distribution, i.e., $\mathbb{P}_\Lambda^{\mathtt{Alg}_0}(D)$, the TV distance between the pre-trained algorithm $\mathtt{Alg}_{\widehat{\boldsymbol{\theta}}_+}$ (resp, $\mathtt{Alg}_{\widehat{\boldsymbol{\theta}}_-}$) and the context algorithm $\mathtt{Alg}_{+,0}$ (resp, $\mathtt{Alg}_{-,0}$) can be bounded via two terms: one from the approximate realizability, i.e., $\varepsilon_{+,\mathrm{real}}$ (resp, $\varepsilon_{-,\mathrm{real}}$), and the other from the limited amount of pre-training trajectories, i.e., finite $N$. While we can diminish the second term via a large pre-training dataset (i.e., sufficient pre-training games), the key question is whether the transformer structure is sufficiently expressive to realize the context algorithm, i.e., having a small $\varepsilon_{+,\mathrm{real}}$, which we affirmatively answer via an example of realizing V-learning [32] in the next subsection.

### 3.2 Realizing V-learning

To demonstrate the capability of transformers in the decentralized game-playing setting, we choose to prove that they can realize the renowned V-learning algorithm [32], the first design that breaks the curse of multiple agents in learning Markov games. Particularly, V-learning leverages techniques from adversarial bandits [5] to perform policy updates without observing the opponent's actions. The details of V-learning are provided in Appendix G.1, where its unique output rule is also elaborated.

In the following theorem, we demonstrate that a transformer can be constructed to exactly perform V-learning with a suitable parameterization. One additional Assumption G.2 on the existence of

---

[1]The covering number $\mathcal{N}_{\Theta_+}$ in Theorem 3.3 (and also the later Theorem C.3) is not concretely discussed in the main paper to ease the presentation. A detailed illustration is provided in Appendix I.

a transformer parameterized by the class of $\Theta_{d,L_D,M_D,d_D,F_D}$ to perform exact division is adopted for the convenience of the proof, while in Appendix G.2, we further demonstrate that the required division operation can be approximated to any arbitrary precision.

**Theorem 3.4.** *With embedding mapping $h_+$ and extraction mapping $A$ defined in Appendix G.3, under Assumption G.2, there exists a transformer $\mathrm{TF}_{\boldsymbol{\theta}_+}$ with*

$$d \lesssim HSA, \quad L \lesssim GHL_D, \quad \max_{l \in [L]} M^{(l)} \lesssim HS^2 + HSA + M_D,$$

$$d' \lesssim G + A + d_D, \quad \|\boldsymbol{\theta}\| \lesssim GH^2S + G^3 + F_D,$$

*which satisfies that for all $D_+^{t-1}, s \in \mathcal{S}, t \in [T]$, $\mathrm{Alg}_{\boldsymbol{\theta}_+}(\cdot|D_+^{t-1}, s) = \mathrm{Alg}_{\text{V-learning}}(\cdot|D_+^{t-1}, s)$. A similar construction $\mathrm{TF}_{\boldsymbol{\theta}_-}$ exists for the min-player's transformer such that for all $D_-^{t-1}, s \in \mathcal{S}, t \in [T]$, $\mathrm{Alg}_{\boldsymbol{\theta}_-}(\cdot|D_-^{t-1}, s) = \mathrm{Alg}_{\text{V-learning}}(\cdot|D_-^{t-1}, s)$.*

The proof of Theorem 3.4 (presented in Appendix G.3) is challenging because V-learning is a model-free design, while UCB-VI [6] studied in Lin et al. [38] and VI-ULCB [8] later presented in Sec. 4.2 are both model-based ones. We believe this result deepens our understanding of the capability of pre-trained transformers in decision-making, i.e., they can realize both model-based and model-free designs, showcasing their further potentials.

More specifically, with the embedded trajectory as the input, the model-based philosophy is natural for the masked attention mechanism, i.e., the value computation at each step is directly over all raw inputs in previous steps. Thus, the construction procedure is straightforward as (input$_1$, input$_2$, ..., input$_t$) $\to$ (value$_1$, value$_2$, ..., value$_t$). However, the model-free designs are different, where value computation at one step requires previous values (instead of raw inputs). In other words, the construction procedure is a recursive one as (input$_1$, input$_2$, ..., input$_t$) $\to$ (value$_1$, input$_2$, ..., input$_t$) $\to$ (value$_1$, value$_2$, ..., input$_t$) $\to$ ... $\to$ (value$_1$, value$_2$, ..., value$_t$), whose realization requires carefully crafted constructions.

## 3.3 The Overall ICGP Capablity

Finally, built upon the obtained results, the following theorem demonstrates the ICGP capability of pre-trained transformers in the decentralized setting.

**Theorem 3.5.** *Let $\Theta_+$ and $\Theta_-$ be the classes of transformers satisfying the requirements in Theorem 3.4 and $(\mathrm{Alg}_{+,0}, \mathrm{Alg}_{-,0})$ both be V-learning. Let $(\hat{\mu}, \hat{\nu})$ be the output policies via the output rule of V-learning. Denoting $\mathrm{Alg}_{\widehat{\boldsymbol{\theta}}} = (\mathrm{Alg}_{\widehat{\boldsymbol{\theta}}_+}, \mathrm{Alg}_{\widehat{\boldsymbol{\theta}}_-})$ and $\mathcal{N}_\Theta = \mathcal{N}_{\Theta_+}\mathcal{N}_{\Theta_-}$. Then, with probability at least $1 - \delta$, it holds that*

$$\mathbb{E}_{D \sim \mathbb{P}_\Lambda^{\mathrm{Alg}_{\widehat{\boldsymbol{\theta}}}}} \left[ V_M^{\dagger,\hat{\nu}}(s^1) - V_M^{\hat{\mu},\dagger}(s^1) \right] \lesssim \sqrt{\frac{H^5 S(A \vee B)\log(SABT)}{G}} + THS\sqrt{\frac{\log(TS\mathcal{N}_\Theta/\delta)}{N}}.$$

With the obtained upper bound on the approximation error of NE, Theorem 3.5 demonstrates the ICGP capability of pre-trained transformers as the algorithms $\mathrm{Alg}_{\widehat{\boldsymbol{\theta}}_+}$ and $\mathrm{Alg}_{\widehat{\boldsymbol{\theta}}_-}$ are fixed during interactions with varying inference games (i.e., no parameter updates). When prompted by the interaction trajectory in the current game, they are capable of deciding the future interaction strategy and finally provide policy pairs that are approximate NE. We further note that during both pre-training and inference, each player's transformer takes inputs of its own observed trajectories, but not the opponent's actions, which reflects the decentralized requirement. Moreover, the approximation error in Theorem 3.5 depends on $A \vee B$ instead of $AB$ as in the later Theorem 4.2, evidencing the benefits of decentralized learning.

**Proof Sketch.** The proof of Theorem 3.5 (presented in Appendix H) rely on the following, decomposition, where $\mathbb{E}_0[\cdot]$ and $\mathbb{E}_{\widehat{\boldsymbol{\theta}}}[\cdot]$ are with respect to $\mathbb{P}_\Lambda^{\mathrm{Alg}_0}$ and $\mathbb{P}_\Lambda^{\mathrm{Alg}_{\widehat{\boldsymbol{\theta}}}}$, respectively:

$$\mathbb{E}_{\widehat{\boldsymbol{\theta}}} \left[ V_M^{\dagger,\hat{\nu}}(s^1) - V_M^{\hat{\mu},\dagger}(s^1) \right] = \mathbb{E}_0 \left[ V_M^{\dagger,\hat{\nu}}(s^1) - V_M^{\hat{\mu},\dagger}(s^1) \right] \tag{2}$$
$$+ \mathbb{E}_{\widehat{\boldsymbol{\theta}}} \left[ V_M^{\dagger,\hat{\nu}}(s^1) \right] - \mathbb{E}_0 \left[ V_M^{\dagger,\hat{\nu}}(s^1) \right] + \mathbb{E}_0 \left[ V_M^{\hat{\mu},\dagger}(s^1) \right] - \mathbb{E}_{\widehat{\boldsymbol{\theta}}} \left[ V_M^{\hat{\mu},\dagger}(s^1) \right],.$$

It can be observed that the first decomposed term is the performance of the considered V-learning, which can be obtained following Jin et al. [32] as in Theorem G.1.

Then, the remaining terms concern the performance of the policy pair $(\hat{\mu}, \hat{\nu})$ learned from $\texttt{Alg}_0$ and $\texttt{Alg}_{\hat{\boldsymbol{\theta}}}$ *against their own best responses*, respectively. This is drastically different from the consideration in Lin et al. [38], which only bounds the performance of the learned policies, i.e.,

$$\mathbb{E}_0 \left[ V_M^{\hat{\mu}, \hat{\nu}}(s^1) \right] - \mathbb{E}_{\hat{\boldsymbol{\theta}}} \left[ V_M^{\hat{\mu}, \hat{\nu}}(s^1) \right].$$

The involvement of best responses complicates the analysis. After careful treatments in Appendix H, we obtain the following lemma to characterize these terms.

**Lemma 3.6.** *For any two decentralized algorithms $\texttt{Alg}_\alpha$ and $\texttt{Alg}_\beta$, we denote their performed policies for episode $g$ are $(\mu_\alpha^g, \nu_\alpha^g)$ and $(\mu_\beta^g, \nu_\beta^g)$, and their final output policies via the output rule of V-learning (see Appendix G.1) are $(\hat{\mu}_\alpha, \hat{\nu}_\alpha)$ and $(\hat{\mu}_\beta, \hat{\nu}_\beta)$. For $\{\hat{\mu}_\alpha, \hat{\mu}_\beta\}$, it holds that*

$$\mathbb{E}_\alpha \left[ V_M^{\hat{\mu}_\alpha, \dagger}(s^1) \right] - \mathbb{E}_\beta \left[ V_M^{\hat{\mu}_\beta, \dagger}(s^1) \right] \lesssim H \cdot \sum_{t \in [T], s \in \mathcal{S}} \mathbb{E}_\alpha \left[ \mathrm{TV} \left( \mu_\alpha^t, \mu_\beta^t | D_+^{t-1}, s \right) \right]$$

$$+ H \cdot \sum_{t \in [T], s \in \mathcal{S}} \mathbb{E}_\alpha \left[ \mathrm{TV} \left( \nu_\alpha^t, \nu_\beta^t | D_-^{t-1}, s \right) \right],$$

*where $\mathbb{E}_\alpha[\cdot]$ and $\mathbb{E}_\beta[\cdot]$ are with respect to $\mathbb{P}_\Lambda^{\texttt{Alg}_\alpha}$ and $\mathbb{P}_\Lambda^{\texttt{Alg}_\beta}$. A similar result holds for $\{\hat{\nu}_\alpha, \hat{\nu}_\beta\}$.*

With Lemma 3.6, we can incorporate Theorem 3.3 to upper bound the TV distance between $\texttt{Alg}_0$ and $\texttt{Alg}_{\hat{\boldsymbol{\theta}}}$, which together with Theorem 3.4 establish $\varepsilon_{\mathrm{real},+} = \varepsilon_{\mathrm{real},-} = 0$ in this case, leading to the desired performance guarantee in Theorem 3.5. We here further note that the effectiveness of Theorem 3.3 in capturing the bound in Lemma 3.6 over all $s \in \mathcal{S}$ credits to the augmented dataset $D'$, which provides diverse data of all $s \in \mathcal{S}$.

# 4 Centralized Learning

In this section, we discuss the scenario of centralized learning, i.e., training one joint model to control both players' interactions. This is also known as the self-play setting [8, 9, 33, 40, 67]. Following a similar procedure as the decentralized discussions, we first provide supervised pre-training guarantees and then demonstrate that transformers are capable of realizing the renowned VI-ULCB algorithm [8]. It is thus established that in a centralized learning setting, the pre-trained transformer can still effectively perform ICGP and approximate NE.

## 4.1 Supervised Pre-training Results

The same training dataset $\{\overline{D}_i : i \in [N]\}$ as in Section 3.1.1 is considered. As the centralized setting is studied here, no further split of the dataset is needed. Moreover, one $d$-dimensional mapping $\mathtt{h} : \mathcal{S} \cup (\mathcal{A} \times \mathcal{B} \times \mathcal{R}) \to \mathbb{R}^d$ can be designed to embed the trajectories, and the induced algorithm $\texttt{Alg}_{\boldsymbol{\theta}}(\cdot, \cdot | D^t, s^t)$ from the transformer $\mathrm{TF}_{\boldsymbol{\theta}}$ can be obtained via a fixed linear extraction mapping $\mathtt{E}$ similarly as Eqn. (1). Finally, the MLE training is performed with $\Theta := \Theta_{d,L,M,d',F}$ as $\widehat{\boldsymbol{\theta}} = \arg\max_{\boldsymbol{\theta} \in \Theta} \frac{1}{N} \sum_{i \in [N]} \sum_{t \in [T]} \sum_{s \in \mathcal{S}} \log \left( \texttt{Alg}_{\boldsymbol{\theta}}(a_{i,s}^t, b_{i,s}^t | D_i^{t-1}, s) \right)$.

Then, a generalization guarantee of $\texttt{Alg}_{\hat{\boldsymbol{\theta}}}$ can be provided similarly as Theorem 3.3, which is deferred to Theorem C.3. This centralized result also implies that the pre-trained centralized algorithm performs similarly as the context algorithm, with errors caused by the approximate realizability and the finite pre-training data.

## 4.2 Realizing VI-ULCB

The VI-ULCB algorithm [8] is one of the first provably efficient centralized learning designs for Markov games. It extends the key idea of using confidence bounds to incorporate uncertainties from stochastic bandits and MDPs [4, 6] to handle competitive environments, and has further inspired many extensions in Markov games [9, 30, 33, 40, 65]. As VI-ULCB is highly representative, we choose it as the example for realization in the centralized setting to demonstrate the capability of transformers.

To make VI-ULCB practically implementable, we adopt an approximate CCE solver powered by multiplicative weight update (MWU) in the place of its originally required general-sum NE solver

(which is computationally demanding). This modification is demonstrated as provably efficient in later works [9, 40, 65]. Then, the following result illustrates that a transformer can be constructed to exactly perform the MWU-version of VI-ULCB.

**Theorem 4.1.** *With embedding mapping $\hbar$ and extraction mapping $\mathtt{E}$ defined in Appendix D.2, there exists a transformer* $\mathrm{TF}_{\boldsymbol{\theta}}$ *with*

$$d \lesssim HS^2AB, \qquad L \lesssim GHS, \quad \max_{l \in [L]} M^{(l)} \lesssim HS^2AB,$$
$$d' \lesssim G^2HS^2AB, \quad \|\boldsymbol{\theta}\| \lesssim HS^2AB + G^3 + GH,$$

*which satisfies that for all $D^{t-1}, s \in \mathcal{S}, t \in [T]$, $\mathtt{Alg}_{\boldsymbol{\theta}}(\cdot, \cdot | D^{t-1}, s) = \mathtt{Alg}_{VI\text{-}ULCB}(\cdot, \cdot | D^{t-1}, s)$.*

One observation from the proof of Theorem 4.1 (presented in Appendix D.2) is that transformer layers can perform MWU so that an approximate CCE can be found, which is not reported in Lin et al. [38] and further demonstrates the in-context learning capability of transformers in playing normal-form games (since MWU is one of the most basic designs).

### 4.3 The Overall ICGP Capability

With Theorem 4.1 showing VI-ULCB can be exactly realized (i.e., $\varepsilon_{\mathrm{real}} = 0$ in Assumption C.2), we can further prove an overall upper bound of the approximation error of NE by $\mathtt{Alg}_{\widehat{\boldsymbol{\theta}}}$ via the following theorem, demonstrating the ICGP capability of transformers.

**Theorem 4.2.** *Let $\Theta$ be the class of transformers satisfying the requirements in Theorem 4.1 and $\mathtt{Alg}_0$ be VI-ULCB. For all $(g, h, s) \in [G] \times [H] \times \mathcal{S}$, let $(\mu^{g,h}(\cdot|s), \nu^{g,h}(\cdot|s))$ be the marginalized policies of $\mathtt{Alg}_{\widehat{\boldsymbol{\theta}}}(\cdot, \cdot | D^{t-1}, s)$. Then, with probability at least $1 - \delta$, it holds that*

$$\mathbb{E}_{\mathbb{P}_{\Lambda}^{\mathtt{Alg}_{\widehat{\boldsymbol{\theta}}}}, \hat{\mu}, \hat{\nu}} \left[ V_M^{\dagger, \hat{\nu}}(s^1) - V_M^{\hat{\mu}, \dagger}(s^1) \right] \lesssim \sqrt{\frac{H^4 S^2 AB \log(SABT)}{G}} + THS \sqrt{\frac{\log(TS\mathcal{N}_\Theta / \delta)}{N}},$$

*where $\hat{\mu}$ and $\hat{\nu}$ are uniformly sampled as $\hat{\mu} \sim \mathtt{Unif}\{\mu^1, \cdots, \mu^G\}$ and $\hat{\nu} \sim \mathtt{Unif}\{\nu^1, \cdots, \nu^G\}$, with $\mu^g := \{\mu^{g,h}(\cdot|s) : (h, s) \in [H] \times \mathcal{S}\}$ and $\nu^g := \{\nu^{g,h}(\cdot|s) : (h, s) \in [H] \times \mathcal{S}\}$, and $\mathbb{E}_{\hat{\mu}, \hat{\nu}}$ is with respect to the process of policy sampling.*

This result demonstrates the ICGP capability of pre-trained transformers in the centralized setting, complementing the discussions in the decentralized results.

## 5 Empirical Experiments

Experiments are performed on two-player zero-sum normal-form games ($H = 1$) and Markov games ($H = 2$), with the decentralized EXP3 [5] (which can be viewed as a one-step V-learning) and the centralized VI-ULCB being the context algorithms as demonstrations, respectively. Additional experimental setups and details can be found in Appendix J. It can be first observed from Fig. 2 that, the transformers pre-trained with $N = 20$ games performs better on the inference tasks than the ones pre-trained with $N = 10$ games. This observation empirically validates the theoretical result that more pre-training games benefit the final game-playing performance during inference (i.e., the $\sqrt{1/N}$-dependencies established in Theorems 3.5 and 4.2). Moreover, when the number of pre-training games is sufficient (i.e., $N = 20$ in Fig. 2), the obtained transformers can indeed learn to approximate NE in an in-context manner (i.e., having a gradually decaying NE gap), and also the obtained performance is similar to the context algorithm, i.e., EXP3 or VI-ULCB. These observations provide empirical pieces of evidence to support the ICGP capabilities of pre-trained transformers, motivating and validating the theoretical analyses performed in this work.

## 6 Related Works

**In-context Learning.** Since GPT-3 [11] demonstrates the ICL capability of pre-trained transformers, growing attention has been paid to this direction. In particular, an emerging line of work targets providing a deeper understanding of the fundamental mechanism behind the ICL capability [3, 24, 29, 31, 37, 44, 59, 60, 64, 66, 68], where many interesting results have been obtained. In particular, transformers have been shown to be capable of performing in-context gradient descent so that

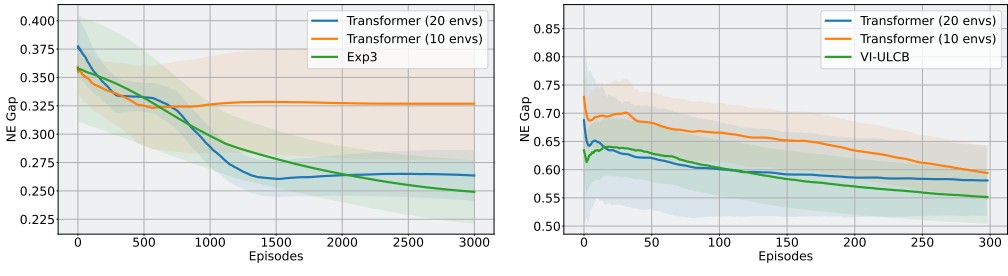

(a) Decentralized Comparisons With EXP3.    (b) Centralized Comparisons With VI-ULCB.

Figure 2: Comparisons of Nash equilibrium (NE) gaps over episodes in both decentralized and centralized learning scenarios, averaged over 10 inference games.

varying optimization-based algorithms can be realized [2, 3, 7, 26, 59]. Also, Giannou et al. [27] demonstrates that looped transformers can emulate basic computing blocks, whose combinations can lead to complex operations.

This work is more focused on the in-context reinforcement learning (ICRL) capability of pre-trained transformers, as demonstrated in Grigsby et al. [28], Laskin et al. [34], Lee et al. [35], Wang et al. [62]. The recent work by Lin et al. [38] initiates the theoretical investigation of this topic. In particular, Lin et al. [38] provides generalization guarantees after pre-training in the single-agent RL scenario, and further constructs transformers to realize provably efficient single-agent bandits and RL algorithms (in particular, LinUCB [1], Thompson sampling [54], UCB-VI [6]). This work extends Lin et al. [38] to the domain of competitive multi-agent RL by studying the in-context game-playing setting. A recent concurrent work [36] also touches upon the in-context game-playing capability of pre-trained transformers, while focusing on practical aspects and exploiting different opponents.

**Competitive Multi-agent RL.** The study of RL in the competitive multi-agent domain has a long and fruitful history [10, 47, 49, 58, 70]. In recent years, researchers have gained a deeper theoretical understanding of this topic. The centralized setting (also known as self-play) has been investigated in Bai and Jin [8], Bai et al. [9], Cui et al. [17], Huang et al. [30], Jin et al. [33], Liu et al. [40], Wang et al. [63], Xiong et al. [67], Zhang et al. [69], and this work focuses on the representative VI-ULCB design [9]. On the other hand, decentralized learning is more challenging, and the major breakthrough is made by V-learning [9, 32, 41, 51], which is thus adopted as the target algorithm in this work.

## 7    Conclusions

This work investigated the in-context game-playing (ICGP) capabilities of pre-trained transformers, broadening the research scope of in-context RL from the single-agent scenario to the more challenging multi-agent competitive games. Focusing on the classical two-player zero-sum Markov games, a general learning framework was first introduced, laying down a solid ground for this and later studies. Through concrete theoretical results, this work further demonstrated that in both decentralized and centralized learning settings, properly pre-trained transformers are capable of approximating Nash equilibrium in an in-context manner. As a key part of the proof, concrete sets of parameterization were provided to demonstrate that the transformer architecture can realize two famous designs, decentralized V-learning and centralized VI-ULCB. Empirical experiments further validate the theoretical results (especially that pre-trained transformers can indeed approximate NE in an in-context manner) and motivate future studies on this under-explored research direction.

## Acknowledgments and Disclosure of Funding

The work of CSs and KY was supported in part by the US National Science Foundation (NSF) under awards CNS-2002902, ECCS- 2029978, ECCS-2143559, and CNS-2313110, and the Bloomberg Data Science Ph.D. Fellowship. The work of JY was supported in part by the US NSF under awards CNS-1956276 and CNS-2114542.

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

# A  An Overview of the Appendix

In this section, an overview of the appendix is provided. First, additional discussions are presented in Appendix B, which cover broader impacts of this work and our thoughts on the future directions.

Then, the proof details omitted in this main paper are provided. While the decentralized learning setting is the major focus in the main paper, the discussions and proofs for the centralized learning setting are first provided to facilitate the presentation and understanding as the decentralized learning setting is more challenging.

- The supervised pre-training guarantee (i.e., Theorem C.3) for the centralized learning setting is proved in Appendix C. The details and realization of VI-ULCB (i.e., Theorem 4.1) are presented in Appendix D. The proofs for the overall performance guarantee (i.e., Theorem 4.2) can be found in Appendix E.
- Subsequently, the proofs for the supervised pre-training guarantee (i.e., Theorem 3.3) in the decentralized learning setting are provided in Appendix F. Appendix G contains the details and realization of V-learning (i.e., Theorem 3.4). The overall performance guarantee (i.e., Theorem 3.5) is proved in Appendix H.
- A detailed discussion of the covering number is provided in Appendix I.

Finally, the setups and details of the experiments presented in Sec. 5 are reported in Appendix J.

# B  Additional Discussions

## B.1  Broader Impacts

This work mainly provides a theoretical understanding of the in-context game-playing capabilities of pre-trained transformers, broadening the research scope of in-context reinforcement learning from single-agent settings to multi-agent competitive games. Due to its theoretical nature, we do not foresee major negative societal impacts; however, we still would like to acknowledge the need for responsible usage of the practical implementation of the proposed game-playing transformers due to their high capability in various environments.

## B.2  Limitations and Future Works

The research direction of in-context game-playing is currently under-explored, and we believe that there are many interesting topics to be further investigated.

• *Different game forms and game-solving algorithms.* This work mainly studies the classical two-player zero-sum Markov games, which can be viewed as the most basic form of competitive games, and has particular focuses on constructing transformers to realize V-learning [32] and VI-ULCB [8]. The cooperative games, on the other hand, are conceptually more similar to the single-agent setting [38]. There are more complicated game forms [46, 48], e.g., the mixed cooperative-competitive games, which requires different game-solving algorithms. We believe the framework built in this work is beneficial to further explore the capabilities of pre-trained models in game-theoretical scenarios.

• *Pre-training dataset construction.* This work considering the pre-training dataset is collected from the context algorithm with an additional augmentation. First, while the current proofs rely on the augmentation, it will be an interesting topic to understand whether it is necessary. As mentioned Sec. 3.1.1, learning in Markov games typically require more diverse data than learning in single-agent settings; however, the minimum requirement to perform effective pre-training is worth further exploring. Moreover, in the study of single-agent RL [35], it is shown that pre-training with the data from the optimal policy is more efficient, which is further theoretically investigated in Lin et al. [38]. In multi-agent competitive games, it is currently unclear whether similar strategies can be incorporated, e.g., pre-training with data collected by Nash equilibrium policies or best responses for certain other policies.

• *Large-scale empirical evaluations.* Due to the limited computation resources, the experiments reported in Sec. 5 are relatively small-scale compared with the current size of practically adopted transformers. It would be an important and interesting direction to further evaluate the ICGP capabilities of pre-trained transformers in large-scale experiments and practical game-theoretic

applications. Also, the training dynamics are also worth further investigation, e.g., the sudden shifts in learning effectiveness reported by Reddy [45].

Besides these directions, from the theoretical perspective, we believe it would be valuable to investigate how to extend the current study on the tabular setting to incorporate function approximation, where we conjecture it is sufficient for the pre-training dataset to cover information of certain representative states and actions (e.g., a well-coverage of the feature space) [72]. Another attractive theoretical question is how to learn from a dataset collected by multiple context algorithms. From the practical perspective, a future study on the impact of the practical training recipe (e.g., model structure, training hyperparameters, etc.) would be desirable to bring additional insights.

## C    Proofs for the Centralized Supervised Pre-training Guarantees

First, the definition of the centralized covering number and an assumption of centralized approximate realizability are introduced to facilitate the analysis, which are also leveraged in Lin et al. [38].

**Definition C.1** (Centralized Covering Number). *For a class of algorithms $\{\mathtt{Alg}_{\boldsymbol{\theta}} : \boldsymbol{\theta} \in \Theta\}$, we say $\tilde{\Theta} \subseteq \Theta$ is a $\rho$-cover of $\Theta$, if $\tilde{\Theta}$ is a finite set such that for any $\boldsymbol{\theta} \in \Theta$, there exists $\tilde{\boldsymbol{\theta}} \in \tilde{\Theta}$ such that for all $D^{t-1}, s \in \mathcal{S}, t \in [T]$, it holds that*

$$\left\| \log \mathtt{Alg}_{\tilde{\boldsymbol{\theta}}}(\cdot, \cdot | D^{t-1}, s) - \log \mathtt{Alg}_{\boldsymbol{\theta}}(\cdot, \cdot | D^{t-1}, s) \right\|_{\infty} \leq \rho.$$

*The covering number $\mathcal{N}_{\Theta}(\rho)$ is the minimal cardinality of $\tilde{\Theta}$ such that $\tilde{\Theta}$ is a $\rho$-cover of $\Theta$.*

**Assumption C.2** (Centralized Approximate Realizability). *There exist $\boldsymbol{\theta}^* \in \Theta$ and $\varepsilon_{\text{real}} > 0$ such that for all $s \in \mathcal{S}, t \in [T], (a, b) \in \mathcal{A} \times \mathcal{B}$, it holds that*

$$\log \left( \mathbb{E}_{D \sim \mathbb{P}_{\Lambda}^{\mathtt{Alg}_0}} \left[ \frac{\mathtt{Alg}_0(a, b | D^{t-1}, s)}{\mathtt{Alg}_{\boldsymbol{\theta}^*}(a, b | D^{t-1}, s)} \right] \right) \leq \varepsilon_{\text{real}}.$$

Then, the following pre-training guarantee can be established.

**Theorem C.3** (Centralized Pre-training Guarantee). *Let $\widehat{\boldsymbol{\theta}}$ be the maximum likelihood pre-training output. Take $\mathcal{N}_{\Theta} = \mathcal{N}_{\Theta}(1/N)$ as in Definition C.1. Then, under Assumption C.2, with probability at least $1 - \delta$, it holds that*

$$\mathbb{E}_{D \sim \mathbb{P}_{\Lambda}^{\mathtt{Alg}_0}} \left[ \sum_{t \in [T], s \in \mathcal{S}} \mathrm{TV}(\mathtt{Alg}_0, \mathtt{Alg}_{\widehat{\boldsymbol{\theta}}} | D^{t-1}, s) \right] \lesssim TS\sqrt{\varepsilon_{\text{real}}} + TS\sqrt{\frac{\log\left(\mathcal{N}_{\Theta} TS/\delta\right)}{N}}.$$

*Proof of Theorem C.3.* This proof extends that of Theorem 6 in Lin et al. [38] to the multi-agent scenario. Let $\tilde{\Theta}$ be a $\rho$-covering set of $\Theta$ with covering number $\mathcal{N}_{\Theta} = \mathcal{N}_{\Theta}(\rho)$ as defined in Definition C.1. With Lemma 15 in Lin et al. [38], we can obtain that for any $\boldsymbol{\theta} \in \Theta$, there exists $\tilde{\boldsymbol{\theta}} \in \tilde{\Theta}$ such that for all $D^{t-1}, t \in [T]$ and $s \in \mathcal{S}$,

$$\mathrm{TV}\left( \mathtt{Alg}_{\tilde{\boldsymbol{\theta}}}, \mathtt{Alg}_{\boldsymbol{\theta}} | D^{t-1}, s) \right) \leq \rho$$

For $m \in [\mathcal{N}_{\Theta}], t \in [T], i \in [N], s \in \mathcal{S}$, we define that

$$\ell_{i,m}^t(s) := \log \left( \frac{\mathtt{Alg}_0\left(a_{i,s}^t, b_{i,s}^t | D_i^{t-1}, s\right)}{\mathtt{Alg}_{\boldsymbol{\theta}_m}\left(a_{i,s}^t, b_{i,s}^t | D_i^{t-1}, s\right)} \right).$$

According to Lemma 14 in Lin et al. [38], with probability at least $1 - \delta$, for all $m \in [\mathcal{N}_{\Theta}], t \in [T], s \in \mathcal{S}$, it holds that

$$\frac{1}{2} \sum_{i \in [N]} \ell_{i,m}^t(s) + \log(\mathcal{N}_{\Theta} TS/\delta) \geq \sum_{i \in [N]} -\log \left( \mathbb{E}\left[ \exp\left( -\frac{\ell_{i,m}^t(s)}{2} \right) \right] \right).$$

Furthermore, it can be established that

$$\mathbb{E}\left[ \exp\left( -\frac{\ell_{i,m}^t(s)}{2} \right) | D_i^{t-1} \right] = \mathbb{E}\left[ \sqrt{\frac{\mathtt{Alg}_{\boldsymbol{\theta}_m}\left(a_{i,s}^t, b_{i,s}^t | D_i^{t-1}, s\right)}{\mathtt{Alg}_0\left(a_{i,s}^t, b_{i,s}^t | D_i^{t-1}, s\right)}} | D_i^{t-1} \right]$$

$$= \sum_{(a,b)\in\mathcal{A}\times\mathcal{B}} \sqrt{\mathrm{Alg}_{\boldsymbol{\theta}_m}\left(a,b|D_i^{t-1},s\right)\mathrm{Alg}_0\left(a,b|D_i^{t-1},s\right)},$$

which implies that

$$\mathbb{E}\left[\exp\left(-\frac{\ell_{i,m}^t(s)}{2}\right)\right] = 1 - \frac{1}{2}\cdot\mathbb{E}\left[\sum_{(a,b)\in\mathcal{A}\times\mathcal{B}}\left(\sqrt{\mathrm{Alg}_{\boldsymbol{\theta}_m}\left(a,b|D_i^{t-1},s\right)} - \sqrt{\mathrm{Alg}_0\left(a,b|D_i^{t-1},s\right)}\right)^2\right]$$

$$\leq 1 - \frac{1}{2}\cdot\mathbb{E}\left[\mathrm{TV}\left(\mathrm{Alg}_{\boldsymbol{\theta}_m},\mathrm{Alg}_0|D_i^{t-1},s\right)^2\right],$$

where the inequality is from the fact that the Hellinger distance is smaller than the TV distance.

Then, we can obtain that for any $\boldsymbol{\theta}$ covered by $\boldsymbol{\theta}_m$, it holds that

$$\left(\mathbb{E}_D\left[\mathrm{TV}\left(\mathrm{Alg}_0,\mathrm{Alg}_{\boldsymbol{\theta}}|D^{t-1},s\right)\right]\right)^2$$

$$\leq \left(\mathbb{E}_D\left[\mathrm{TV}\left(\mathrm{Alg}_0,\mathrm{Alg}_{\boldsymbol{\theta}_m}|D^{t-1},s\right)\right] + \mathbb{E}_D\left[\mathrm{TV}\left(\mathrm{Alg}_{\boldsymbol{\theta}_m},\mathrm{Alg}_{\boldsymbol{\theta}}|D^{t-1},s\right)\right]\right)^2$$

$$\leq 2\left(\mathbb{E}_D\left[\mathrm{TV}\left(\mathrm{Alg}_0,\mathrm{Alg}_{\boldsymbol{\theta}_m}|D^{t-1},s\right)\right]\right)^2 + 2\left(\mathbb{E}_D\left[\mathrm{TV}\left(\mathrm{Alg}_{\boldsymbol{\theta}_m},\mathrm{Alg}_{\boldsymbol{\theta}}|D^{t-1},s\right)\right]\right)^2$$

$$\leq 2\mathbb{E}_D\left[\mathrm{TV}\left(\mathrm{Alg}_0,\mathrm{Alg}_{\boldsymbol{\theta}_m}|D^{t-1},s\right)^2\right] + 2\rho^2$$

$$\leq 4 - 4\mathbb{E}\left[\exp\left(-\frac{\ell_{i,m}^t(s)}{2}\right)\right] + 2\rho^2$$

$$\leq -4\log\left(\mathbb{E}\left[\exp\left(-\frac{\ell_{i,m}^t(s)}{2}\right)\right]\right) + 2\rho^2,$$

which further implies that

$$N\sum_{s\in\mathcal{S}}\sum_{t\in[T]}\left(\mathbb{E}_D\left[\mathrm{TV}\left(\mathrm{Alg}_0,\mathrm{Alg}_{\boldsymbol{\theta}}|D^{t-1},s\right)\right]\right)^2$$

$$\leq -4\sum_{s\in\mathcal{S}}\sum_{t\in[T]}\sum_{i\in[N]}\log\left(\mathbb{E}\left[\exp\left(-\frac{\ell_{i,m}^t(s)}{2}\right)\right]\right) + 2NST\rho^2$$

$$\leq 2\sum_{s\in\mathcal{S}}\sum_{t\in[T]}\sum_{i\in[N]}\ell_{i,m}^t(s) + 2NST\rho^2 + 4ST\log(\mathcal{N}_\Theta TS/\delta)$$

$$= 2\sum_{s\in\mathcal{S}}\sum_{t\in[T]}\sum_{i\in[N]}\log\left(\frac{\mathrm{Alg}_0\left(a_{i,s}^t,b_{i,s}^t|D_i^{t-1},s\right)}{\mathrm{Alg}_{\boldsymbol{\theta}_m}\left(a_{i,s}^t,b_{i,s}^t|D_i^{t-1},s\right)}\right) + 2NST\rho^2 + 4ST\log(\mathcal{N}_\Theta TS/\delta)$$

$$\leq 2\sum_{s\in\mathcal{S}}\sum_{t\in[T]}\sum_{i\in[N]}\log\left(\frac{\mathrm{Alg}_0\left(a_{i,s}^t,b_{i,s}^t|D_i^{t-1},s\right)}{\mathrm{Alg}_{\boldsymbol{\theta}}\left(a_{i,s}^t,b_{i,s}^t|D_i^{t-1},s\right)}\right) + 2NST\rho^2 + 2NST\rho + 4ST\log(\mathcal{N}_\Theta TS/\delta).$$

Thus, for the obtained $\widehat{\boldsymbol{\theta}}$, with probability at least $1-\delta$, it holds that

$$N\sum_{s\in\mathcal{S}}\sum_{t\in[T]}\left(\mathbb{E}_D\left[\mathrm{TV}\left(\mathrm{Alg}_0,\mathrm{Alg}_{\widehat{\boldsymbol{\theta}}}|D^{t-1},s\right)\right]\right)^2$$

$$\leq 2\sum_{s\in\mathcal{S}}\sum_{t\in[T]}\sum_{i\in[N]}\log\left(\frac{\mathrm{Alg}_0\left(a_{i,s}^t,b_{i,s}^t|D_i^{t-1},s\right)}{\mathrm{Alg}_{\widehat{\boldsymbol{\theta}}}\left(a_{i,s}^t,b_{i,s}^t|D_i^{t-1},s\right)}\right) + 2NST\rho^2 + 2NST\rho + 4ST\log(\mathcal{N}_\Theta TS/\delta)$$

$$\leq 2\sum_{s\in\mathcal{S}}\sum_{t\in[T]}\sum_{i\in[N]}\log\left(\frac{\mathrm{Alg}_0\left(a_{i,s}^t,b_{i,s}^t|D_i^{t-1},s\right)}{\mathrm{Alg}_{\boldsymbol{\theta}^*}\left(a_{i,s}^t,b_{i,s}^t|D_i^{t-1},s\right)}\right) + 2NST\rho^2 + 2NST\rho + 4ST\log(\mathcal{N}_\Theta TS/\delta)$$

$$\leq 2\sum_{s\in\mathcal{S}}\sum_{t\in[T]}\sum_{i\in[N]}\log\left(\mathbb{E}\left[\frac{\mathrm{Alg}_0\left(a_{i,s}^t,b_{i,s}^t|D_i^{t-1},s\right)}{\mathrm{Alg}_{\boldsymbol{\theta}^*}\left(a_{i,s}^t,b_{i,s}^t|D_i^{t-1},s\right)}\right]\right) + ST\log(TS/\delta)$$

$$+ 2NST\rho^2 + 2NST\rho + 4ST\log(\mathcal{N}_\Theta TS/\delta)$$

$$\leq 2NST\varepsilon_{\text{real}} + ST\log(TS/\delta) + 2NST\rho^2 + 2NST\rho + 4ST\log(\mathcal{N}_\Theta TS/\delta),$$

Further by Cauchy-Schwarz inequality, we can obtain that

$$\sum_{s\in\mathcal{S}}\sum_{t\in[T]}\left(\mathbb{E}_D\left[\text{TV}\left(\text{Alg}_0, \text{Alg}_{\widehat{\theta}}|D^{t-1}, s\right)\right]\right)$$

$$\leq \sqrt{ST\sum_{s\in\mathcal{S}}\sum_{t\in[T]}\left(\mathbb{E}_D\left[\text{TV}\left(\text{Alg}_0, \text{Alg}_{\widehat{\theta}}|D^{t-1}, s\right)\right]\right)^2}$$

$$= O\left(ST\sqrt{\varepsilon_{\text{real}}} + ST\sqrt{\frac{\log(\mathcal{N}_\Theta ST)}{N}} + ST\sqrt{\rho} + ST\rho\right)$$

Taking $\rho = 1/N$ concludes the proof. $\qquad\square$

## D Proofs for Realizing VI-ULCB

### D.1 Details of MWU VI-ULCB

We here note one distinction from the VI-ULCB design considered in this work from its vanilla version proposed in Bai and Jin [8], which makes VI-ULCB practically implementable. Especially, Bai and Jin [8] requires an oracle solver that can provide the exact NE policy pair $(\mu^*, \nu^*)$ from any two general input payoff matrices $(\overline{Q}, \underline{Q}) \in \mathbb{R}^{A\times B} \times \mathbb{R}^{A\times B}$. However, it is known that approximating such a general-sum NE is computationally hard (specifically, PPAD-complete) [19], which makes this vanilla version impractical. Luckily, later studies [9, 40, 65] have demonstrated that a solver finding one weaker notation of equilibrium, i.e., coarse correlated equilibrium (CCE), is already sufficient. Following these recent results, we replace the NE solver with an approximate CCE solver in VI-ULCB. Moreover, we consider finding such CCEs via no-regret learning.[2] In particular, both players virtually run *multiplicative weight update (MWU)* (which is also known as *Hedge*), a classical no-regret algorithm, with payoff matrices $(\overline{Q}, \underline{Q})$ for several rounds; then, an aggregated policy can be generated as an approximate CCE. The details of the VI-ULCB algorithm are provided in Alg. 1.

More specifically, we consider that an approximate CCE solver is adopted such that with each pair of inputs $(\overline{Q}^h(s,\cdot,\cdot), \underline{Q}^h(s,\cdot,\cdot))$, we can obtain an $\varepsilon_{\text{CCE}}$-approximate CCE policy $\pi^h(\cdot,\cdot|s)$ which satisfies that

$$\mathbb{E}_{(a,b)\sim\pi^h(\cdot,\cdot|s)}\left[\overline{Q}(s,a,b)\right] \geq \max_{a^*\in\mathcal{S}}\mathbb{E}_{(a,b)\sim\pi^h(\cdot,\cdot|s)}\left[\overline{Q}(s,a^*,b)\right] - \varepsilon_{\text{CCE}}$$

$$\mathbb{E}_{(a,b)\sim\pi^h(\cdot,\cdot|s)}\left[\underline{Q}(s,a,b)\right] \leq \min_{b^*\in\mathcal{S}}\mathbb{E}_{(a,b)\sim\pi^h(\cdot,\cdot|s)}\left[\underline{Q}(s,a,b^*)\right] + \varepsilon_{\text{CCE}}.$$

We also specifically choose to obtain such approximate CCEs by having both players (virtually) perform MWU against each other. The details of MWU are included in Alg. 2, where we use the following notations to denote normalized losses:

$$\overline{L}^h(s,a,b) := \frac{H - \overline{Q}^h(s,a,b)}{H}, \quad \underline{L}^h(s,a,b) := \frac{H - \underline{Q}^h(s,a,b)}{H}.$$

Standard online learning results [13] guarantee that using learning rates $\eta_A = \sqrt{\log(A)/N_{\text{MWU}}}$ and $\eta_B = \sqrt{\log(B)/N_{\text{MWU}}}$, after $N_{\text{MWU}}$ rounds of MWU, the policy

$$\pi^h(\cdot,\cdot|s) = \frac{1}{N_{\text{MWU}}}\sum_{n\in[N]}\mu_n^h(\cdot|s)\nu_n^h(\cdot|s)$$

is an $\varepsilon_{\text{CCE}}$-approximate CCE policy, with

$$\varepsilon_{\text{CCE}} = H\sqrt{\frac{\log(A+B)}{N_{\text{MWU}}}}.$$

---

[2]Another common method to find CCEs is through linear programming (LP). It will be an interesting direction to investigate whether transformers can be LP solvers, which is however out of the scope of this paper.

---

**Algorithm 1** VI-ULCB

---

1: **Initialize**: for any $(s, a, b, h)$, $\overline{Q}^h(s, a, b) \leftarrow H$, $\underline{Q}^h(s, a, b) \leftarrow 0$, $N^h(s, a, b) \leftarrow 0$, $N^h(s, a, b, s') \leftarrow 0$
2: **for** episode $g = 1, \cdots, G$ **do**
3:    **for** $(s, a, b) \in \mathcal{S} \times \mathcal{A} \times \mathcal{B}$ **do**
4:       Compute $\overline{Q}^h(s, a, b) \leftarrow \min \left\{ \hat{r}^h(s, a, b) + \left[ \hat{\mathbb{P}}^h \overline{V}^{h+1} \right](s, a, b) + c \sqrt{\frac{H^2 S \iota}{N^h(s^h, a^h, b^h)}}, H \right\}$
5:       Compute $\underline{Q}^h(s, a, b) \leftarrow \max \left\{ \hat{r}^h(s, a, b) + \left[ \hat{\mathbb{P}}^h \underline{V}^{h+1} \right](s, a, b) - c \sqrt{\frac{H^2 S \iota}{N^h(s^h, a^h, b^h)}}, 0 \right\}$
6:    **end for**
7:    **for** $s \in \mathcal{S}$ **do**
8:       Update $\pi_h(\cdot, \cdot | s) \leftarrow \varepsilon_N$-approximate CCE $\left( \overline{Q}^h(s, \cdot, \cdot), \underline{Q}^h(s, \cdot, \cdot) \right)$ solved by $N$-round MWU
9:       Compute $\overline{V}^h(s) \leftarrow \sum_{a,b} \pi^h(a, b | s) \overline{Q}^h(s, a, b)$
10:      Compute $\underline{V}^h(s) \leftarrow \sum_{a,b} \pi^h(a, b | s) \underline{Q}^h(s, a, b)$
11:   **end for**
12:   **for** step $h = 1, \cdots, H$ **do**
13:      Take action $(a^h, b^h) \sim \pi^h(\cdot, \cdot | s^h)$
14:      Observe reward $r^h$ and next state $s^{h+1}$
15:      Update $N_h(s^h, a^h, b^h)$ and $N^h(s^h, a^h, b^h, s^{h+1})$
16:      Update $\hat{\mathbb{P}}^h(\cdot | s^h, a^h, b^h)$ and $\hat{r}^h(s^h, a^h, b^h)$
17:   **end for**
18: **end for**

---

**Algorithm 2** MWU

---

1: **Input**: learning rates $\eta_A = \sqrt{\log(A)/N}$ and $\eta_B = \sqrt{\log(B)/N}$, action sets $\mathcal{A}$ and $\mathcal{B}$ with size $A$ and $B$, loss matrices $\overline{L}^h(s, \cdot, \cdot)$ and $\underline{L}^h(s, \cdot, \cdot)$
2: **Initialize**: cumulative loss $O_+ \leftarrow \mathbf{0}_A$ and $O_- \leftarrow \mathbf{0}_A$
3: **for** $n = 1, \cdots, N$ **do**
4:    Compute $\mu_n^h(\cdot | s) \leftarrow \sigma_s(-\eta_A O_+) \in \Delta(\mathcal{A})$
5:    Compute $\nu_n^h(\cdot | s) \leftarrow \sigma_s(-\eta_B O_-) \in \Delta(\mathcal{B})$
6:    Observe vectors $o_{+,n} \in \mathbb{R}^A$ with $o_{+,n}(a) = \nu_n^h(\cdot | s) \cdot \overline{L}^h(s, a, \cdot)$
7:    Observe vectors $o_{-,n} \in \mathbb{R}^B$ with $o_{-,n}(b) = \mu_n^h(\cdot | s) \cdot \underline{L}^h(s, \cdot, b)$
8:    Update $O_+ = O_+ + o_{+,n}$ and $O_- = O_- + o_{-,n}$
9: **end for**
10: **Output**: policy $\sum_{n \in [N]} \mu_n^h(\cdot | s) \nu_n^h(\cdot | s) / N$

---

Furthermore, for a certain bounded $\varepsilon_{\mathrm{CCE}}$, Xie et al. [65] demonstrated that the performance degradation can still be controlled. Following the results therein, the following theorem can be easily established.

**Theorem D.1** (Modified from Theorem 2 from Bai and Jin [8])**.** *With probability at least $1 - \delta$, in any environment $M$, the output policies $\{ (\mu^g, \nu^g) : g \in [G] \}$ from the MWU-version of VI-ULCB satisfy that*

$$\sum_{g \in [G]} V_M^{\dagger, \nu^g}(s^1) - V_M^{\mu^g, \dagger}(s^1) = O \left( \sqrt{H^3 S^2 ABT \log(SABT/\delta)} + T \varepsilon_{\mathrm{CCE}} \right).$$

With $\delta = 1/T$, to have a non-dominant loss caused by the approximate CCE solver, we can choose $N_{\mathrm{MWU}} = G$. Then, for any environment $M$, it holds that

$$\mathbb{E}_{D \sim \mathbb{P}_M^{\mathrm{VI\text{-}ULCB}}} \left[ \sum_{g \in [G]} V_M^{\dagger, \nu^g}(s^1) - V_M^{\mu^g, \dagger}(s^1) \right] = O \left( \sqrt{H^3 S^2 ABT \log(SABT)} \right).$$

## D.2 Proof of Theorem 4.1: The Realization Construction

### D.2.1 Embeddings and Extraction Mapping

We consider each episode of observations to be embedded in $2H$ tokens. In particular, for each $t \in [T]$, we construct that

$$
\boldsymbol{h}_{2t-1} = \mathrm{h}(s^{g,h}) = \begin{bmatrix} \boldsymbol{0}_A \\ \boldsymbol{0}_B \\ 0 \\ \hdashline s^{g,h} \\ \hdashline \boldsymbol{0}_{AB} \\ \boldsymbol{0} \\ \mathbf{pos}_{2t-1} \end{bmatrix} =: \begin{bmatrix} \boldsymbol{h}^{\mathrm{pre},a}_{2t-1} \\ \hdashline \boldsymbol{h}^{\mathrm{pre},b}_{2t-1} \\ \hdashline \boldsymbol{h}^{\mathrm{pre},c}_{2t-1} \\ \hdashline \boldsymbol{h}^{\mathrm{pre},d}_{2t-1} \end{bmatrix},
$$

$$
\boldsymbol{h}_{2t} = \mathrm{h}(a^{g,h}, b^{g,h}, r^{g,h}) = \begin{bmatrix} a^{g,h} \\ b^{g,h} \\ r^{g,h} \\ \hdashline \boldsymbol{0}_S \\ \hdashline \boldsymbol{0}_{AB} \\ \boldsymbol{0} \\ \mathbf{pos}_{2t} \end{bmatrix} =: \begin{bmatrix} \boldsymbol{h}^{\mathrm{pre},a}_{2t} \\ \hdashline \boldsymbol{h}^{\mathrm{pre},b}_{2t} \\ \hdashline \boldsymbol{h}^{\mathrm{pre},c}_{2t} \\ \hdashline \boldsymbol{h}^{\mathrm{pre},d}_{2t} \end{bmatrix},
$$

where $s^{g,h}, a^{g,h}, b^{g,h}$ are represented via one-hot embedding. The positional embedding $\mathbf{pos}_i$ is defined as

$$
\mathbf{pos}_i := \begin{bmatrix} g \\ h \\ t \\ \boldsymbol{e}_h \\ v_i \\ i \\ i^2 \\ 1 \end{bmatrix},
$$

where $\boldsymbol{e}_h$ is a one-hot vector with the $h$-th element being 1 and $v_i := \mathbb{1}\{\boldsymbol{h}^a_i = \boldsymbol{0}\}$ denote the tokens that do not embed actions and rewards.

In summary, for observations $D^{t-1} \cup \{s^t\}$, we obtain the following tokens of length $2t - 1$:

$$
\boldsymbol{H} := h(D^{t-1}, s^t) = [\boldsymbol{h}_1, \boldsymbol{h}_2, \cdots, \boldsymbol{h}_{2t-1}] = [\mathrm{h}(s^1), \mathrm{h}(a^1, b^1, r^1), \cdots, \mathrm{h}(s^t)].
$$

With the above input $\boldsymbol{H}$, the transformer outputs $\overline{\boldsymbol{H}} = \mathrm{TF}_{\boldsymbol{\theta}_+}(\boldsymbol{H})$ of the same size as $\boldsymbol{H}$. The extraction mapping $\mathrm{E}$ is directly set to satisfy the following

$$
\mathrm{E} \cdot \overline{\boldsymbol{h}}_{-1} = \mathrm{E} \cdot \overline{\boldsymbol{h}}_{2t-1} = \overline{\boldsymbol{h}}^c_{2t-1} \in \mathbb{R}^{AB},
$$

i.e., the part $c$ of the output tokens is used to store the learned policy.

### D.2.2 An Overview of the Proof

In the following, for the convenience of notations, we will consider step $t + 1$, i.e., with observations $D^t \cup \{s^{t+1}\}$. Given an input token matrix

$$
\boldsymbol{H} = h(D^t, s^{t+1}) = [\boldsymbol{h}_1, \boldsymbol{h}_2, \cdots, \boldsymbol{h}_{2t+1}],
$$

we construct a transformer to perform the following steps

$$
\begin{bmatrix} \boldsymbol{h}^{\mathrm{pre},a}_{2t+1} \\ \boldsymbol{h}^{\mathrm{pre},b}_{2t+1} \\ \boldsymbol{h}^{\mathrm{pre},c}_{2t+1} \\ \boldsymbol{h}^{\mathrm{pre},d}_{2t+1} \end{bmatrix} \xrightarrow{\text{step 1}} \begin{bmatrix} \boldsymbol{h}^{\mathrm{pre},\{a,b,c\}}_{2t+1} \\ N^h(s,a,b) \\ N^h(s,a,b,s') \\ N^h(s,a,b)r^h(s,a,b) \\ \star \\ \boldsymbol{0} \\ \mathbf{pos}_{2t+1} \end{bmatrix} \xrightarrow{\text{step 2}} \begin{bmatrix} \boldsymbol{h}^{\mathrm{pre},\{a,b,c\}}_{2t+1} \\ \hat{\mathbb{P}}^h(s'|s,a,b) \\ \hat{r}^h(s,a,b) \\ \star \\ \boldsymbol{0} \\ \mathbf{pos}_{2t+1} \end{bmatrix} \xrightarrow{\text{step 3}} \begin{bmatrix} \boldsymbol{h}^{\mathrm{pre},\{a,b,c\}}_{2t+1} \\ \overline{Q}^h(s,a,b) \\ \underline{Q}^h(s,a,b) \\ \star \\ \boldsymbol{0} \\ \mathbf{pos}_{2t+1} \end{bmatrix}
$$

$$\xrightarrow{\text{step 4}} \begin{bmatrix} \boldsymbol{h}_{2t+1}^{\text{pre},\{a,b,c\}} \\ \pi^h(a,b|s) \\ \star \\ \mathbf{0} \\ \mathbf{pos}_{2t+1} \end{bmatrix} \xrightarrow{\text{step 5}} \begin{bmatrix} \boldsymbol{h}_{2t+1}^{\text{pre},\{a,b,c\}} \\ \overline{V}^h(s) \\ \underline{V}^h(s) \\ \star \\ \mathbf{0} \\ \mathbf{pos}_{2t+1} \end{bmatrix} \xrightarrow{\text{step 6}} \begin{bmatrix} \boldsymbol{h}_{2t+1}^{\text{pre},\{a,b\}} \\ \pi^{h+1}(\cdot,\cdot|s^{h+1}) \\ \boldsymbol{h}_{2t+1}^{\text{post},d} \end{bmatrix} := \begin{bmatrix} \boldsymbol{h}_{2t+1}^{\text{post},a} \\ \boldsymbol{h}_{2t+1}^{\text{post},b} \\ \boldsymbol{h}_{2t+1}^{\text{post},c} \\ \boldsymbol{h}_{2t+1}^{\text{post},d} \\ \boldsymbol{h}_{2t+1}^{\text{post},d} \end{bmatrix},$$

where we use $N^h(s,a,b)$, $N^h(s,a,b,s')$, $N^h(s,a,b)r^h(s,a,b)$, $\hat{\mathbb{P}}^h(s'|s,a,b)$, $\hat{r}^h(s,a,b)$, $\overline{Q}^h(s,a,b)$, $\underline{Q}^h(s,a,b)$ and $\pi^h(a,b|s)$ to denote their entire vectors over $h \in [H], s \in \mathcal{S}, a \in \mathcal{A}, b \in \mathcal{B}, s' \in \mathcal{S}$. The notation $\star$ denotes other quantities in $\boldsymbol{h}_{2(t+1)}^d$.

The following provides a sketch of the proof.

Step 1 There exists an attention-only transformer $\text{TF}_{\boldsymbol{\theta}}$ to complete Step 1 with
$$L = O(1), \quad \max_{l \in [L]} M^{(l)} = O(HS^2AB), \quad \|\boldsymbol{\theta}\| = O(HG + HS^2AB).$$

Step 2 There exists a transformer $\text{TF}_{\boldsymbol{\theta}}$ to complete Step 2 with
$$L = O(1), \quad \max_{l \in [L]} M^{(l)} = O(HS^2AB), \quad d' = O(G^2HS^2AB)$$
$$\|\boldsymbol{\theta}\| = O(HS^2AB + G^3 + GH).$$

Step 3 There exists a transformer $\text{TF}_{\boldsymbol{\theta}}$ to complete Step 3 with
$$L = O(H), \quad \max_{l \in [L]} M^{(l)} = O(SAB), \quad d'^{(l)} = O(SAB), \quad \|\boldsymbol{\theta}\| = O(H + SAB).$$

Step 4 There exists a transformer $\text{TF}_{\boldsymbol{\theta}}$ to complete Step 4 with
$$L = O(GHS), \quad \max_{l \in [L]} M^{(l)} = O(AB), \quad d'^{(l)} = O(AB), \quad \|\boldsymbol{\theta}\| = O(H + AB).$$

Step 5 There exists an attention-only transformer $\text{TF}_{\boldsymbol{\theta}}$ to complete Step 5 with
$$L = O(1), \quad \max_{l \in [L]} M^{(l)} = O(HS), \quad \|\boldsymbol{\theta}\| = O(HS).$$

Step 6 There exists an attention-only transformer $\text{TF}_{\boldsymbol{\theta}}$ to complete Step 6 with
$$L = O(1), \quad \max_{l \in [L]} M^{(l)} = O(HS), \quad \|\boldsymbol{\theta}\| = O(HS + GH).$$

Thus, the overall transformer $\text{TF}_{\boldsymbol{\theta}}$ can be summarized as
$$L = O(GHS), \quad \max_{l \in [L]} M^{(l)} = O(HS^2AB), \quad d' = O(G^2HS^2AB),$$
$$\|\boldsymbol{\theta}\| = O(HS^2AB + G^3 + GH).$$

Also, from the later construction, we can observe that $\log(R) = \tilde{\mathcal{O}}(1)$.

### D.2.3 Proof of Step 1: Update $N^h(s,a,b)$, $N^h(s,a,b,s')$ and $N^h(s,a,b)r^h(s,a,b)$

This can be similarly completed by an attention-only transformer constructed in Step 1 of realizing UCB-VI in Lin et al. [38].

### D.2.4 Proof of Step 2: Update $\hat{\mathbb{P}}^h(s'|s,a,b)$ and $\hat{r}(s,a,b)$

This can be similarly completed by a transformer constructed in Step 2 of realizing UCB-VI in Lin et al. [38].

### D.2.5 Proof of Step 3: Compute $\overline{Q}^h(s'|s,a,b)$ and $\underline{Q}^h(s'|s,a,b)$

The computation of $\overline{Q}^h(s'|s,a,b)$ can be similarly completed by a transformer constructed in Step 3 of realizing UCB-VI in Lin et al. [38]. The $\underline{Q}$ part can also be obtained by modifying a few plus signs to minuses.

### D.2.6 Proof of Step 4: Compute CCE

This is the most challenging part of realizing the VI-ULCB design, which distinguishes it from the single-agent algorithms, e.g., UCB-VI [6]. As mentioned in Appendix D.1, we obtain an approximate CCE via virtually playing MWU. In the following, for one tuple $(s, h)$, we prove that one transformer can be constructed to perform a one-step MWU update with that

$$L = O(1), \quad \max_{l \in [L]} M^{(l)} = O(AB), \quad d' = O(AB), \quad \|\boldsymbol{\theta}\| = O(H + AB).$$

To obtain this result, we construct a transformer to perform the following computation from inputs to output for all $t' \leq t$:

$$\boldsymbol{h}_{2t} = \begin{bmatrix} \mathbf{0} \end{bmatrix}, \quad \boldsymbol{h}_{2t+1} = \begin{bmatrix} \overline{L}^h(s, \cdot, \cdot) \\ \underline{L}^h(s, \cdot, \cdot) \\ \sum_{\tau < n} o_{+,\tau} \\ \sum_{\tau < n} o_{-,\tau} \\ \mu_n(\cdot|s) \\ \nu_n(\cdot|s) \\ \sum_{\tau \leq n} \mu_\tau(\cdot)\nu_\tau(\cdot) \\ \mathbf{0} \end{bmatrix}$$

$$\xrightarrow{\text{compute}} \overline{\boldsymbol{h}}_{2t} = \begin{bmatrix} \mathbf{0} \end{bmatrix}, \quad \overline{\boldsymbol{h}}_{2t+1} = \begin{bmatrix} \overline{L}^h(s, \cdot, \cdot) \\ \underline{L}^h(s, \cdot, \cdot) \\ \sum_{\tau < n+1} o_{+,\tau} \\ \sum_{\tau < n+1} o_{-,\tau} \\ \mu_{n+1}(\cdot|s) \\ \nu_{n+1}(\cdot|s) \\ \sum_{\tau \leq n+1} \mu_\tau(\cdot)\nu_\tau(\cdot) \\ \mathbf{0} \end{bmatrix}.$$

Note that here we again use the notations $\overline{L}^h(s, \cdot, \cdot) = \frac{H - \overline{Q}^h(s, \cdot, \cdot)}{H}$ and $\underline{L}^h(s, \cdot, \cdot) = \frac{H - Q^h(s, \cdot, \cdot)}{H}$ to denote the normalized losses. It can be seen that this computation can be performed via one ReLU MLP layer.

**Step 4.1: Get $o_{+,n}$ and $o_{-,n}$.**

First, we can construct that for all $t' \leq t$

$$\boldsymbol{Q}_{a,1}^{(1)} \boldsymbol{h}_{2t'} = \begin{bmatrix} v_{2t'} - 1 \\ \mathbf{0} \\ t' \\ H \end{bmatrix}, \quad \boldsymbol{Q}_{a,1}^{(1)} \boldsymbol{h}_{2t'+1} = \begin{bmatrix} v_{2t'+1} - 1 \\ \nu_n(\cdot|s) \\ t' + 1 \\ H \end{bmatrix};$$

$$\boldsymbol{K}_{a,1}^{(1)} \boldsymbol{h}_{2t'} = \begin{bmatrix} H \\ \mathbf{0} \\ -H \\ t' \end{bmatrix}, \quad \boldsymbol{K}_{a,1}^{(1)} \boldsymbol{h}_{2t'+1} = \begin{bmatrix} H \\ \overline{L}^h(a, \cdot|s) \\ -H \\ t' + 1 \end{bmatrix},$$

$$\boldsymbol{V}_{a,1}^{(1)} \boldsymbol{h}_{2t'} = 2t', \quad \boldsymbol{V}_{a,1}^{(1)} \boldsymbol{h}_{2t'+1} = 2t' + 1.$$

With ReLU activation, this constructed transformer leads to updates that $\boldsymbol{h}_{2t'}^d = 0$ and $\boldsymbol{h}_{2t'+1}^d = o_{+,n}(a)$. With another $A - 1$ paralleling heads, the whole vector $o_{+,n}$ can be computed. Similarly, with $B$ more paralleling heads, the whole vector $o_{-,n}$ can be computed.

Then, with one ReLU MLP layer, we can obtain that

$$\boldsymbol{h}_{2t'}^d = \mathbf{0}, \quad \boldsymbol{h}_{2t'+1}^d = \begin{bmatrix} \sum_{\tau < n} o_{+,\tau} \\ \sum_{\tau < n} o_{-,\tau} \end{bmatrix} + \boldsymbol{W}_2^{(1)} \sigma_r \left( \boldsymbol{W}_1^{(1)} \boldsymbol{h}_{2t'+1} \right) = \begin{bmatrix} \sum_{\tau < n+1} o_{+,\tau} \\ \sum_{\tau < n+1} o_{-,\tau} \end{bmatrix}.$$

The required transformer can be summarized as

$$L = 1, \quad M^{(1)} = O(A + B), \quad d' = O(A + B), \quad \|\boldsymbol{\theta}\| = O(H).$$

**Step 4.2: Get $\mu_{n+1}(\cdot|s)$ and $\nu_{n+1}(\cdot|s)$.**

We can construct a softmax MLP layer such that

$$\boldsymbol{W}_1^{(1)}\boldsymbol{h}_{2t'} = \boldsymbol{0}_A, \quad \sigma_{\mathrm{s}}(\boldsymbol{W}_1^{(1)}\boldsymbol{h}_{2t'}) = \frac{1}{A}\cdot\boldsymbol{1}_A$$

$$\boldsymbol{W}_1^{(1)}\boldsymbol{h}_{2t'+1} = \left[ \; -\eta_A \sum_{\tau<n+1} o_{+,n}^{t'} \; \right], \quad \sigma_{\mathrm{s}}(\boldsymbol{W}_1^{(1)}\boldsymbol{h}_{2t'+1}) = \left[ \; \mu_{n+1}(\cdot|s) \; \right],$$

where $\eta_A = \sqrt{\log(A)/G}$. Thus, $\mu_{n+1}(\cdot|s)$ can be provided. Similarly, another MLP layer $\{W_1^{(2)}, W_2^{(2)}\}$ with softmax activation can provide $\nu_{n+1}(\cdot|s)$. The current output can be expressed as

$$\boldsymbol{h}_{2t'}^d = \left[ \begin{array}{c} \boldsymbol{0} \\ \frac{1}{A}\cdot\boldsymbol{1}_A \\ \frac{1}{B}\cdot\boldsymbol{1}_B \end{array} \right], \quad \boldsymbol{h}_{2t'+1}^d = \left[ \begin{array}{c} \mu_n(\cdot|s) \\ \nu_n(\cdot|s) \\ \mu_{n+1}(\cdot|s) \\ \nu_{n+1}(\cdot|s) \end{array} \right].$$

We can further construct one more attention layer as

$$\boldsymbol{Q}_1^{(4)}\boldsymbol{h}_{2t'} = \left[ \begin{array}{c} 1-v_{2t'} \\ t' \\ 1 \end{array} \right], \quad \boldsymbol{Q}_1^{(3)}\boldsymbol{h}_{2t'+1} = \left[ \begin{array}{c} 1-v_{2t'+1} \\ t'+1 \\ 1 \end{array} \right];$$

$$\boldsymbol{K}_1^{(3)}\boldsymbol{h}_{2t'} = \left[ \begin{array}{c} 1 \\ -1 \\ t' \end{array} \right], \quad \boldsymbol{K}_1^{(3)}\boldsymbol{h}_{2t'+1} = \left[ \begin{array}{c} 1 \\ -1 \\ t'+1 \end{array} \right];$$

$$\boldsymbol{V}_1^{(3)}\boldsymbol{h}_{2t'} = 2t'\cdot\left[ \begin{array}{c} -\frac{1}{A}\cdot\boldsymbol{1}_A \\ -\frac{1}{B}\cdot\boldsymbol{1}_B \end{array} \right], \quad \boldsymbol{V}_1^{(3)}\boldsymbol{h}_{2t'+1} = (2t'+1)\cdot\left[ \begin{array}{c} -\frac{1}{A}\cdot\boldsymbol{1}_A \\ -\frac{1}{B}\cdot\boldsymbol{1}_B \end{array} \right].$$

With ReLU activation, this construction would result in that

$$\boldsymbol{h}_{2t'} = \left[ \begin{array}{c} \frac{1}{A}\cdot\boldsymbol{1}_A \\ \frac{1}{B}\cdot\boldsymbol{1}_B \end{array} \right] + \left[ \begin{array}{c} -\frac{1}{A}\cdot\boldsymbol{1}_A \\ -\frac{1}{B}\cdot\boldsymbol{1}_B \end{array} \right] = \boldsymbol{0}, \quad \boldsymbol{h}_{2t'+1} = \left[ \begin{array}{c} \mu_{n+1}(\cdot|s) \\ \nu_{n+1}(\cdot|s) \end{array} \right] + \boldsymbol{0} = \left[ \begin{array}{c} \mu_{n+1}(\cdot|s) \\ \nu_{n+1}(\cdot|s) \end{array} \right].$$

At last, one ReLU MLP layer $\{\boldsymbol{W}_1^{(3)}, \boldsymbol{W}_2^{(3)}\}$ can be constructed to replace $\mu_n(\cdot|s)$ and $\nu_n(\cdot|s)$ with $\mu_{n+1}(\cdot|s)$ and $\nu_{n+1}(\cdot|s)$:

$$\boldsymbol{W}_2^{(3)}\sigma_{\mathrm{r}}\left(\boldsymbol{W}_1^{(3)}\boldsymbol{h}_{2t'}\right) = \boldsymbol{0}, \quad \boldsymbol{W}_2^{(3)}\sigma_{\mathrm{r}}\left(\boldsymbol{W}_1^{(3)}\boldsymbol{h}_{2t'+1}\right) = \left[ \begin{array}{c} \mu_{n+1}(\cdot|s) - \mu_n(\cdot|s) \\ \nu_{n+1}(\cdot|s) - \nu_n(\cdot|s) \end{array} \right].$$

The required transformer can be summarized as

$$L = 3, \quad \max_{l\in[L]} M^{(l)} = O(1), \quad d' = O(A+B), \quad \|\boldsymbol{\theta}\| = O(\sqrt{\log(A)/G} + \sqrt{\log(B)/G} + 1).$$

**Step 4.3: Get $\sum_{\tau\le n+1} \mu_\tau(\cdot|s)\nu_\tau(\cdot|s)/N$.**

We can construct

$$\boldsymbol{Q}_1^{(1)}\boldsymbol{h}_{2t'} = \left[ \begin{array}{c} 0 \\ t' \\ 1 \end{array} \right], \quad \boldsymbol{Q}_1^{(1)}\boldsymbol{h}_{2t'+1} = \left[ \begin{array}{c} \mu_{n+1}(a|s) \\ t'+1 \\ 1 \end{array} \right];$$

$$\boldsymbol{K}_1^{(1)}\boldsymbol{h}_{2t'} = \left[ \begin{array}{c} 0 \\ -1 \\ t' \end{array} \right], \quad \boldsymbol{K}_1^{(1)}\boldsymbol{h}_{2t'+1} = \left[ \begin{array}{c} \nu_{n+1}(b|s) \\ -1 \\ t'+1 \end{array} \right];$$

$$\boldsymbol{V}_1^{(5)}\boldsymbol{h}_{2t'} = 2t', \quad \boldsymbol{V}_1^{(5)}\boldsymbol{h}_{2t'+1} = 2t'+1.$$

With ReLU activation, this construction can update that

$$\boldsymbol{h}_{2t'} = 0, \quad \boldsymbol{h}_{2t'+1} = \mu_{n+1}(a|s)\nu_{n+1}(b|s).$$

Using an overall $AB$ paralleling heads, we can then obtain $\mu_{n+1}(\cdot|s)\nu_{n+1}(\cdot|s)$. Then, with a ReLU MLP layer $\{W_1^{(5)}, W_2^{(5)}\}$, we can obtain $\sum_{\tau\le n+1} \mu_\tau(\cdot|s)\nu_\tau(\cdot|s)/N$.

The required transformer can be summarized as

$$L = 1, \quad M^{(1)} = O(AB), \quad d' = O(AB), \quad \|\boldsymbol{\theta}\| = O(AB).$$

Combining all the sub-steps provides proof of a one-step MWU update. The same transformer can be stacked for $G$ times, which completes the $G$-step MWU.

### D.2.7 Proof of Step 5: Compute $\overline{V}^h(s)$ and $\underline{V}^h(s)$

We can construct that

$$\boldsymbol{Q}_1^{(1)}\boldsymbol{h}_{2t'} = \begin{bmatrix} 0 \\ t' \\ H \end{bmatrix}, \quad \boldsymbol{Q}_1^{(1)}\boldsymbol{h}_{2t'+1} = \begin{bmatrix} \pi^h(\cdot,\cdot|s) \\ t'+1 \\ H \end{bmatrix};$$

$$\boldsymbol{K}_1^{(1)}\boldsymbol{h}_{2t'} = \begin{bmatrix} 0 \\ -H \\ t' \end{bmatrix}, \quad \boldsymbol{K}_1^{(1)}\boldsymbol{h}_{2t'+1} = \begin{bmatrix} \overline{Q}^h(s,\cdot,\cdot) \\ -H \\ t'+1 \end{bmatrix};$$

$$\boldsymbol{V}_1^{(1)}\boldsymbol{h}_{2t'} = 2t', \quad \boldsymbol{V}_1^{(1)}\boldsymbol{h}_{2t'+1} = 2t'+1.$$

With ReLU activation, this construction leads to that

$$\boldsymbol{h}_{2t'} = 0, \quad \boldsymbol{h}_{2t'+1} = \pi^h(\cdot,\cdot|s) \cdot \overline{Q}^h(s,\cdot,\cdot) = \overline{V}^h(s).$$

Thus, with overall $2HS$ paralleling heads, the values of $\{\overline{V}^h(s), \underline{V}^h(s) : h \in [H], s \in \mathcal{S}\}$ can be computed.

The required transformer can be summarized as

$$L = 1, \quad M^{(1)} = O(HS), \quad \|\boldsymbol{\theta}\| = O(HS).$$

### D.2.8 Proof of Step 6: Obtain $\pi^{h+1}(\cdot,\cdot|s^{h+1})$

We can construct one $HS$-head transformer that for all $(s,h) \in \mathcal{S} \times [H]$

$$\boldsymbol{Q}_{h,s}^{(1)}\boldsymbol{h}_{2t'} = \begin{bmatrix} \boldsymbol{0} \\ \boldsymbol{e}_{h'} \\ t' \\ 1 \\ 1 \end{bmatrix}, \quad \boldsymbol{Q}_{h,s}^{(1)}\boldsymbol{h}_{2t'+1} = \begin{bmatrix} s^{t'+1} \\ \boldsymbol{e}_{h'+1} \\ t'+1 \\ 1 \\ 1 \end{bmatrix};$$

$$\boldsymbol{K}_{h,s}^{(1)}\boldsymbol{h}_{2t'} = \begin{bmatrix} \boldsymbol{e}_s \\ \boldsymbol{e}_h \\ -1 \\ t' \\ -1 \end{bmatrix}, \quad \boldsymbol{K}_{h,s}^{(1)}\boldsymbol{h}_{2t'+1} = \begin{bmatrix} \boldsymbol{e}_s \\ \boldsymbol{e}_h \\ -1 \\ t'+1 \\ -1 \end{bmatrix};$$

$$\boldsymbol{V}_{h,s}^{(1)}\boldsymbol{h}_{2t'} = \boldsymbol{0}, \quad \boldsymbol{V}_{h,s}^{(1)}\boldsymbol{h}_{2t'+1} = \pi^h(\cdot,\cdot|s).$$

With ReLU activation, this construction leads to the update that

$$\boldsymbol{h}_{2t'} = \boldsymbol{0}, \quad \boldsymbol{h}_{2t'+1} = \frac{1}{2t'+1} \cdot \pi^{h'+1}(\cdot,\cdot|s^{t'+1}).$$

Then, we can construct that

$$\boldsymbol{Q}_1^{(1)}\boldsymbol{h}_{2t'} = \begin{bmatrix} 2t' \\ 2GHt' \\ 1 \end{bmatrix}, \quad \boldsymbol{Q}_1^{(1)}\boldsymbol{h}_{2t'+1} = \begin{bmatrix} 2t'+1 \\ 2GH(t'+1) \\ 1 \end{bmatrix};$$

$$\boldsymbol{K}_1^{(1)}\boldsymbol{h}_{2t'} = \begin{bmatrix} 1 \\ -1 \\ 2GHt' \end{bmatrix}, \quad \boldsymbol{K}_1^{(1)}\boldsymbol{h}_{2t'+1} = \begin{bmatrix} 1 \\ -1 \\ 2GH(t'+1) \end{bmatrix};$$

$$\boldsymbol{V}_1^{(1)}\boldsymbol{h}_{2t'} = \boldsymbol{0}, \quad \boldsymbol{V}_{h,s}^{(1)}\boldsymbol{h}_{2t'+1} = \frac{1}{2t'+1} \cdot \pi^{h'+1}(\cdot,\cdot|s^{t'+1}).$$

With ReLU activation, this construction leads to the update that

$$\boldsymbol{h}_{2t'} = \boldsymbol{0}, \quad \boldsymbol{h}_{2t'+1} = \pi^{h'+1}(\cdot,\cdot|s^{t'+1}).$$

The required transformer can be summarized as

$$L = 2, \quad \max_{l \in [L]} M^{(l)} = O(HS), \quad \|\boldsymbol{\theta}\| = O(HS + GH).$$

# E  Proofs for the Centralized Overall Performance

*Proof of Theorem 4.2.* First, we can obtain the decomposition that

$$
\mathbb{E}_{M\sim\Lambda,D\sim\mathbb{P}_M^{\mathtt{Alg}_{\widehat{\theta}}}}\left[\sum_{g\in[G]} V_M^{\dagger,\nu^g}(s^1) - V_M^{\mu^g,\dagger}(s^1)\right]
$$

$$
= \mathbb{E}_{M\sim\Lambda,D\sim\mathbb{P}_M^{\mathtt{Alg}_0}}\left[\sum_{g\in[G]} V_M^{\dagger,\nu^g}(s^1) - V_M^{\mu^g,\dagger}(s^1)\right]
$$

$$
+ \mathbb{E}_{M\sim\Lambda,D\sim\mathbb{P}_M^{\mathtt{Alg}_{\widehat{\theta}}}}\left[\sum_{g\in[G]} V_M^{\dagger,\nu^g}(s^1)\right] - \mathbb{E}_{M\sim\Lambda,D\sim\mathbb{P}_M^{\mathtt{Alg}_0}}\left[\sum_{g\in[G]} V_M^{\dagger,\nu^g}(s^1)\right]
$$

$$
+ \mathbb{E}_{M\sim\Lambda,D\sim\mathbb{P}_M^{\mathtt{Alg}_0}}\left[\sum_{g\in[G]} V_M^{\mu^g,\dagger}(s^1)\right] - \mathbb{E}_{M\sim\Lambda,D\sim\mathbb{P}_M^{\mathtt{Alg}_{\widehat{\theta}}}}\left[\sum_{g\in[G]} V_M^{\mu^g,\dagger}(s^1)\right].
$$

Via Theorem D.1, it holds that

$$
\mathbb{E}_{M\sim\Lambda,D\sim\mathbb{P}_M^{\mathtt{Alg}_0}}\left[\sum_{g\in[G]} V_M^{\dagger,\nu^g}(s^1) - V_M^{\mu^g,\dagger}(s^1)\right] = O\left(\sqrt{H^3 S^2 ABT \log(SABT)}\right).
$$

Then, via Lemma E.1 and Theorem C.3, we can obtain that

$$
\mathbb{E}_{M\sim\Lambda,D\sim\mathbb{P}_M^{\mathtt{Alg}_0}}\left[\sum_{g\in[G]} V_M^{\mu^g,\dagger}(s^1)\right] - \mathbb{E}_{M\sim\Lambda,D\sim\mathbb{P}_M^{\mathtt{Alg}_{\widehat{\theta}}}}\left[\sum_{g\in[G]} V_M^{\mu^g,\dagger}(s^1)\right]
$$

$$
= O\left(T\cdot\mathbb{E}_{M\sim\Lambda,D\sim\mathbb{P}_M^{\mathtt{Alg}_0}}\left[\sum_{t\in[T]}\sum_{s\in\mathcal{S}}\left[\mathtt{TV}\left(\mathtt{Alg}_0,\mathtt{Alg}_{\widehat{\theta}}|D^{t-1},s\right)\right]\right]\right)
$$

$$
= O\left(T^2 S\sqrt{\varepsilon_{\mathrm{real}}} + T^2 S\sqrt{\frac{\log\left(\mathcal{N}_\Theta TS/\delta\right)}{N}}\right),
$$

and similarly,

$$
\mathbb{E}_{M\sim\Lambda,D\sim\mathbb{P}_M^{\mathtt{Alg}_{\widehat{\theta}}}}\left[\sum_{g\in[G]} V_M^{\dagger,\nu^g}(s^1)\right] - \mathbb{E}_{M\sim\Lambda,D\sim\mathbb{P}_M^{\mathtt{Alg}_0}}\left[\sum_{g\in[G]} V_M^{\dagger,\nu^g}(s^1)\right]
$$

$$
= O\left(T^2 S\sqrt{\varepsilon_{\mathrm{real}}} + T^2 S\sqrt{\frac{\log\left(\mathcal{N}_\Theta TS/\delta\right)}{N}}\right).
$$

With Theorem 4.1 providing that $\varepsilon_{\mathrm{real}} = 0$, combining the above terms completes the proof of the regret bound. The bound on the covering number, i.e., $\log(\mathcal{N}_\Theta)$ can be obtained via Lemma I.4. □

**Lemma E.1.** *For any two centralized algorithms $\mathtt{Alg}_\alpha$ and $\mathtt{Alg}_\beta$, we denote their performed policies for episode g are $(\pi_\alpha^g, \pi_\beta^g)$, whose marginal policies are $(\mu_\alpha^g, \nu_\alpha^g)$ and $(\mu_\beta^g, \nu_\beta^g)$. For $\{\mu_\alpha^g, \nu_\beta^g\}$, it holds that*

$$
\mathbb{E}_\alpha\left[\sum_{g\in[G]} V_M^{\mu_\alpha^g,\dagger}(s^1)\right] - \mathbb{E}_\beta\left[\sum_{g\in[G]} V_M^{\mu_\beta^g,\dagger}(s^1)\right] \lesssim T\cdot\mathbb{E}_\alpha\left[\sum_{t\in[T],s\in\mathcal{S}}\mathtt{TV}\left(\pi_\alpha^t,\pi_\beta^t|D^{t-1},s\right)\right],
$$

*where $\mathbb{E}_\alpha[\cdot]$ and $\mathbb{E}_\beta[\cdot]$ are with respect to $\mathbb{P}_\Lambda^{\mathtt{Alg}_\alpha}$ and $\mathbb{P}_\Lambda^{\mathtt{Alg}_\beta}$. A similar result holds for $\{\nu_\alpha^g, \nu_\beta^g\}$.*

*Proof of Lemma E.1.* It holds that

$$
\mathbb{E}_{D\sim\mathbb{P}_M^{\mathtt{Alg}_\alpha}}\left[\sum_{g\in[G]} V_M^{\mu_\alpha^g,\dagger}(s^1)\right] - \mathbb{E}_{D\sim\mathbb{P}_M^{\mathtt{Alg}_\beta}}\left[\sum_{g\in[G]} V_M^{\mu_\beta^g,\dagger}(s^1)\right]
$$

$$= \mathbb{E}_{D \sim \mathbb{P}_M^{\mathtt{Alg}_\alpha}} \underbrace{\left[ \sum_{g \in [G]} V_M^{\mu_\alpha^g, \dagger}(s^1) - \sum_{g \in [G]} V_M^{\mu_\beta^g, \dagger}(s^1) \right]}_{:=(\text{term I})}$$

$$+ \underbrace{\mathbb{E}_{D \sim \mathbb{P}_M^{\mathtt{Alg}_\alpha}} \left[ \sum_{g \in [G]} V_M^{\mu_\beta^g, \dagger}(s^1) \right] - \mathbb{E}_{D \sim \mathbb{P}_M^{\mathtt{Alg}_\beta}} \left[ \sum_{g \in [G]} V_M^{\mu_\beta^g, \dagger}(s^1) \right]}_{:=(\text{term II})}$$

Denoting $D^H := D^{(g-1)H+1:gH}$ and $f(D^H) = \sum_{h \in [H]} r^{g,h}$, we can further obtain that

$$V_M^{\mu_\alpha^g, \dagger}(s^1) - V_M^{\mu_\beta^g, \dagger}(s^1)$$

$$\overset{(a)}{\leq} V_M^{\mu_\alpha^g, \nu_\dagger(\mu_\beta^g)}(s^1) - V_M^{\mu_\beta^g, \nu_\dagger(\mu_\beta^g)}(s^1)$$

$$= \mathbb{E}_{D^H \sim \mathbb{P}_M^{\mu_\alpha^g, \nu_\dagger(\mu_\beta^g)}} \left[ f(D^H) \right] - \mathbb{E}_{D^H \sim \mathbb{P}_M^{\mu_\beta^g, \nu_\dagger(\mu_\beta^g)}} \left[ f(D^H) \right]$$

$$= \sum_{h \in [H]} \mathbb{E}_{D^{1:h} \sim \mathbb{P}_M^{\mu_\alpha^g, \nu_\dagger(\mu_\beta^g)}, D^{h+1:H} \sim \mathbb{P}_M^{\mu_\beta^g, \nu_\dagger(\mu_\beta^g)}} \left[ f(D^H) \right]$$

$$- \sum_{h \in [H]} \mathbb{E}_{D^{1:h-1} \sim \mathbb{P}_M^{\mu_\alpha^g, \nu_\dagger(\mu_\beta^g)}, D^{h:H} \sim \mathbb{P}_M^{\mu_\beta^g, \nu_\dagger(\mu_\beta^g)}} \left[ f(D^H) \right]$$

$$\overset{(b)}{\leq} 2H \sum_{h \in [H]} \mathbb{E}_{D^{1:h-1} \sim \mathbb{P}_M^{\mu_\alpha^g, \nu_\dagger(\mu_\beta^g)}, s^{g,h}} \left[ \mathrm{TV}\left( \mu_\alpha^{g,h} \times \nu_\dagger^h(\mu_\beta^g)(\cdot, \cdot | s^{g,h}), \mu_\beta^{g,h} \times \nu_\dagger^h(\mu_\beta^g)(\cdot, \cdot | s^{g,h}) \right) \right]$$

$$\overset{(c)}{=} 2H \sum_{h \in [H]} \mathbb{E}_{D^{1:h-1} \sim \mathbb{P}_M^{\mu_\alpha^g, \nu_\dagger(\mu_\beta^g)}, s^{g,h}} \left[ \mathrm{TV}\left( \mu_\alpha^{g,h}(\cdot | s^{g,h}), \mu_\beta^{g,h}(\cdot | s^{g,h}) \right) \right]$$

$$\overset{(d)}{\leq} 2H \sum_{h \in [H]} \sum_{s \in \mathcal{S}} \mathrm{TV}\left( \pi_\alpha^{g,h}(\cdot, \cdot | s), \pi_\beta^{g,h}(\cdot, \cdot | s) \right),$$

where (a) is from the definition of best responses, (b) is from the variational representation of the TV distance, (c) is from the fact that $\mathrm{TV}(\mu \times \nu, \mu' \times \nu) = \mathrm{TV}(\mu, \mu')$, and (d) is from the fact that $\mathrm{TV}(\sum_b \pi(\cdot, b), \sum_b \pi'(\cdot, b)) \leq \mathrm{TV}(\pi(\cdot, \cdot), \pi'(\cdot, \cdot))$. The above relationship further leads to that

$$(\text{term I}) := \mathbb{E}_{D \sim \mathbb{P}_M^{\mathtt{Alg}_\alpha}} \left[ \sum_{g \in [G]} V_M^{\mu_\alpha^g, \dagger}(s^1) - \sum_{g \in [G]} V_M^{\mu_\beta^g, \dagger}(s^1) \right]$$

$$\leq 2H \cdot \mathbb{E}_{D \sim \mathbb{P}_M^{\mathtt{Alg}_\alpha}} \left[ \sum_{t \in [T]} \sum_{s \in \mathcal{S}} \mathrm{TV}\left( \pi_\alpha^t, \pi_\beta^t | D^{t-1}, s \right) \right].$$

Also, denoting $g(D) = \sum_{g \in [G]} V^{\mu_\beta^g, \dagger}(s^1)$, it holds that

$$(\text{term II}) := \mathbb{E}_{D \sim \mathbb{P}_M^{\mathtt{Alg}_\alpha}} \left[ \sum_{g \in [G]} V^{\mu_\beta^g, \dagger}(s^1) \right] - \mathbb{E}_{D \sim \mathbb{P}_M^{\mathtt{Alg}_\beta}} \left[ \sum_{g \in [G]} V^{\mu_\beta^g, \dagger}(s^1) \right]$$

$$= \sum_{t \in [T]} \mathbb{E}_{D^t \sim \mathbb{P}_M^{\mathtt{Alg}_\alpha}, D^{t+1:T} \sim \mathbb{P}_M^{\mathtt{Alg}_\beta}} [g(D)] - \sum_{t \in [T]} \mathbb{E}_{D^{t-1} \sim \mathbb{P}_M^{\mathtt{Alg}_\alpha}, D^{t:T} \sim \mathbb{P}_M^{\mathtt{Alg}_\beta}} [g(D)]$$

$$\leq 2T \sum_{t \in [T]} \mathbb{E}_{D^{t-1} \sim \mathbb{P}_M^{\mathtt{Alg}_\alpha}, s^t} \left[ \mathrm{TV}\left( \mathtt{Alg}_\alpha, \mathtt{Alg}_\beta | D^{t-1}, s^t \right) \right]$$

$$\leq 2T \cdot \mathbb{E}_{D \sim \mathbb{P}_M^{\mathtt{Alg}_\alpha}} \left[ \sum_{t \in [T]} \sum_{s \in \mathcal{S}} \mathrm{TV}\left( \pi_\alpha^t, \pi_\beta^t | D^{t-1}, s \right) \right].$$

Combining (term I) and (term II) finishes the proof. $\qquad \square$

# F   Proofs for the Decentralized Supervised Pre-training Guarantees

**Definition F.1** (The Complete Version of Definition 3.1). *For a class of algorithms $\{\mathtt{Alg}_{\boldsymbol{\theta}_+} : \boldsymbol{\theta}_+ \in \Theta_+\}$, we say $\tilde{\Theta}_+ \subseteq \Theta_+$ is a $\rho_+$-cover of $\Theta_+$, if $\tilde{\Theta}_+$ is a finite set such that for any $\boldsymbol{\theta}_+ \in \Theta_+$, there exists $\tilde{\boldsymbol{\theta}}_+ \in \tilde{\Theta}_+$ such that for all $D_+^{t-1}, s \in \mathcal{S}, t \in [T]$, it holds that*

$$\left\| \log \mathtt{Alg}_{\tilde{\boldsymbol{\theta}}_+}(\cdot, \cdot | D_+^{t-1}, s) - \log \mathtt{Alg}_{\boldsymbol{\theta}_+}(\cdot | D_+^{t-1}, s) \right\|_\infty \le \rho_+.$$

*The covering number $\mathcal{N}_{\Theta_+}(\rho_+)$ is the minimal cardinality of $\tilde{\Theta}_+$ such that $\tilde{\Theta}_+$ is a $\rho_+$-cover of $\Theta_+$.*

*Similarly, for a class of algorithms $\{\mathtt{Alg}_{\boldsymbol{\theta}_-} : \boldsymbol{\theta}_- \in \Theta_-\}$, we say $\tilde{\Theta}_- \subseteq \Theta_-$ is a $\rho_-$-cover of $\Theta_-$, if $\tilde{\Theta}_-$ is a finite set such that for any $\boldsymbol{\theta}_- \in \Theta_-$, there exists $\tilde{\boldsymbol{\theta}}_- \in \tilde{\Theta}_-$ such that for all $D_-^{t-1}, s \in \mathcal{S}, t \in [T]$, it holds that*

$$\left\| \log \mathtt{Alg}_{\tilde{\boldsymbol{\theta}}_-}(\cdot, \cdot | D_-^{t-1}, s) - \log \mathtt{Alg}_{\boldsymbol{\theta}_-}(\cdot | D_-^{t-1}, s) \right\|_\infty \le \rho_-.$$

*The covering number $\mathcal{N}_{\Theta_-}(\rho_-)$ is the minimal cardinality of $\tilde{\Theta}_-$ such that $\tilde{\Theta}_-$ is a $\rho_-$-cover of $\Theta_-$.*

**Assumption F.2** (The Complete Version of Assumption 3.2). *There exists $\boldsymbol{\theta}_+^* \in \Theta_+$ such that there exists $\varepsilon_{+,\mathrm{real}} > 0$, for all $t \in [T], s \in \mathcal{S}, a \in \mathcal{A}$, it holds that*

$$\log \left( \mathbb{E}_{M \sim \Lambda, D \sim \mathbb{P}_M^{\mathtt{Alg0}}} \left[ \frac{\mathtt{Alg}_{+,0}(a | D_+^{t-1}, s)}{\mathtt{Alg}_{\boldsymbol{\theta}_+^*}(a | D_+^{t-1}, s)} \right] \right) \le \varepsilon_{+,\mathrm{real}}.$$

*Similarly, there exists $\boldsymbol{\theta}_-^* \in \Theta_-$ such that there exists $\varepsilon_{-,\mathrm{real}} > 0$, for all $t \in [T], s \in \mathcal{S}, b \in \mathcal{B}$, it holds that*

$$\log \left( \mathbb{E}_{M \sim \Lambda, D \sim \mathbb{P}_M^{\mathtt{Alg0}}} \left[ \frac{\mathtt{Alg}_{-,0}(b | D_-^{t-1}, s)}{\mathtt{Alg}_{\boldsymbol{\theta}_-^*}(b | D_-^{t-1}, s)} \right] \right) \le \varepsilon_{-,\mathrm{real}}.$$

**Theorem F.3** (The Complete Version of Theorem 3.3). *Let $\widehat{\boldsymbol{\theta}}_+$ be the max-player's pre-training output defined in Section 3.1.1. Take $\mathcal{N}_{\Theta_+} = \mathcal{N}_{\Theta_+}(1/N)$ as in Definition F.1. Then, under Assumption F.2, with probability at least $1 - \delta$, it holds that*

$$\mathbb{E}_{M \sim \Lambda, D \sim \mathbb{P}_M^{\mathtt{Alg0}}} \left[ \sum_{t \in [T], s \in \mathcal{S}} \mathrm{TV} \left( \mathtt{Alg}_{+,0}, \mathtt{Alg}_{\widehat{\boldsymbol{\theta}}_+} | D_+^{t-1}, s \right) \right]$$
$$= O \left( TS \sqrt{\varepsilon_{+,\mathrm{real}}} + TS \sqrt{\frac{\log \left( \mathcal{N}_{\Theta_+} TS/\delta \right)}{N}} \right).$$

*Let $\widehat{\boldsymbol{\theta}}_-$ be the min-player's pre-training output defined in Section 3.1.1. Take $\mathcal{N}_{\Theta_-} = \mathcal{N}_{\Theta_-}(1/N)$ as in Definition F.1. Then, under Assumption F.2, with probability at least $1 - \delta$, it holds that*

$$\mathbb{E}_{M \sim \Lambda, D \sim \mathbb{P}_M^{\mathtt{Alg0}}} \left[ \sum_{t \in [T], s \in \mathcal{S}} \mathrm{TV} \left( \mathtt{Alg}_{-,0}, \mathtt{Alg}_{\widehat{\boldsymbol{\theta}}_-} | D_+^{t-1}, s \right) \right]$$
$$= O \left( TS \sqrt{\varepsilon_{-,\mathrm{real}}} + TS \sqrt{\frac{\log \left( \mathcal{N}_{\Theta_-} TS/\delta \right)}{N}} \right).$$

*Proof of Theorem F.3.* This theorem can be similarly proved as Theorem 3.3. $\qquad\square$

# G   Proofs for Realizing V-learning

In the following proof, we focus on the max-player's perspective, which can be easily extended for the min-player.

## G.1 Details of V-learning

The details of V-learning [32], discussed in Sec. 3.2, are presented in Alg. 3, where the following notations are adopted

$$\alpha_n = \frac{H+1}{H+n}, \quad \beta_n = c \cdot \sqrt{\frac{H^3 A \log(HSAG/\delta)}{n}},$$

$$\gamma_n = \eta_n = \sqrt{\frac{H \log(A)}{An}}, \quad \omega_n = \alpha_n \left( \prod_{\tau=2}^{n} (1 - \alpha_\tau) \right)^{-1}.$$

After the learning process, V-learning requires an additional procedure to provide the output policy. We include this procedure in Alg. 4, where the following notations are adopted

$$N^{g,h}(s) = \text{the number of times } s \text{ is visited at step } h \text{ before episode } g,$$

$$g_i^h(s) = \text{the index of the episode } s \text{ is visited at step } h \text{ for the } i\text{-th time,}$$

$$\alpha_{n,i} = \alpha_i \prod_{j=i+1}^{n} (1 - \alpha_j).$$

Following the results in Jin et al. [32], we can obtain the following performance guarantee of V-learning.

**Theorem G.1** (Theorem 4 in Jin et al. [32]). *With probability at least $1 - \delta$, in any environment $M$, the output policies $(\hat{\mu}, \hat{\nu})$ from V-learning satisfy that*

$$V_M^{\dagger,\hat{\nu}}(s^1) - V_M^{\hat{\mu},\dagger}(s^1) = O\left( \sqrt{\frac{H^5 S(A+B) \log(SABT/\delta)}{G}} \right).$$

Thus, with $\delta = 1/T$, we can obtain that

$$\mathbb{E}_{D \sim \mathbb{P}_M^{\text{V-learning}}} \left[ V_M^{\dagger,\hat{\nu}}(s^1) - V_M^{\hat{\mu},\dagger}(s^1) \right] = O\left( \sqrt{\frac{H^5 S(A+B) \log(SABT)}{G}} \right).$$

---

**Algorithm 3** V-learning [32]

---

1: **Initialize:** for any $(s, a, h) \in \mathcal{S} \times \mathcal{A} \times [H]$, $V^h(s) \leftarrow H + 1 - h$, $N^h(s) \leftarrow 0$, $\mu^h(a|s) \leftarrow 1/A$
2: **for** episode $g = 1, \cdots, G$ **do**
3:     receive $s^{g,1}$
4:     **for** step $h = 1, \cdots, H$ **do**
5:         Take action $a^{g,h} \sim \pi^h(\cdot|s^{g,h})$, observe reward $r^{g,h}$ and next state $s^{g,h+1}$
6:         Update $n = N^h(s^{g,h}) \leftarrow N^h(s^{g,h}) + 1$
7:         Update $\tilde{V}^h(s^{g,h}) \leftarrow (1 - \alpha_n) \tilde{V}^h(s^{g,h}) + \alpha_n \left( r^{g,h} + V^{h+1}(s^{g,h+1}) + \beta_n \right)$
8:         Update $V^h(s^{g,h}) \leftarrow \min \left\{ H + 1 - h, \tilde{V}^h(s^{g,h}) \right\}$
9:         Compute $\tilde{\ell}_n^h(s^{g,h}, a) \leftarrow \frac{H - r^{g,h} - V^{h+1}(s^{g,h+1})}{H} \cdot \frac{\mathbf{1}\{a = a^{g,h}\}}{\mu^h(a^{g,h}|s^{g,h}) + \gamma_n}$ for all $a \in \mathcal{A}$
10:        Update $\mu^h(a|s^{g,h}) \propto \exp \left( -\frac{\eta_n}{\omega_n} \cdot \sum_{\tau \in [n]} \omega_\tau \cdot \tilde{\ell}_\tau^h(s^{g,h}, a) \right)$ for all $a \in \mathcal{A}$
11:     **end for**
12: **end for**

---

## G.2 An Additional Assumption About Transformers Performing Division

Before digging into the proof of Theorem 3.4, we first state the following assumption that there exists one transformer that can perform the division operation.

**Assumption G.2.** *There exists one transformer* $\text{TF}_{\boldsymbol{\theta}}$ *with*

$$L = L_D, \quad \max_{l \in [L]} M^{(l)} = M_D, \quad d' = d_D, \quad \|\boldsymbol{\theta}\| = F_D,$$

**Algorithm 4** V-learning Executing Output Policy $\hat{\mu}$ [32]

1: Sample $g \sim \text{Unif}(1, 2, \cdots, G)$
2: **for** step $h = 1, \cdots, H$ **do**
3:   Observe $s^h$ and set $n \leftarrow N^{g,h}(s^h)$
4:   Set $g \leftarrow g_i^h(s^h)$ where $i \in [n]$ is sampled with probability $\alpha_{n,i}$
5:   Take action $a^h \sim \mu^{g,h}(\cdot|s^h)$
6: **end for**

---

such that for any $x \in [0, 1], y \in [\sqrt{\log(A)/(AG)}, \sqrt{\log(A)/A} + 1], \varphi > 0$, with input $\boldsymbol{H} = [\boldsymbol{h}_1, \cdots, \boldsymbol{h}_e, \cdots]$, where

$$\boldsymbol{h}_i = \begin{bmatrix} 0 \\ \varphi \\ \boldsymbol{0} \end{bmatrix}, \ \ \forall i \neq e; \ \ \boldsymbol{h}_e = \begin{bmatrix} x \\ y \\ \boldsymbol{0} \end{bmatrix},$$

the transformer can provide output $\overline{\boldsymbol{H}} = [\overline{\boldsymbol{h}}_1, \cdots, \overline{\boldsymbol{h}}_e, \cdots]$

$$\overline{\boldsymbol{h}}_i = \begin{bmatrix} 0 \\ \varphi \\ 0 \\ \boldsymbol{0} \end{bmatrix}, \ \ \forall i \neq e; \ \ \overline{\boldsymbol{h}}_e = \begin{bmatrix} x \\ y \\ x/y \\ \boldsymbol{0} \end{bmatrix}.$$

We note that this assumption on performing exact division is only for the convenience of the proof as one can approximate the division to any arbitrary precision only via one MLP layer with ReLU activation.

In particular, let $\text{Ball}_\infty^k(R) = [-R, R]^\infty$ denote the standard $\ell_\infty$ ball in $\mathbb{R}^k$ with radius $R > 0$, we introduce the following definitions and result.

**Definition G.3** (Approximability by Sum of ReLUs, Definition 12 in Bai et al. [7]). *A function $g : \mathbb{R}^k \to \mathbb{R}$ is $(\varepsilon_{\text{approx}}, R, M, C)$-approximable by sum of ReLUs, if there exists a "$(M, C)$-sum of ReLUs" function*

$$f_{M,C}(\boldsymbol{z}) = \sum_{m=1}^{M} c_m \sigma(\boldsymbol{a}_m^\top [\boldsymbol{z}; 1])$$

*with*

$$\sum_{m=1}^{M} |c_m| \leq C, \ \ \max_{m \in [M]} \|\boldsymbol{a}_m\|_1 \leq 1, \ \ \boldsymbol{a}_m \in \mathbb{R}^{k+1}, \ \ c_m \in \mathbb{R}$$

*such that*

$$\sup_{\boldsymbol{z} \in \text{Ball}_\infty^k(R)} |g(\boldsymbol{z}) - f_{M,C}(\boldsymbol{z})| \leq \varepsilon_{\text{approx}}.$$

**Definition G.4** (Sufficiently Smooth $k$-variable Function, Definition A.1 in Bai et al. [7]). *We say a function $g : \mathbb{R}^k \to \mathbb{R}$ is $(R, C_\ell)$-smooth if for $s = \lceil (k-1)/2 \rceil + 2$, $g$ is a $C^s$ function on $\text{Ball}_\infty^k(R)$, and*

$$\sup_{\boldsymbol{z} \in \text{Ball}_\infty^k(R)} \|\nabla^i g(\boldsymbol{z})\|_\infty = \sup_{\boldsymbol{z} \in \text{Ball}_\infty^k(R)} \max_{j_1, \cdots, j_i \in [k]} |\partial_{z_{j_1} \cdots z_{j_i}} g(\boldsymbol{z})| \leq L_i$$

*for all $i \in \{0, 1, \cdots, s\}$, with*

$$\max_{0 \leq i \leq s} L_i R^i \leq C_\ell.$$

**Proposition G.5** (Approximating Smooth $k$-variable Functions, Proposition A.1 in Bai et al. [7]). *For any $\varepsilon_{\text{approx}} > 0, R \geq 1, C_\ell > 0$, we have the following: Any $(R, C_\ell)$-smooth function $g : \mathbb{R}^k \to \mathbb{R}$ is $(\varepsilon_{\text{approx}}, R, M, C)$ approximable by sum of ReLUs with $M \leq C(k)C_\ell^2 \log(1 + C_\ell/\varepsilon_{approx})/\varepsilon_{approx}^2$ and $C \leq C(k)C_\ell$, where $C(k) > 0$ is a constant that depends only on $k$.*

Then, we can consider $g(x, y) = (c_1 + x)/(c_2 + y)$ with $c_1 \geq 1$, $c_2 \geq 1 + c_2' > 1$, $x \in [-1, 1]$ and $y \in [-1, 1]$. It can be verified that

$$|g(x,y)| = \frac{c_1 + x}{c_2 + y} \leq \frac{1 + c_1}{c_2'}; \quad |\partial_x g(x,y)| = \frac{1}{c + y} \leq \frac{1}{c_2'}; \quad |\partial_y g(x,y)| = \frac{c_1 + x}{(c_2 + y)^2} \leq \frac{1 + c_1}{(c_2')^2};$$

$$|\partial_x^2 g(x,y)| = 0; \quad |\partial_y^2 g(x,y)| = \frac{2(c_1 + x)}{(c_2 + y)^3} \leq \frac{2(1 + c_1)}{(c_2')^3}; \quad |\partial_x \partial_y g(x,y)| = \frac{1}{(c + y)^2} \leq \frac{1}{(c_2')^2};$$

$$|\partial_x^3 g(x,y)| = 0; \quad |\partial_y^3 g(x,y)| = \frac{6(c_1 + x)}{(c_2 + y)^4} \leq \frac{6(1 + c_1)}{(c_2')^4};$$

$$|\partial_x^2 \partial_y g(x,y)| = 0; \quad |\partial_x \partial_y^2 g(x,y)| = \frac{2}{(c + y)^3} \leq \frac{2}{(c_2')^3}.$$

Then, from Definition G.4 there exists one $C_l$ such that $g(x, y)$ is $(1, C_l)$-smooth. Thus, by Proposition G.5, this function can be approximated by a sum of ReLUs (defined in Definition G.3). With this observation, the division operation required in Assumption G.2 can be approximated.

### G.3 Proof of Theorem 3.4: The Realization Construction

#### G.3.1 Embeddings and Extraction Mappings

We consider each episode of the max-player's observations to be embedded in $2H$ tokens. In particular, for each $t \in [T]$, we construct that

$$\boldsymbol{h}_{2t-1} = \mathtt{h}_+(s^{g,h}) = \begin{bmatrix} \mathbf{0}_A \\ 0 \\ \hline s^{g,h} \\ \hline \mathbf{0}_A \\ 0 \\ \hline \mathbf{pos}_{2t-1} \end{bmatrix} =: \begin{bmatrix} \boldsymbol{h}_{2t-1}^{\mathrm{pre},a} \\ \boldsymbol{h}_{2t-1}^{\mathrm{pre},b} \\ \boldsymbol{h}_{2t-1}^{\mathrm{pre},c} \\ \boldsymbol{h}_{2t-1}^{\mathrm{pre},d} \end{bmatrix},$$

$$\boldsymbol{h}_{2t} = \mathtt{h}_+(a^{g,h}, r^{g,h}) = \begin{bmatrix} a^{g,h} \\ r^{g,h} \\ \hline \mathbf{0}_S \\ \mathbf{0}_A \\ 0 \\ \hline \mathbf{pos}_{2t} \end{bmatrix} =: \begin{bmatrix} \boldsymbol{h}_{2t}^{\mathrm{pre},a} \\ \boldsymbol{h}_{2t}^{\mathrm{pre},b} \\ \boldsymbol{h}_{2t}^{\mathrm{pre},c} \\ \boldsymbol{h}_{2t}^{\mathrm{pre},d} \end{bmatrix},$$

where $s^{g,h}, a^{g,h}$ are represented via one-hot embedding. The positional embedding $\mathbf{pos}_i$ is defined as

$$\mathbf{pos}_i := \begin{bmatrix} g \\ h \\ t \\ \boldsymbol{e}_h \\ v_i \\ i \\ i^2 \\ 1 \end{bmatrix},$$

where $v_i := \mathbb{1}\{\boldsymbol{h}_i^a = \mathbf{0}\}$ denote the tokens that do not embed actions and rewards.

In summary, for observations $D_+^{t-1} \cup \{s^t\}$, we obtain the tokens of length $2t - 1$ which can be expressed as the following form:

$$\boldsymbol{H} := h_+(D_+^{t-1}, s^t) = [\boldsymbol{h}_1, \boldsymbol{h}_2, \cdots, \boldsymbol{h}_{2t-1}] = [\mathtt{h}_+(s^1), \mathtt{h}_+(a^1, r^1), \cdots, \mathtt{h}_+(s^t)].$$

With the above input $\boldsymbol{H}$, the transformer outputs $\overline{\boldsymbol{H}} = \mathrm{TF}_{\boldsymbol{\theta}_+}(\boldsymbol{H})$ of the same size as $\boldsymbol{H}$. The extraction mapping $\mathtt{A}$ is directly set to satisfy the following

$$\mathtt{A} \cdot \overline{\boldsymbol{h}}_{-1} = \mathtt{A} \cdot \overline{\boldsymbol{h}}_{2t-1} = \overline{\boldsymbol{h}}_{2t-1}^c \in \mathbb{R}^A,$$

i.e., the part $c$ of the output tokens is used to store the learned policy.

### G.3.2 An Overview of the Proof

In the following, for the convenience of notations, we will consider step $t + 1$, i.e., with observations $D_+^t \cup \{s^{t+1}\}$. Given an input token matrix

$$\boldsymbol{H} = h_+(D_+^t, s^{t+1}) = [\boldsymbol{h}_1, \boldsymbol{h}_2, \cdots, \boldsymbol{h}_{2t+1}],$$

we construct a transformer to perform the following steps

$$
\begin{bmatrix} \boldsymbol{h}_{2t+1}^{\mathrm{pre},a} \\ \boldsymbol{h}_{2t+1}^{\mathrm{pre},b} \\ \boldsymbol{h}_{2t+1}^{\mathrm{pre},c} \\ \boldsymbol{h}_{2t+1}^{\mathrm{pre},d} \end{bmatrix}
\xrightarrow{\text{step 1}}
\begin{bmatrix} \boldsymbol{h}_{2t+1}^{\mathrm{pre},\{a,b,c\}} \\ \tilde{N}^t(s^t) \\ \star \\ \boldsymbol{0} \\ \mathbf{pos}_{2t+1} \end{bmatrix}
\xrightarrow{\text{step 2}}
\begin{bmatrix} \boldsymbol{h}_{2t+1}^{\mathrm{pre},\{a,b,c\}} \\ \tilde{V}^t(s^t) \\ V^t(s^t) \\ \star \\ \boldsymbol{0} \\ \mathbf{pos}_{2t+1} \end{bmatrix}
\xrightarrow{\text{step 3}}
\begin{bmatrix} \boldsymbol{h}_{2t+1}^{\mathrm{pre},\{a,b,c\}} \\ \ell_n(s^t, \cdot) \\ \mu^t(\cdot | s^t) \\ \star \\ \boldsymbol{0} \\ \mathbf{pos}_{2t+1} \end{bmatrix}
$$

$$
\xrightarrow{\text{step 4}}
\begin{bmatrix} \boldsymbol{h}_{2t+1}^{\mathrm{pre},\{a,b\}} \\ \mu^{t+1}(\cdot | s^{t+1}) \\ \boldsymbol{h}_{2t+1}^d \end{bmatrix}
:=
\begin{bmatrix} \boldsymbol{h}_{2t+1}^{\mathrm{post},a} \\ \boldsymbol{h}_{2t+1}^{\mathrm{post},b} \\ \boldsymbol{h}_{2t+1}^{\mathrm{post},c} \\ \boldsymbol{h}_{2t+1}^{\mathrm{post},d} \end{bmatrix},
$$

where $n := N^h(s^{g,h})$.

To ease the notations, in the proof, we will slightly abuse $\mathrm{TF}_{\boldsymbol{\theta}}$ as $\mathrm{TF}_{\boldsymbol{\theta}_+}$. The following provides a sketch of the proof.

**Step 1** There exists an attention-only transformer $\mathrm{TF}_{\boldsymbol{\theta}}$ to complete Step 1 with

$$L = O(1), \quad \max_{l \in [L]} M^{(l)} = O(1), \quad \|\boldsymbol{\theta}\| = O(HG)$$

**Step 2** There exists a transformer $\mathrm{TF}_{\boldsymbol{\theta}}$ to complete Step 2 with

$$L = O(HG), \quad \max_{l \in [L]} M^{(l)} = O(HS^2), \quad d' = O(G), \quad \|\boldsymbol{\theta}\| = O(G^3 + GH),$$

**Step 3** There exists a transformer $\mathrm{TF}_{\boldsymbol{\theta}}$ to complete Step 3 with

$$L = O(GHL_D), \quad \max_{l \in [L]} M^{(l)} = O(HSA + M_D), \quad d' = O(d_D + A + G),$$
$$\|\boldsymbol{\theta}\| = O(GH^2S + F_D + G^3),$$

**Step 4** There exists a transformer $\mathrm{TF}_{\boldsymbol{\theta}}$ to complete Step 4 with

$$L = O(1), \quad \max_{l \in [L]} M^{(l)} = O(HS), \quad \|\boldsymbol{\theta}\| = O(HS + GH).$$

Thus, the overall transformer $\mathrm{TF}_{\boldsymbol{\theta}}$ can be summarized as

$$L = O(GHL_D), \quad \max_{l \in [L]} M^{(l)} = O(HS^2 + HSA + M_D), \quad d' = O(G + A + d_D),$$
$$\|\boldsymbol{\theta}\| = O(GH^2S + G^3 + F_D).$$

Also, from the later construction, we can observe that $\log(R) = \tilde{\mathcal{O}}(1)$. The bound on the covering number, i.e., $\log(N_{\Theta_+})$ can be obtained via Lemma I.4.

### G.3.3 Proof of Step 1: Update $N^h(s^{g,h})$.

From the proof of Step 1 in realizing UCB-VI in Lin et al. [38], we know that there exists a transformer with 3 heads that for all $t' \le t$, can move $s^{t'}, (a^{t'}, r^{t'})$ from $\boldsymbol{h}_{2t'-1}^a$ and $\boldsymbol{h}_{2t'}^b$ to $\boldsymbol{h}_{2t'+1}^d$ while maintaining $\boldsymbol{h}_{2t'}^d$ not updated. This constructed transformer is shown in Lin et al. [38] to be an attention-only one with ReLU activation and

$$L = 2, \quad \max_{l \in [L]} M^{(l)} \le 3, \quad \|\boldsymbol{\theta}\| \le O(HG).$$

Then, still following Lin et al. [38], another attention-only transformer can be constructed so that $N^t(s^t)$ can be computed in $h^d_{2t+1}$. The constructed transformer has ReLU activation and

$$L = 2, \quad \max_{l \in [L]} M^{(l)} = O(1), \quad \|\boldsymbol{\theta}\| = O(H).$$

In summary, at the end of Step 1, we can obtain that for all $t' \le t$,

$$\boldsymbol{h}^d_{2t'} = \begin{bmatrix} \mathbf{0} \\ \mathbf{pos}_{2t'} \end{bmatrix}, \quad \boldsymbol{h}^d_{2t'+1} = \begin{bmatrix} s^{t'} \\ a^{t'} \\ r^{t'} \\ N^{t'}(s^{t'}) \\ \mathbf{0} \\ \mathbf{pos}_{2t'+1} \end{bmatrix},$$

with an attention-only transformer that uses ReLU activation and

$$L = 4, \quad \max_{l \in [L]} M^{(l)} = O(1), \quad \|\boldsymbol{\theta}\| = O(HG).$$

### G.3.4   Proof of Step 2: Compute $\tilde{V}^h(s^{g,h})$ and $V^h(s^{g,h})$

In Step 2, let $n^t = N^t(s^t)$, the following computations will be performed:

$$\tilde{V}^t_{\text{new}}(s^t) \leftarrow (1 - \alpha_{n^t}) \tilde{V}^t_{\text{old}}(s^t) + \alpha_{n^t} \left( r^t + V^{t+1}_{\text{old}}(s^{t+1}) + \beta_{n^t} \right),$$

$$V^t_{\text{new}}(s^t) \leftarrow \min \left\{ H + 1 - h, \tilde{V}^t_{\text{new}}(s^t) \right\}.$$

First, similar to Step 2 in realizing UCB-VI in Lin et al. [38], we can obtain $\alpha_{n^{t'}}$ and $\beta_{n^{t'}}$ in each $\boldsymbol{h}^d_{2t'+1}$ via a transformer with ReLU activation and

$$L = O(1), \quad \max_{l \in [L]} M^{(l)} = O(1), \quad d' = O(G), \quad \|\boldsymbol{\theta}\| = O(G^3).$$

Then, we can assume that the values of $\{\tilde{V}^h_{\text{old}}(s), V^h_{\text{old}}(s) : h \in [H], s \in \mathcal{S}\}$ is already computed in token $\boldsymbol{h}^d_{2\tau-1}$, and prove via induction to show that the set of new values can be computed in token $\boldsymbol{h}^d_{2\tau+1}$, i.e.,

$$\boldsymbol{h}^d_{2\tau-1} = \begin{bmatrix} \star \\ \tilde{V}^{1:H}_{\text{old}}(\cdot) \\ V^{1:H}_{\text{old}}(\cdot) \\ \mathbf{0} \\ \mathbf{pos}_{2\tau-1} \end{bmatrix}, \quad \boldsymbol{h}^d_{2\tau} = \begin{bmatrix} \mathbf{0} \\ \mathbf{pos}_{2\tau} \end{bmatrix}, \quad \boldsymbol{h}^d_{2\tau+1} = \begin{bmatrix} \mathbf{0} \\ \mathbf{pos}_{2\tau+1} \end{bmatrix}$$

$$\xrightarrow{\text{compute}} \boldsymbol{h}^d_{2\tau} = \begin{bmatrix} \mathbf{0} \\ \mathbf{pos}_{2\tau} \end{bmatrix}, \quad \boldsymbol{h}^d_{2\tau+1} = \begin{bmatrix} \star \\ \tilde{V}^{1:H}_{\text{new}}(\cdot) \\ V^{1:H}_{\text{new}}(\cdot) \\ \mathbf{0} \\ \mathbf{pos}_{2\tau+1} \end{bmatrix}.$$

In the following, we will show that this one-step update can be completed via a transformer that requires

$$L = O(1), \quad \max_{l \in [L]} M^{(l)} = O(HS^2), \quad d' = O(G), \quad \|\boldsymbol{\theta}\| = O(G^3 + GH),$$

and the overall updates can be completed via stacking $T$ similar transformers.

**Step 2.1: Obtain $\tau - |t' - 1 - \tau|$.**

First, we obtain an auxiliary value $\tau - |t' - 1 - \tau|$ in each $\boldsymbol{h}^d_{2t'-1}$ and $\boldsymbol{h}^d_{2t'}$ for all $t' \le t$. In particular, we can construct three MLP layers with ReLU activation, i.e., $\{\boldsymbol{W}^{(1)}_1, \boldsymbol{W}^{(1)}_2, \boldsymbol{W}^{(2)}_1, \boldsymbol{W}^{(2)}_2, \boldsymbol{W}^{(3)}_1, \boldsymbol{W}^{(3)}_2\}$ to sequentially compute that

$$[\ 0\ ] \to \sigma_r([\ t' - 1 - \tau\ ]);$$

$$\sigma_r\left(\begin{bmatrix} t'-1-\tau \end{bmatrix}\right) \to \sigma_r\left(\begin{bmatrix} t'-1-\tau \end{bmatrix}\right) + \sigma_r\left(\begin{bmatrix} \tau-t'+1 \end{bmatrix}\right) = \begin{bmatrix} |t'-1-\tau| \end{bmatrix}$$
$$\begin{bmatrix} |t'-1-\tau| \end{bmatrix} \to \begin{bmatrix} \tau - |t'-1-\tau| \end{bmatrix}.$$

It can be observed that $\tau - |t'-1-\tau|$ reaches its maximum $\tau$ at $t' = \tau+1$.

The required transformer can be summarized as

$$d' = 1, \ \ \|\boldsymbol{\theta}\| = O(GH).$$

**Step 2.2: Move $\tilde{V}_{\text{old}}^{1:H}(\cdot)$ and $V_{\text{old}}^{1:H}(\cdot)$.**

Then, we move $\tilde{V}_{\text{old}}^{1:H}(\cdot)$ and $V_{\text{old}}^{1:H}(\cdot)$ from $\boldsymbol{h}_{2\tau-1}$ to $\boldsymbol{h}_{2\tau+1}$. In particular, we can construct that

$$\boldsymbol{Q}_1 \boldsymbol{h}_{2t'} = \begin{bmatrix} v_{2t'}-1 \\ \tau-|t'-1-\tau| \\ -1 \\ t' \\ 1 \end{bmatrix}, \ \ \boldsymbol{Q}_1 \boldsymbol{h}_{2t'+1} = \begin{bmatrix} v_{2t'+1}-1 \\ \tau-|t'-\tau| \\ -1 \\ t'+1 \\ 1 \end{bmatrix};$$

$$\boldsymbol{K}_1 \boldsymbol{h}_{2t'} = \begin{bmatrix} 1 \\ 1 \\ \tau \\ -1 \\ t'+1 \end{bmatrix}, \ \ \boldsymbol{K}_1 \boldsymbol{h}_{2t'+1} = \begin{bmatrix} 1 \\ 1 \\ \tau \\ -1 \\ t'+2 \end{bmatrix};$$

$$\boldsymbol{V}_1 \boldsymbol{h}_{2\tau-1} = (2\tau+1) \begin{bmatrix} \tilde{V}_{\text{old}}^{1:h}(\cdot) \\ V_{\text{old}}^{1:H}(\cdot) \end{bmatrix}, \ \ \boldsymbol{V}_1 \boldsymbol{h}_{2\tau} = \boldsymbol{0}, \ \ \boldsymbol{V}_1 \boldsymbol{h}_{2\tau+1} = \boldsymbol{0}.$$

With ReLU activation, this construction leads to that

$$\boldsymbol{h}_{2t'} = \boldsymbol{h}_{2t'} + \frac{1}{2t'} \sum_{i \le 2t'} \sigma_r\left(-1-|t'-1-\tau|-t'+t(i)+1\right) \cdot \boldsymbol{V}_1 \boldsymbol{h}_i = \boldsymbol{0}, \ \ \forall t' \le t$$

$$\boldsymbol{h}_{2t'+1} = \boldsymbol{h}_{2t'+1} + \frac{1}{2t'+1} \sum_{i \le 2t'+1} \sigma_r\left(-|t'-\tau|-t'+t(i)+1\right) \cdot \boldsymbol{V}_1 \boldsymbol{h}_i$$

$$= \begin{cases} \frac{1}{2t'+1}\left(\boldsymbol{V}_1 \boldsymbol{h}_{2t'-1} + \boldsymbol{V}_1 \boldsymbol{h}_{2t'} + 2\boldsymbol{V}_1 \boldsymbol{h}_{2t'+1}\right) = \begin{bmatrix} \tilde{V}_{\text{old}}^{1:h}(\cdot) \\ V_{\text{old}}^{1:H}(\cdot) \end{bmatrix} & \text{if } t' = \tau \\ \boldsymbol{0} & \text{otherwise} \end{cases}.$$

The transformer required in Step 2.3 can be summarized as

$$L = 1, \ \ M^{(1)} = O(1), \ \ \|\boldsymbol{\theta}\| = O(GH).$$

**Step 2.3: Compute $\tilde{V}_{\text{new}}^{1:H}(\cdot)$ and $V_{\text{new}}^{1:H}(\cdot)$.**

Finally, we get to compute $\tilde{V}_{\text{new}}^{1:H}(\cdot)$ and $V_{\text{new}}^{1:H}(\cdot)$, where

$$\tilde{V}_{\text{new}}^{\tau}(s^{\tau}) = \tilde{V}_{\text{old}}^{\tau}(s^{\tau}) - \alpha_{n^{\tau}} \tilde{V}_{\text{old}}^{\tau}(s^{\tau}) + \alpha_{n^{\tau}} \left(r^{\tau} + V_{\text{old}}^{\tau+1}(s^{\tau+1}) + \beta_{n^{\tau}}\right).$$

We can have a $HS$-head transformer that

$$\boldsymbol{Q}_{h,s} \boldsymbol{h}_{2t'} = \begin{bmatrix} v_{2t'}-1 \\ \tau-|t'-1-\tau| \\ -1 \\ t' \\ 1 \\ \boldsymbol{0} \\ \boldsymbol{e}_{h'} \\ 1 \\ 0 \end{bmatrix}, \ \ \boldsymbol{Q}_{h,s} \boldsymbol{h}_{2t'+1} = \begin{bmatrix} v_{2t'+1}-1 \\ \tau-|t'-\tau| \\ -1 \\ t'+1 \\ 1 \\ s^{t'} \\ \boldsymbol{e}_{h'+1} \\ 1 \\ \alpha_{n^{t'}} \end{bmatrix},$$

$$\boldsymbol{K}_{h,s}\boldsymbol{h}_{2t'} = \begin{bmatrix} 1 \\ 1 \\ \tau \\ -1 \\ t' \\ \boldsymbol{e}_s \\ \boldsymbol{e}_{h+1} \\ -2 \\ 1 \end{bmatrix}, \quad \boldsymbol{K}_{h,s}\boldsymbol{h}_{2t'+1} = \begin{bmatrix} 1 \\ 1 \\ \tau \\ -1 \\ t'+1 \\ \boldsymbol{e}_s \\ \boldsymbol{e}_{h+1} \\ -2 \\ 1 \end{bmatrix},$$

$$\boldsymbol{V}_{h,s}\boldsymbol{h}_{2\tau+1} = -(2\tau+1)\cdot V^h_{\text{old}}(s)\cdot[\ \boldsymbol{e}_{h,s}\ ].$$

It holds that for all $t' \le t$,

$$\boldsymbol{h}_{2t'} = \boldsymbol{h}_{2t'} + \frac{1}{2t'}\sum_{h,s}\sum_{i\le 2t'}\sigma_r\left(-1-|t'-1-\tau|-t'+t(i)+\boldsymbol{e}_{h'}\cdot\boldsymbol{e}_{h+1}-2\right)\boldsymbol{V}_{h,s}\boldsymbol{h}_i = \boldsymbol{0}$$

$$\boldsymbol{h}_{2t'+1} = \boldsymbol{h}_{2t'+1} + \frac{1}{2t'+1}\sum_{h,s}\sum_{i\le 2t'+1}\sigma_r\left(-|t'-\tau|-t'-1+t(i)+s^{t'}\cdot\boldsymbol{e}_s+\boldsymbol{e}_{h'+1}\cdot\boldsymbol{e}_h+\alpha_{n^{t'}}-2\right)\boldsymbol{V}_{h,s}\boldsymbol{h}_i$$

$$= \begin{cases} V^{1:H}_{\text{old}}(\cdot) - \alpha_{n^{t'}}V^{h'}_{\text{old}}(s^{t'})\cdot\boldsymbol{e}_{h',s^{t'}} & \text{if } t' = \tau \\ \boldsymbol{0} & \text{otherwise} \end{cases},$$

which completes the computation $(1-\alpha_{n^\tau})\tilde{V}^\tau_{\text{old}}(s^\tau)$.

Two similar $HS$-head transformers can further perform $(1-\alpha_{n^\tau})\tilde{V}^\tau_{\text{old}}(s^\tau) + \alpha_{n^\tau}r^\tau + \alpha_{n^\tau}\beta_{n^\tau}$. and a similar $HS^2$-head transformer can finalize $\tilde{V}^\tau_{\text{new}}(s^\tau)$ as $(1-\alpha_{n^\tau})\tilde{V}^\tau_{\text{old}}(s^\tau) + \alpha_{n^\tau}r^\tau + \alpha_{n^\tau}\beta_{n^\tau} + \alpha_{n^\tau}V^{\tau+1}_{\text{old}}(s^{\tau+1})$.

Finally, from the proof of Step 3 in realizing UCB-VI in Lin et al. [38], one MLP layer can perform

$$V^\tau_{\text{old}}(s^\tau) = \min\left\{\tilde{V}^\tau_{\text{new}}(s^\tau), H-h+1\right\}.$$

The required transformer in step 2.4 can be summarized as

$$L = 4, \quad \max_{l\in[L]}M^{(l)} = O(HS^2), \quad d' = O(1), \quad \|\boldsymbol{\theta}\| = O(GH).$$

Thus, we can see that the update of $\boldsymbol{h}_{2\tau+1}$ can be done. Repeating the similar step for each $\tau \in [T]$ would complete the overall updates.

### G.3.5 Proof of Step 3: Compute $\tilde{\ell}_{n^t}(s^{g,h}, a)$ and $\mu^h(a|s^{g,h})$

In step 3, let $n^t = N^t(s^t)$, for step $t$, we update

$$\tilde{\ell}^t_{n^t}(s^t, a) \leftarrow \frac{H - r^t - V^{t+1}(s^{t+1})}{H}\cdot\frac{\mathbf{1}\{a = a^t\}}{\mu^t_{\text{old}}(a^t|s^t) + \gamma_{n^t}};$$

$$\mu^t_{\text{new}}(a|s^t) \propto \exp\left(-\frac{\eta_{n^t}}{\omega_{n^t}}\cdot\sum_{i\in[n^t]}\omega_i\cdot\tilde{\ell}^t_i(s^t, a)\right).$$

First, similar to Step 2, we can obtain $\omega_{n^{t'}}, \eta_{n^{t'}}/\omega_{n^{t'}}$ and $\gamma_{n^{t'}}$ in each $\boldsymbol{h}^d_{2t'+1}$ via a transformer with ReLU activation and

$$L = O(1), \quad \max_{l\in[L]}M^{(l)} = O(1), \quad d' = O(G), \quad \|\boldsymbol{\theta}\| = O(G^3).$$

Denoting vectors $\mathcal{L} \in \mathbb{R}^{HSA}$ and $\boldsymbol{\mu} \in \mathbb{R}^{HSA}$ containing the cumulative losses and policies, i.e., $\sum_{i\in[n]}\omega_i\tilde{\ell}^h_i(s, a)$ and $\mu^h(a|s)$. Similar to step 2, we will assume that the values of $\{\mathcal{L}_{\text{old}}, \boldsymbol{\mu}_{\text{old}}\}$,

which are computed via information before time $\tau$, are already contained in token $\boldsymbol{h}_{2\tau-1}^d$, and prove via induction to show that the set of new values can be computed in token $\boldsymbol{h}_{2\tau+1}^d$, i.e.,

$$\boldsymbol{h}_{2\tau-1}^d = \begin{bmatrix} \star \\ \mathcal{L}_{\text{old}} \\ \boldsymbol{\mu}_{\text{old}} \\ \boldsymbol{0} \\ \mathbf{pos}_{2\tau-1} \end{bmatrix}, \ \boldsymbol{h}_{2\tau}^d = \begin{bmatrix} \boldsymbol{0} \\ \mathbf{pos}_{2\tau} \end{bmatrix}, \ \boldsymbol{h}_{2\tau+1}^d = \begin{bmatrix} \boldsymbol{0} \\ \mathbf{pos}_{2\tau+1} \end{bmatrix}$$

$$\xrightarrow{\text{compute}} \boldsymbol{h}_{2\tau}^d = \begin{bmatrix} \boldsymbol{0} \\ \mathbf{pos}_{2\tau} \end{bmatrix}, \ \boldsymbol{h}_{2\tau+1}^d = \begin{bmatrix} \star \\ \mathcal{L}_{\text{new}} \\ \boldsymbol{\mu}_{\text{new}} \\ \boldsymbol{0} \\ \mathbf{pos}_{2\tau+1} \end{bmatrix}.$$

In the following, we will show that this one-step update can be completed via a transformer that requires

$$L = O(L_D), \quad \max_{l\in[L]} M^{(l)} = O(HSA + M_D), \quad d' = O(d_D + A), \quad \|\boldsymbol{\theta}\| = O(GH^2S + F_D),$$

and the overall updates can be completed via stacking $T$ similar transformers.

**Step 3.1: Obtain $\tau - |t' - 1 - \tau|$.**

First, similar to step 2.1, we can have an auxiliary value $\tau - |t' - 1 - \tau|$ in each $\boldsymbol{h}_{2t'-1}^d$ and $\boldsymbol{h}_{2t'}^d$ for all $t' < t$ via three MLP layers. The required transformer can be summarized as

$$d' = 1, \quad \|\boldsymbol{\theta}\| = O(GH).$$

**Step 3.2: Compute $\tilde{\ell}_{n^\tau}^\tau(s^\tau, a)$.**

First, similar to Step 2.2, we can move $\mathcal{L}_{\text{old}}$ and $\boldsymbol{\mu}_{\text{old}}$ from $\boldsymbol{h}_{2\tau-1}^d$ to $\boldsymbol{h}_{2\tau+1}^d$ while keeping other tokens unchanged. In particular, we can construct that

$$\boldsymbol{Q}_1 \boldsymbol{h}_{2t'} = \begin{bmatrix} v_{2t'} - 1 \\ \tau - |t' - 1 - \tau| \\ -1 \\ t' \\ 1 \end{bmatrix}, \quad \boldsymbol{Q}_1 \boldsymbol{h}_{2t'+1} = \begin{bmatrix} v_{2t'+1} - 1 \\ \tau - |t' - \tau| \\ -1 \\ t' + 1 \\ 1 \end{bmatrix};$$

$$\boldsymbol{K}_1 \boldsymbol{h}_{2t'} = \begin{bmatrix} 1 \\ 1 \\ \tau \\ -1 \\ t' + 1 \end{bmatrix}, \quad \boldsymbol{K}_1 \boldsymbol{h}_{2t'+1} = \begin{bmatrix} 1 \\ 1 \\ \tau \\ -1 \\ t' + 2 \end{bmatrix};$$

$$\boldsymbol{V}_1 \boldsymbol{h}_{2\tau-1} = (2\tau + 1)\begin{bmatrix} \mathcal{L}_{\text{old}} \\ \boldsymbol{\mu}_{\text{old}} \end{bmatrix}, \quad \boldsymbol{V}_1 \boldsymbol{h}_{2\tau} = \boldsymbol{0}, \quad \boldsymbol{V}_1 \boldsymbol{h}_{2\tau+1} = \boldsymbol{0}.$$

With ReLU activation, this construction leads to that

$$\boldsymbol{h}_{2t'} = \boldsymbol{0}, \ \forall t' \leq t; \quad \boldsymbol{h}_{2t'+1} = \begin{cases} \begin{bmatrix} \mathcal{L}_{\text{old}} \\ \boldsymbol{\mu}_{\text{old}} \end{bmatrix} & \text{if } t' = \tau \\ \boldsymbol{0} & \text{otherwise} \end{cases}.$$

Then, using similar constructions as Step 2.3, we can specifically extract $V^{\tau+1}(s^{\tau+1})$ and $\mu_{\text{old}}^\tau(a^\tau|s^\tau)$ to $\boldsymbol{h}_{2\tau+1}^d$ while keeping other tokens unchanged. Following the extraction, one MLP layer can compute the value $\frac{H - r^\tau - V^{\tau+1}(s^{\tau+1})}{H}$ and add it to $\boldsymbol{h}_{2\tau+1}^d$. The required transformer can be summarized as

$$L = O(1), \quad \max_{l\in[L]} M^{(l)} = O(HSA), \quad d' = 1, \quad \|\boldsymbol{\theta}\| = O(GH).$$

With Assumption G.2, for token $\boldsymbol{h}_{2\tau+1}$, we can further have one transformer to compute

$$\tilde{\ell}_{n^\tau}^\tau(s^\tau, a^\tau) = \frac{H - r^\tau - V^{\tau+1}(s^{\tau+1})}{H} \cdot \frac{1}{\mu_{\text{old}}^\tau(a^\tau|s^\tau) + \gamma_{n^\tau}}.$$

The overall required transformer can be summarized as

$$L = O(L_D), \quad \max_{l \in [L]} M^{(l)} = O(HSA + M_D), \quad d' = O(d_D), \quad \|\boldsymbol{\theta}\| = O(GH + F_D).$$

**Step 3.3: Update $\mu^\tau(a|s^\tau)$.**

First, one transformer can be constructed to update $\mathcal{L}_{\text{old}}$ to $\mathcal{L}_{\text{new}}$ in $\boldsymbol{h}_{2\tau+1}^d$. In particular, we can have

$$\boldsymbol{Q}_{h,s}\boldsymbol{h}_{2t'} = \begin{bmatrix} v_{2t'} - 1 \\ \tau - |t' - 1 - \tau| \\ 1 \\ t' \\ 1 \\ \boldsymbol{0} \\ \boldsymbol{e}_{h'} \\ 1 \end{bmatrix}, \quad \boldsymbol{Q}_{h,s}\boldsymbol{h}_{2t'+1} = \begin{bmatrix} v_{2t'+1} - 1 \\ \tau - |t' - \tau| \\ 1 \\ t' + 1 \\ 1 \\ s^{t'} \\ \boldsymbol{e}_{h'+1} \\ 1 \end{bmatrix},$$

$$\boldsymbol{K}_{h,s}\boldsymbol{h}_{2t'} = \begin{bmatrix} 1 \\ 1 \\ -\tau \\ -1 \\ t' \\ \boldsymbol{e}_s \\ \boldsymbol{e}_{h+1} \\ -1 \end{bmatrix}, \quad \boldsymbol{K}_{h,s}\boldsymbol{h}_{2t'+1} = \begin{bmatrix} 1 \\ 1 \\ -\tau \\ -1 \\ t' + 1 \\ \boldsymbol{e}_s \\ \boldsymbol{e}_{h+1} \\ -1 \end{bmatrix},$$

$$\boldsymbol{V}_{h,s}\boldsymbol{h}_{2\tau+1} = -(2\tau + 1) \cdot \mathcal{L}_{\text{old}}(h, s, \cdot).$$

With ReLU activation, this construction leads to that

$$\boldsymbol{h}_{2t'} = \boldsymbol{0}, \quad \forall t' \le t; \quad \boldsymbol{h}_{2t'+1} = \begin{cases} \mathcal{L}_{\text{old}}(h(\tau), s^\tau, \cdot) & \text{if } t' = \tau \\ \boldsymbol{0} & \text{otherwise} \end{cases}.$$

With a similar $A$-head transformer, we can add $\omega_{n^\tau} \cdot \tilde{\ell}_{n^\tau}^\tau(s^\tau, a^\tau)$ to $\mathcal{L}_{\text{old}}(h(\tau), s^\tau, a^\tau)$ and thus obtain the new values $\mathcal{L}_{\text{new}}(h(\tau), s^\tau, \cdot)$. Furthermore, a one-head transformer can obtain

$$\boldsymbol{h}_{2\tau+1} = \frac{\eta_{n^\tau}}{\omega_{n^\tau}} \cdot \mathcal{L}_{\text{new}}(h(\tau), s^\tau, \cdot)$$

Finally, with another softmax MLP layer, we can obtain $\mu_{\text{new}}^\tau(\cdot|s^\tau)$.

The required transformer can be summarized as

$$L = O(1), \quad \max_{l \in [L]} M^{(l)} = O(HS + A), \quad d' = O(A), \quad \|\boldsymbol{\theta}\| = O(GH^2S).$$

### G.3.6 Proof of Step 4: Obtain $\mu^{h+1}(\cdot|s^{h+1})$

This step is essentially the same as Step 6 in realizing VI-ULCB, which can be completed with a transformer that

$$L = 1, \quad M^{(1)} = HS, \quad \|\boldsymbol{\theta}\| = O(HS).$$

## H   Proofs for the Decentralized Overall Performance

*Proof of Theorem 3.5.* We use the decomposition that

$$\mathbb{E}_{M \sim \Lambda, D \sim \mathbb{P}_M^{\text{Alg}_{\hat{\boldsymbol{\theta}}}}} \left[ V_M^{\dagger,\hat{\nu}}(s^1) - V_M^{\hat{\mu},\dagger}(s^1) \right] = \mathbb{E}_{M \sim \Lambda, D \sim \mathbb{P}_M^{\text{Alg}_0}} \left[ V_M^{\dagger,\hat{\nu}}(s^1) - V_M^{\hat{\mu},\dagger}(s^1) \right]$$

$$+ \mathbb{E}_{M\sim\Lambda, D\sim\mathbb{P}_M^{\mathtt{Alg}\hat{\theta}}} \left[ V_M^{\dagger,\hat{\nu}}(s^1) \right] - \mathbb{E}_{M\sim\Lambda, D\sim\mathbb{P}_M^{\mathtt{Alg}0}} \left[ V_M^{\dagger,\hat{\nu}}(s^1) \right]$$

$$+ \mathbb{E}_{M\sim\Lambda, D\sim\mathbb{P}_M^{\mathtt{Alg}0}} \left[ V_M^{\hat{\mu},\dagger}(s^1) \right] - \mathbb{E}_{M\sim\Lambda, D\sim\mathbb{P}_M^{\mathtt{Alg}\hat{\theta}}} \left[ V_M^{\hat{\mu},\dagger}(s^1) \right].$$

First, via Theorem G.1, it holds that

$$\mathbb{E}_{M\sim\Lambda, D\sim\mathbb{P}_M^{\mathtt{Alg}0}} \left[ V_M^{\dagger,\hat{\nu}}(s^1) - V_M^{\hat{\mu},\dagger}(s^1) \right] = O\left( \sqrt{\frac{H^5 S(A+B)\log(SABT)}{G}} \right).$$

Then, via Lemma 3.6 and Theorem 3.3, we can obtain that

$$\mathbb{E}_{M\sim\Lambda, D\sim\mathbb{P}_M^{\mathtt{Alg}0}} \left[ V_M^{\hat{\mu},\dagger}(s^1) \right] - \mathbb{E}_{M\sim\Lambda, D\sim\mathbb{P}_M^{\mathtt{Alg}\hat{\theta}}} \left[ V_M^{\hat{\mu},\dagger}(s^1) \right]$$

$$= O\left( H \cdot \sum_{t\in[T],s\in\mathcal{S}} \mathbb{E}_\alpha \left[ \mathrm{TV}\left( \mu_\alpha^t, \mu_\beta^t | D_+^{t-1}, s \right) + \mathrm{TV}\left( \nu_\alpha^t, \nu_\beta^t | D_-^{t-1}, s \right) \right] \right)$$

$$= O\left( THS\sqrt{\varepsilon_{+,\mathrm{real}}} + THS\sqrt{\varepsilon_{-,\mathrm{real}}} + THS\sqrt{\frac{\log\left(\mathcal{N}_{\Theta_+} TS/\delta\right)}{N}} + THS\sqrt{\frac{\log\left(\mathcal{N}_{\Theta_-} TS/\delta\right)}{N}} \right),$$

and similarly,

$$\mathbb{E}_{M\sim\Lambda, D\sim\mathbb{P}_M^{\mathtt{Alg}\hat{\theta}}} \left[ V_M^{\dagger,\hat{\nu}}(s^1) \right] - \mathbb{E}_{M\sim\Lambda, D\sim\mathbb{P}_M^{\mathtt{Alg}0}} \left[ V_M^{\dagger,\hat{\nu}}(s^1) \right]$$

$$= O\left( THS\sqrt{\varepsilon_{+,\mathrm{real}}} + THS\sqrt{\varepsilon_{-,\mathrm{real}}} + THS\sqrt{\frac{\log\left(\mathcal{N}_{\Theta_+} TS/\delta\right)}{N}} + THS\sqrt{\frac{\log\left(\mathcal{N}_{\Theta_-} TS/\delta\right)}{N}} \right).$$

With Theorem 3.4 providing that $\varepsilon_{+,\mathrm{real}} = \varepsilon_{-,\mathrm{real}} = 0$, combining the above terms completes the proof of the regret bound. The bound on the covering number, i.e., $\log(N_{\Theta_+})$ and $\log(N_{\Theta_-})$ can be obtained via Lemma I.4. $\qquad\square$

In the following, we provide the proof for the Lemma 3.6, which plays a key role in the above proof of the overall performance.

*Proof of Lemma 3.6.* It holds that

$$\mathbb{E}_{D\sim\mathbb{P}_M^{\mathtt{Alg}\alpha}} \left[ V_M^{\hat{\mu}_\alpha,\dagger}(s^1) \right] - \mathbb{E}_{D\sim\mathbb{P}_M^{\mathtt{Alg}\beta}} \left[ V_M^{\hat{\mu}_\beta,\dagger}(s^1) \right]$$

$$= \underbrace{\mathbb{E}_{D\sim\mathbb{P}_M^{\mathtt{Alg}\alpha}} \left[ V_M^{\hat{\mu}_\alpha,\dagger}(s^1) - V_M^{\hat{\mu}_\beta,\dagger}(s^1) \right]}_{:=(\text{term I})} + \underbrace{\mathbb{E}_{D\sim\mathbb{P}_M^{\mathtt{Alg}\alpha}} \left[ V_M^{\hat{\mu}_\beta,\dagger}(s^1) \right] - \mathbb{E}_{D\sim\mathbb{P}_M^{\mathtt{Alg}\beta}} \left[ V_M^{\hat{\mu}_\beta,\dagger}(s^1) \right]}_{:=(\text{term II})}.$$

Denoting $f(D^H) = \sum_{h\in[H]} r^h$, we can obtain that

$$V_M^{\hat{\mu}_\alpha,\dagger}(s^1) - V_M^{\hat{\mu}_\beta,\dagger}(s^1)$$

$$\overset{(a)}{\leq} V_M^{\hat{\mu}_\alpha,\nu_\dagger(\hat{\mu}_\beta)}(s^1) - V_M^{\hat{\mu}_\beta,\nu_\dagger(\hat{\mu}_\beta)}(s^1)$$

$$\overset{(b)}{=} \frac{1}{G} \sum_{g\in[G]} \left[ V_M^{\hat{\mu}_\alpha^g,\nu_\dagger(\hat{\mu}_\beta)}(s^1) - V_M^{\hat{\mu}_\beta^g,\nu_\dagger(\hat{\mu}_\beta)}(s^1) \right]$$

$$\leq \frac{1}{G} \sum_{g\in[G]} \sum_{h\in[H]} \left[ \mathbb{E}_{D^{1:h}\sim\mathbb{P}_M^{\hat{\mu}_\alpha^g,\nu_\dagger(\hat{\mu}_\beta)}, D^{h+1:H}\sim\mathbb{P}_M^{\hat{\mu}_\beta^g,\nu_\dagger(\hat{\mu}_\beta)}} \left[ f(D^H) \right] \right]$$

$$- \frac{1}{G} \sum_{g\in[G]} \sum_{h\in[H]} \left[ \mathbb{E}_{D^{1:h-1}\sim\mathbb{P}_M^{\hat{\mu}_\alpha^g,\nu_\dagger(\hat{\mu}_\beta)}, D^{h:H}\sim\mathbb{P}_M^{\hat{\mu}_\beta^g,\nu_\dagger(\hat{\mu}_\beta)}} \left[ f(D^H) \right] \right]$$

$$\overset{(c)}{\leq} \frac{2H}{G} \sum_{g\in[G]} \sum_{h\in[H]} \mathbb{E}_{D^{1:h-1}\sim\mathbb{P}_M^{\hat{\mu}_\alpha^g,\nu_\dagger(\hat{\mu}_\beta)}, s^h} \left[ \mathrm{TV}\left( \hat{\mu}_\alpha^{g,h} \times \nu_\dagger^h(\hat{\mu}_\beta)(\cdot,\cdot|s^h), \hat{\mu}_\beta^{g,h} \times \nu_\dagger^h(\hat{\mu}_\beta)(\cdot,\cdot|s^h) \right) \right]$$

$$= \frac{2H}{G} \sum_{g \in [G]} \sum_{h \in [H]} \mathbb{E}_{D^{1:h-1} \sim \mathbb{P}_M^{\hat{\mu}_\alpha^g, \nu_\dagger(\hat{\mu}_\beta)}, s^h} \left[ \mathrm{TV} \left( \hat{\mu}_\alpha^{g,h}(\cdot | s^h), \hat{\mu}_\beta^{g,h}(\cdot | s^h) \right) \right]$$

$$\overset{(d)}{=} \frac{2H}{G} \sum_{g \in [G]} \sum_{h \in [H]} \mathbb{E}_{D^{1:h-1} \sim \mathbb{P}_M^{\hat{\mu}_\alpha^g, \nu_\dagger(\hat{\mu}_\beta)}, s^h} \left[ \mathrm{TV} \left( \sum_{i \in [n]} \alpha_{n,i} \mu_\alpha^{g_i, h}(\cdot | s^h), \sum_{i \in [n]} \alpha_{n,i} \mu_\beta^{g_i, h}(\cdot | s^h) \right) \right]$$

$$\overset{(e)}{\leq} \frac{2H}{G} \sum_{g \in [G]} \sum_{h \in [H]} \mathbb{E}_{D^{1:h-1} \sim \mathbb{P}_M^{\hat{\mu}_\alpha^g, \nu_\dagger(\hat{\mu}_\beta)}, s^h} \left[ \sum_{i \in [n]} \alpha_{n,i} \mathrm{TV} \left( \mu_\alpha^{g_i, h}(\cdot | s^h), \mu_\beta^{g_i, h}(\cdot | s^h) \right) \right]$$

$$\leq \frac{2H}{G} \sum_{g \in [G]} \sum_{h \in [H]} \sum_{s \in \mathcal{S}} \sum_{i \in [n]} \alpha_{n,i} \mathrm{TV} \left( \mu_\alpha^{g_i, h}(\cdot | s), \mu_\beta^{g_i, h}(\cdot | s) \right)$$

$$\overset{(f)}{\leq} 2H \sum_{g \in [G]} \sum_{h \in [H]} \sum_{s \in \mathcal{S}} \mathrm{TV} \left( \mu_\alpha^{g, h}(\cdot | s), \mu_\beta^{g, h}(\cdot | s) \right),$$

where (a) is from the definition of best responses, (b) uses the notation $\hat{\mu}^g$ to denote the output policy from Alg. 4 with the initial random sampling result to be $g$, (c) uses the variational representation of the TV distance, (d) uses the abbreviations $n = N^{g,h}(s^h)$ and $g_i = g_i^h(s^h)$, (e) leverages the property of TV distance, and (f) uses the fact that $\alpha_{n,i} < 1$.

With the above result, we can further obtain that

$$(\text{term I}) := \mathbb{E}_{D \sim \mathbb{P}_M^{\mathrm{Alg}_\alpha}} \left[ V_M^{\hat{\mu}_\alpha, \dagger}(s^1) - V_M^{\hat{\mu}_\beta, \dagger}(s^1) \right] \leq \mathbb{E}_{D \sim \mathbb{P}_M^{\mathrm{Alg}_\alpha}} \left[ 2H \sum_{t \in [T]} \sum_{s \in \mathcal{S}} \mathrm{TV} \left( \mu_\alpha^t(\cdot | s), \mu_\beta^t(\cdot | s) \right) \right].$$

Also, for term (II), denoting $g(D) = V_M^{\hat{\mu}_\beta, \dagger}(s^1)$, it holds that

$$(\text{term II}) := \mathbb{E}_{D \sim \mathbb{P}_M^{\mathrm{Alg}_\alpha}} \left[ V_M^{\hat{\mu}_\beta, \dagger}(s^1) \right] - \mathbb{E}_{D \sim \mathbb{P}_M^{\mathrm{Alg}_\beta}} \left[ V_M^{\hat{\mu}_\beta, \dagger}(s^1) \right]$$

$$= \sum_{t \in [T]} \mathbb{E}_{D^t \sim \mathbb{P}_M^{\mathrm{Alg}_\alpha}, D^{t+1:T} \sim \mathbb{P}_M^{\mathrm{Alg}_\beta}} [g(D)] - \sum_{t \in [T]} \mathbb{E}_{D^{t-1} \sim \mathbb{P}_M^{\mathrm{Alg}_\alpha}, D^{t:T} \sim \mathbb{P}_M^{\mathrm{Alg}_\beta}} [g(D)]$$

$$\leq 2H \sum_{t \in [T]} \mathbb{E}_{D^{t-1} \sim \mathbb{P}_M^{\mathrm{Alg}_\alpha}, s^t} \left[ \mathrm{TV} \left( \mathrm{Alg}_\alpha(\cdot, \cdot | D^{t-1}, s^t), \mathrm{Alg}_\beta(\cdot, \cdot | D^{t-1}, s^t) \right) \right]$$

$$\leq 2H \sum_{t \in [T]} \sum_{s \in \mathcal{S}} \mathbb{E}_{D^{t-1} \sim \mathbb{P}_M^{\mathrm{Alg}_\alpha}} \left[ \mathrm{TV} \left( \mu_\alpha^t(\cdot | D_+^{t-1}, s^t), \mu_\beta^t(\cdot, \cdot | D_+^{t-1}, s^h) \right) \right]$$

$$+ 2H \sum_{t \in [T]} \sum_{s \in \mathcal{S}} \mathbb{E}_{D^{t-1} \sim \mathbb{P}_M^{\mathrm{Alg}_\alpha}} \left[ \mathrm{TV} \left( \nu_\alpha^t(\cdot | D_+^{t-1}, s), \nu_\beta^t(\cdot, \cdot | D_+^{t-1}, s) \right) \right].$$

Combining (term I) and (term II) concludes the proof. □

# I Discussions on the Covering Number

In the following, we characterize the covering number of algorithms induced by transformers parameterized by $\theta \in \Theta_{d,L,M,d',F}$, i.e., $\{\mathrm{Alg}_\theta : \theta \in \Theta_{d,L,M,d',F}\}$. Here we will mainly take the perspective of the centralized setting, while the extension to the decentralized setting is straightforward.

To facilitate the discussion, we introduce the following clipped transformer, where an additional clip operator is adopted to bound the output of the transformer.

**Definition I.1** (Clipped Decoder-based Transformer). *An $L$-layer clipped decoder-based transformer, denoted as $\mathrm{TF}_\theta^R(\cdot)$, is a composition of $L$ masked attention layers, each followed by an MLP layer and a clip operation: $\mathrm{TF}_\theta^R(\boldsymbol{H}) = \boldsymbol{H}^{(L)} \in \mathbb{R}^{d \times N}$, where $\boldsymbol{H}^{(L)}$ is defined iteratively by taking $\boldsymbol{H}^{(0)} = \mathrm{clip}_R(\boldsymbol{H}) \in \mathbb{R}^{d \times N}$ and for $l \in [L]$,*

$$\boldsymbol{H}^{(l)} = \mathrm{clip}_R \left( \mathrm{MLP}_{\theta_{\mathrm{mlp}}^{(l)}} \left( \mathrm{Attn}_{\theta_{\mathrm{mattn}}^{(l)}} \left( \boldsymbol{H}^{(l-1)} \right) \right) \right) \in \mathbb{R}^{d \times N},$$

*where $\mathrm{clip}_R(\boldsymbol{H}) = [\mathrm{proj}_{\|\boldsymbol{h}\|_2 \leq R}(\boldsymbol{h}_i) : i \in [N]]$.*

Furthermore, for $\zeta \in (0, 1]$, we define the following $\zeta$-biased algorithm induced by transformer $\mathrm{TF}_{\boldsymbol{\theta}}^R(\boldsymbol{H})$ as

$$\mathtt{Alg}_{\boldsymbol{\theta}}^{\zeta}(\cdot, \cdot | D^{t-1}, s^t) = (1 - \zeta) \cdot \mathrm{proj}_{\Delta} \left( \mathtt{E} \cdot \mathrm{TF}_{\boldsymbol{\theta}}^R \left( \mathtt{h}(D^{t-1}, s^t) \right)_{-1} \right) + \frac{\zeta}{AB} \cdot \mathbf{1}_{AB},$$

with $\mathbf{1}_{AB}$ denoting an all-one vector of dimension $AB$, which introduces a lower bound $\zeta/AB$ for the probably each pair $(a, b) \in \mathcal{A} \times \mathcal{B}$ to be sampled.

Finally, for any $\boldsymbol{H} = [\boldsymbol{h}_1, \cdots, \boldsymbol{h}_N] \in \mathbb{R}^{d \times N}$, we denote

$$\|\boldsymbol{H}\|_{2,\infty} := \max_{i \in [N]} \|\boldsymbol{h}_i\|_2.$$

**Proposition I.2** (Modified from Proposition J.1 in Bai et al. [7]). *For any $\boldsymbol{\theta}_1, \boldsymbol{\theta}_2 \in \Theta_{d,L,M,d',F}$, we have*

$$\|\mathrm{TF}_{\boldsymbol{\theta}_1}^R(\boldsymbol{H}) - \mathrm{TF}_{\boldsymbol{\theta}_2}^R(\boldsymbol{H})\|_{2,\infty} \leq LF_H^{L-1}F_{\Theta}\|\boldsymbol{\theta}_1 - \boldsymbol{\theta}_2\|,$$

*where $F_{\Theta} := FR(2 + FR^2 + F^3R^2)$ and $F_H := (1 + F^2)(1 + F^2R^3)$.*

*Proof.* This proposition can be obtained similarly as Proposition J.1 in Bai et al. [7] with the following Lemma I.3 in the place of Lemma J.1 in Bai et al. [7]. $\square$

**Lemma I.3** (Modified from Lemma J.1 in Bai et al. [7]). *For a single MLP layer $\boldsymbol{\theta}_{\mathrm{mlp}} = (\boldsymbol{W}_1, \boldsymbol{W}_2)$, we introduce its norm*

$$\|\boldsymbol{\theta}_{\mathrm{mlp}}\| = \|\boldsymbol{W}_1\|_{\mathrm{op}} + \|\boldsymbol{W}_2\|_{\mathrm{op}}.$$

*For any fixed hidden dimension $D'$, we consider*

$$\Theta_{\mathrm{mlp},F} := \{\boldsymbol{\theta}_{\mathrm{mlp}} : \|\boldsymbol{\theta}_{\mathrm{mlp}}\| \leq F\}.$$

*Then, for any $\boldsymbol{H} \in \mathcal{H}_R, \boldsymbol{\theta}_{\mathrm{mlp}} \in \Theta_{\mathrm{mlp},F}$, it holds that $\mathrm{MLP}_{\boldsymbol{\theta}_{\mathrm{mlp}}}(\boldsymbol{H})$ is $2FR$-Lipschitz in $\boldsymbol{\theta}_{\mathrm{mlp}}$ and $(1 + F^2)$-Lipschitz in $\boldsymbol{H}$.*

*Proof.* It holds that

$$\|\mathrm{MLP}_{\boldsymbol{\theta}_{\mathrm{mlp}}}(\boldsymbol{H}) - \mathrm{MLP}_{\boldsymbol{\theta}'_{\mathrm{mlp}}}(\boldsymbol{H})\|_{2,\infty}$$
$$= \max_i \|\boldsymbol{W}_2\sigma(\boldsymbol{W}_1\boldsymbol{h}_i) - \boldsymbol{W}'_2\sigma(\boldsymbol{W}'_1\boldsymbol{h}_i)\|_2$$
$$\leq \max_i \|\boldsymbol{W}_2 - \boldsymbol{W}'_2\|_{\mathrm{op}}\|\sigma(\boldsymbol{W}'_1\boldsymbol{h}_i)\|_2 + \|\boldsymbol{W}'_2\|_{\mathrm{op}}\|\sigma(\boldsymbol{W}_1\boldsymbol{h}_i) - \sigma(\boldsymbol{W}'_1\boldsymbol{h}_i)\|_2$$
$$\overset{(a)}{\leq} \max_i \|\boldsymbol{W}_2 - \boldsymbol{W}'_2\|_{\mathrm{op}}\max\{1, \|\boldsymbol{W}_1\boldsymbol{h}_i\|_2\} + \|\boldsymbol{W}'_2\|_{\mathrm{op}}\|\boldsymbol{W}_1\boldsymbol{h}_i - \boldsymbol{W}'_1\boldsymbol{h}_i\|_2$$
$$\leq (1 + FR)\|\boldsymbol{W}_2 - \boldsymbol{W}'_2\|_{\mathrm{op}} + FR\|\boldsymbol{W}_1 - \boldsymbol{W}'_1\|_2.$$

Inequality (a) is from that

$$\|\sigma_{\mathrm{r}}(x)\|_2 \leq \|x\|_2, \quad \|\sigma_{\mathrm{s}}(x)\|_2 \leq 1,$$

and

$$\|\sigma_{\mathrm{r}}(x) - \sigma_{\mathrm{r}}(y)\|_2 \leq \|x - y\|_2, \quad \|\sigma_{\mathrm{s}}(x) - \sigma_{\mathrm{s}}(y)\|_2 \leq \|x - y\|_2,$$

where the last result on the Lipschitzness of softmax is adopted from Gao and Pavel [25].

Similarly, we can obtain that

$$\|\mathrm{MLP}_{\boldsymbol{\theta}_{\mathrm{mlp}}}(\boldsymbol{H}) - \mathrm{MLP}_{\boldsymbol{\theta}_{\mathrm{mlp}}}(\boldsymbol{H}')\|_{2,\infty}$$
$$= \max_i \|\boldsymbol{h}_i + \boldsymbol{W}_1\sigma(\boldsymbol{W}_2\boldsymbol{h}_i) - \boldsymbol{h}'_i - \boldsymbol{W}_1\sigma(\boldsymbol{W}_2\boldsymbol{h}'_i)\|$$
$$\leq \|\boldsymbol{H} - \boldsymbol{H}'\|_{2,\infty} + \|\boldsymbol{W}_1\|_{\mathrm{op}}\max_i \|\sigma(\boldsymbol{W}_2\boldsymbol{h}_i) - \sigma(\boldsymbol{W}_2\boldsymbol{h}'_i)\|_2$$
$$\leq \|\boldsymbol{H} - \boldsymbol{H}'\|_{2,\infty} + \|\boldsymbol{W}_1\|_{\mathrm{op}}\max_i \|\boldsymbol{W}_2\boldsymbol{h}_i - \boldsymbol{W}_2\boldsymbol{h}'_i\|_2$$
$$\leq \|\boldsymbol{H} - \boldsymbol{H}'\|_{2,\infty} + F^2\|\boldsymbol{H} - \boldsymbol{H}'\|_{2,\infty},$$

which concludes the proof. $\square$

**Lemma I.4** (Modified from Lemma 16 in Lin et al. [38]). *For the space of transformers $\{\mathrm{TF}_{\boldsymbol{\theta}}^R : \boldsymbol{\theta} \in \Theta_{d,L,M,d',F}\}$, the covering number of the induced algorithms $\{\mathtt{Alg}_{\boldsymbol{\theta}}^\zeta : \boldsymbol{\theta} \in \Theta_{d,L,M,d',F}\}$ satisfies that*

$$\log \mathcal{N}_{\Theta_{d,L,M,d',F}}(\rho) = O\left(L^2 d(Md + d') \log\left(1 + \frac{\max\{F, R, L\}}{\zeta \rho}\right)\right).$$

*Proof.* First, similar to Lemma 16 in Lin et al. [38], we can use Example 5.8 in Wainwright [61] to obtain that the $\delta$-covering number of the ball $B_{\|\cdot\|}(F)$ with radius $F$ under norm $\|\cdot\|$, i.e., $B_{\|\cdot\|}(F) = \{\boldsymbol{\theta} : \|\boldsymbol{\theta}\| \leq F\}$, can be bounded as

$$\log N(\delta; B_{\|\cdot\|}(F), \|\cdot\|) \leq L(3Md^2 + 2dd') \log(1 + 2F/\delta).$$

Recall that the projection operation to a probability simplex is Lipschitz continuous, i.e.,

$$\|\mathrm{proj}_\Delta(\boldsymbol{x}) - \mathrm{proj}_\Delta(\boldsymbol{y})\|_2 \leq \|\boldsymbol{x} - \boldsymbol{y}\|_2.$$

Then, we can see that there exists a subset $\tilde{\Theta} \subset \Theta_{d,L,M,d',F}$ with size

$$\log |\Theta_{d,L,M,d',F}| \leq L(3Md^2 + 2dd') \log(1 + 2F/\delta)$$

such that for any $\boldsymbol{\theta} \in \Theta_{d,L,M,d',F}$, there exists $\tilde{\boldsymbol{\theta}} \in \tilde{\Theta}$ with

$$\left\|\log \mathtt{Alg}_{\tilde{\boldsymbol{\theta}}}^\zeta(\cdot, \cdot | D^{t-1}, s) - \log \mathtt{Alg}_{\boldsymbol{\theta}}^\zeta(\cdot, \cdot | D^{t-1}, s)\right\|_\infty$$

$$\leq \frac{AB}{\zeta} \left\|\mathtt{Alg}_{\tilde{\boldsymbol{\theta}}}^\zeta(\cdot, \cdot | D^{t-1}, s) - \mathtt{Alg}_{\boldsymbol{\theta}}^\zeta(\cdot, \cdot | D^{t-1}, s)\right\|_\infty$$

$$\leq \frac{AB}{\zeta} \left\|\mathrm{TF}_{\tilde{\boldsymbol{\theta}}}^R(\boldsymbol{H}) - \mathrm{TF}_{\boldsymbol{\theta}}^R(\boldsymbol{H})\right\|_{2,\infty}$$

$$\leq \frac{AB}{\zeta} \cdot L F_H^{L-1} F_\Theta \cdot \|\boldsymbol{\theta} - \tilde{\boldsymbol{\theta}}\|$$

$$\leq \frac{AB}{\zeta} L F_H^{L-1} F_\Theta \delta.$$

Let $\delta = \frac{\zeta \rho}{ABL F_H^{L-1} F_\Theta}$, we can obtain $\|\log \mathtt{Alg}_{\tilde{\boldsymbol{\theta}}}^\zeta(\cdot, \cdot | D^{t-1}, s) - \log \mathtt{Alg}_{\boldsymbol{\theta}}^\zeta(\cdot, \cdot | D^{t-1}, s)\|_\infty \leq \rho$, which proves that

$$\log(\mathcal{N}_{\Theta_{d,L,M,d',F}}(\rho))$$

$$\leq L(3Md^2 + 2dd') \log(1 + 2F/\delta)$$

$$= L(3Md^2 + 2dd') \log\left(1 + \frac{ABL F_H^{L-1} F_\Theta}{\zeta \rho}\right)$$

$$= O\left(L(3Md^2 + 2dd') \log\left(1 + \frac{ABL(1 + F^2)^{L-1}(1 + F^2 R^3)^{L-1} FR(2 + FR^2 + F^3 R^2)}{\zeta \rho}\right)\right)$$

$$= O\left(L^2(Md^2 + dd') \log\left(1 + \frac{\max\{A, B, F, R, L\}}{\zeta \rho}\right)\right),$$

which concludes the proof. $\qquad \square$

From the transformer construction in the proofs for Theorems 3.4 and 4.1, we can observe that it is sufficient to specify one $R$ with $\log(R) = \tilde{O}(1)$ without impacting the transformers' operations. Also, in Theorems 3.3 and C.3, $\rho$ is taken to be $1/N$.

Finally, for the introduced $\zeta$ parameter, it can be recognized that the induced algorithms discussed in the main paper, i.e. $\mathtt{Alg}_{\boldsymbol{\theta}}$, can be interpreted as $\zeta = 0$, which does not lead to a meaningful $\log(\mathcal{N}_{\Theta_{d,L,M,d',F}}(\rho))$ provided in Lemma I.4. However, a non-zero $\zeta$ can tackle this situation by only introducing an additional realization error. Especially, assuming Assumption C.2 can be achieved with $\varepsilon_{\mathrm{real}} = 0$, i.e., exactly realizing the context algorithm (as in Theorem 4.1), we can obtain that

$$\log\left(\mathbb{E}_{D \sim \mathbb{P}_\Lambda^{\mathtt{Alg}_0}}\left[\frac{\mathtt{Alg}_0(a, b | D^{t-1}, s)}{\mathtt{Alg}_{\boldsymbol{\theta}^*}^\zeta(a, b | D^{t-1}, s)}\right]\right)$$

$$= \log \left( \mathbb{E}_{D \sim \mathbb{P}_\Lambda^{\mathrm{Alg}_0}} \left[ \frac{\mathrm{Alg}_0(a,b|D^{t-1},s)}{\mathrm{Alg}_{\boldsymbol{\theta}^*}(a,b|D^{t-1},s)} \cdot \frac{\mathrm{Alg}_{\boldsymbol{\theta}^*}(a,b|D^{t-1},s)}{\mathrm{Alg}_{\boldsymbol{\theta}^*}^\zeta(a,b|D^{t-1},s)} \right] \right)$$

$$= \log \left( \mathbb{E}_{D \sim \mathbb{P}_\Lambda^{\mathrm{Alg}_0}} \left[ \frac{\mathrm{Alg}_{\boldsymbol{\theta}^*}(a,b|D^{t-1},s)}{(1-\zeta) \cdot \mathrm{Alg}_{\boldsymbol{\theta}^*}(a,b|D^{t-1},s) + \zeta/(AB)} \right] \right)$$

$$\leq \log \left( \frac{1}{(1-\zeta) + \zeta/(AB)} \right)$$

$$\leq \log \left( \frac{1}{1-\zeta} \right).$$

With $\zeta = O(1/N)$, an additional realization error $\varepsilon_{\mathrm{real}} = O(1/N)$ occurs, whose impact on the overall performance bound (i.e., Theorems 3.5 and 4.2) is non-dominating. As a result, for the parameterization provided in Theorem 4.1, a covering number satisfies $\log(\mathcal{N}_\Theta) = \tilde{O}(\mathrm{poly}(GHSAB)\log(N))$ can be obtained. Similarly, the results can be extended to the decentralized setting, where for the parameterization provided in Theorem 3.4, it holds that $\log(\mathcal{N}_{\Theta_+}) = \tilde{O}(\mathrm{poly}(GHSAL_DM_Dd_D)\log(NF_D))$.

# J  Details of Experiments

## J.1  Detail of Games

• **The normal-form games for the decentralized setting.** The normal-form games (i.e., matrix games) used in decentralized experiments of Sec. 5 have $A = B = 5$ actions for both players and $G = 3000$ episodes (i.e., a horizon of $T = 3000$ with $H = 1$), which can be interpreted as having $S = 1$ state.

• **The Markov games for the centralized setting.** The Markov games used in centralized experiments of Sec. 5 have $A = B = 5$ actions for both players, $S = 4$ states, $H = 2$ steps in each episode and $G = 300$ episodes (i.e., a horizon of $T = GH = 600$). The transitions are fixed for different games: when both players take the same actions, the state transits to the next one (i.e., $1 \rightarrow 2, \cdots, 4 \rightarrow 1$); otherwise, the state stays the same.

At the start of each game (during both training and inference), a $A \times B$ reward matrix $R_h(s, \cdot, \cdot)$ is generated for each step $h \in [H]$ and state $s \in \mathcal{S}$ with its elements independently sampled from a standard Gaussian distribution truncated on $[0,1]$. Then, the interactions proceed as follows: at each time step $h \in [H]$, the players select action $a$ and $b$ on state $s$ based on their computed policy distributions. After selecting their actions, the players receive rewards $R_h(s, a, b)$ and $-R_h(s, a, b)$, respectively, while the state transits to the new one.

## J.2  Collection of Pre-training Data

• **The EXP3 algorithm for the decentralized setting.** In the decentralized setting, both players are equipped with the EXP3 algorithm [5] to collect pre-training data. Up to time step $t$, the trajectory of the max-player is recorded as "$a_1, r_1, \cdots, a_t, r_t$", and that of the min-player as "$b_1, 1 - r_1, \cdots, b_t, 1 - r_t$",

• **The VI-ULCB algorithm for the centralized setting.** In the centralized setting, both players jointly follow the VI-ULCB algorithm [8] to collect pre-training data. Up to time step $t$, the trajectory is recorded as "$s_1, a_1, b_1, r_1, 1 - r_1, \cdots, s_t, a_t, b_t, r_t, 1 - r_t$".

We note that the decimal digits of the rewards are limited to two to facilitate tokenization, while $1 - r_i$ instead of $-r_i$ is adopted for the min-player to avoid the additional complexity of negative numbers.

## J.3  Transformer Structure and Training

The transformer architecture employed in our experiments is primarily based on the well-known GPT-2 model [43], and our implementation follows the miniGPT realization[3] for simplicity. The

---

[3]https://github.com/karpathy/minGPT

numbers of transformer layers and attention heads have been modified to make the entire transformer much smaller. In particular, we utilize a transformer with 2 layers and 4 heads. Given that the transformer is used to compute the policy, we modify the output layer of the transformer to make it aligned with the action dimension. We focus solely on the last output from this layer to determine the action according to the computed transformer policy.

For the training procedure, we use one Nvidia 6000 Ada to train the transformer with a batch size of 32, trained for 100 epochs, and we set the learning rate as $5 \times 10^{-4}$. The experimental codes are available at `https://github.com/ShenGroup/ICGP`.

## J.4 Performance Measurement

To test the performance of the pre-trained transformer (in particular, how well it approximates EXP3 and VI-ULCB), we adopt the measurement of Nash equilibrium gap. In particular, for either the transformer induced policy or EXP3, we denote the max-player's policy at time step $t$ as $\mu^t$, and the min-player's policy at time step $t$ as $\nu^t$. Furthermore, the average policy is computed as

$$\bar{\mu}^t = \frac{1}{t} \sum_{\tau \in [t]} \mu^t, \qquad \bar{\nu}^t = \frac{1}{t} \sum_{\tau \in [t]} \nu^t.$$

The NE gap at step $t$ is computed as

$$\max_{a \in \mathcal{A}} R\bar{\nu}^t - \min_{b \in \mathcal{B}} (\bar{\mu}^t)^\top R.$$

For VI-ULCB, the process is similar except that $\mu^t$ and $\nu^t$ are taken as the marginalized policies of the joint policy learned at step $t$ while the NE gap is cumulated over one episode. The NE gaps averaged over 10 randomly realized games at each step are plotted in Fig. 2.

