# OpenReview forum: "Transformers as Game Players: Provable In-context Game-playing Capabilities of Pre-trained Models"
_NeurIPS.cc/2024/Conference — NeurIPS 2024 poster_

### Official Review · Reviewer_yLUv · 2024-07-02

**Soundness:** 3
**Presentation:** 3
**Contribution:** 2
**Rating:** 6
**Confidence:** 2

**Summary:**

This paper investigates the in-context learning capabilities of pre-trained transformer models in competitive multi-agent games, i.e., in-context game-playing (ICGP). The authors provide extensive theoretical guarantees to validate that pre-trained transformers can approximate NE in an in-context manner for both decentralized and centralized learning settings. They also demonstrate that the transformer architecture can realize well-known multi-agent game-playing algorithms. Finally, experimental results support their theoretical findings, showing that transformers can effectively approximate NE in an in-context manner.

**Strengths:**

The main strengths of this paper are:

1. The paper focuses on a novel and important aspect of in-context learning in competitive multi-agent settings. Specifically, they introduce a framework for modeling ICGP via transformers.

2. This paper provides strong theoretical guarantees and backs them with empirical validation.

3. The methodology is sound, and the paper is well-structured, with clear explanations and thorough analyses.

**Weaknesses:**

1. The paper seems to be an extended work that generalizes the analysis of in-context learning in RL to the analysis of in-context learning in game theory. Certainly, this setting has not been explored before. However, this paper did not explain why this research problem is important.

2. Although the paper provides strong theoretical results, the empirical experiments are conducted on relatively simple two-player zero-sum normal-form games. This limits the generalizability of the findings.

3. The paper lacks a more detailed discussion of the practical implications.  Including such a discussion would provide a more comprehensive understanding of the applicability of the results.

**Questions:**

1. The empirical experiments are currently limited to simple two-player zero-sum normal-form games. Have you considered testing your methods in more complex environments, such as multi-player games or games with larger state-action spaces? If so, what were the results, and if not, why?

2. This paper focuses on two-player zero-sum Markov games. Could your method be applied to other types of games, such as cooperative or general-sum games? Do you have any insights or challenges you anticipate in these games?

3. Could you discuss more about the broader impacts and potential limitations of your work?

**Limitations:**

Yes, the authors provide limitations in the Appendix.

---

> ### Author Rebuttal · Authors · 2024-08-07
>
> Thank you for reviewing this work! Please find a point-by-point response provided in the following, with the reviews compressed due to the length limit.
>
> ---
>
> >**Weakness 1.** The paper ... the analysis of in-context learning in game theory ... why this research problem is important.
>
> **Response 1.** As the reviewer noted, previous works on in-context RL are mostly focused on the single-agent RL tasks. In reality, however, RL has found successful applications in broader applications, especially multi-agent tasks. Exploring the possibilities of in-context learning in multi-agent RL, as done in this work, can contribute to a deeper theoretical understanding of the capabilities of pre-trained transformers. Moreover, it also provides design guidance for practical utilization of transformers in these tasks. Given these aspects, we believe this research problem is of both theoretical and practical importance.
>
>
> ---
> >**Weakness 2.** ... the empirical experiments are conducted on relatively simple two-player zero-sum normal-form games ...
>
> >**Question 1.** The empirical experiments are currently limited to simple two-player zero-sum normal-form games ...
>
> **Response 2.** During the rebuttal period, we have performed further experiments on a more complex environment with state transitions. The results are reported in the attached PDF. It can be observed that in this more complex environment, the pre-trained transformer can still learn to approximate NE in an in-context manner and have a performance similar to that of the context algorithm, corroborating the results in the paper. We believe this empirical evidence can alleviate the reviewer's concerns, and as mentioned in Lines 581-585, it is also our hope to encourage further experiments.
>
> ---
>
> >**Weakness 3.** The paper lacks a more detailed discussion of the practical implications ...
>
> **Response 3.** We will incorporate additional discussions on the practical implications in the revised paper. Here we briefly discuss the following aspects.
>
> - First, this work can serve as an initial feasibility validation to employ transformers in performing multi-agent competitive RL tasks via an in-context fashion. Such guarantees and validations may encourage further attempts to benefit practical applications.
>
> - Moreover, our current theoretical results have the following useful implications for practical considerations.
>
>    - The context algorithm quality and the data volume are the key factors to obtain a well-performed pre-trained model. As in Theorems 3.3 and C.3, the pre-trained model will behave similarly to the context algorithm, with an error inversely related to the volume of pre-training data. The empirical results in Fig. 2 also corroborate the usefulness of involving more training data.
>
>    - The size of the adopted transformer model should scale with the complexity of the targeted game environment. The required parameterization in Theorems 3.4 and 4.1 indicates that the transformer model should have a size comparable to that of the game environment (e.g., state and action space, horizon, etc.).
>
>   - The adopted tokenizer should embed sufficiently expressive information about the game-playing trajectory. The theoretically constructed embeddings in the proofs of Theorems 3.4 and 4.1 are representative examples containing information about trajectories in different parts.
>
> ---
>
> >**Question 2.** ... Could your method be applied to other types of games, such as cooperative or general-sum games ...
>
> **Response 4.** As the first attempt in this direction, this work focuses on the two-player zero-sum Markov games due to their representativeness in game-theoretic studies. We believe the results, especially the in-context game-playing capabilities of pre-trained transformers, can potentially extend to other types of games.
>
> It is conceivable that general-sum games can be built upon the obtained results, together with an equilibrium solver powered by transformers. The extension to the cooperative games would also conceptually benefit from our study in the centralized setup and the previous single-agent investigations. As has been noted in Lines 564-570, it is our hope that this work can be a starting point and valuable foundation for exploring the pre-trained transformers in game-theoretical scenarios. We will highlight this extension as an important future direction in the revised paper.
>
> ---
>
> >**Question 3.** ... more about the broader impacts and potential limitations ...
>
> **Response 5.** Thank you for this suggestion. We will incorporate more discussions on these aspects in the revised paper, on top of those currently appearing in Appendices B.2 and B.3. Here we would like to briefly note the following aspects.
>
> - [Broader impact] The obtained theoretical guarantees and empirical evidences are helpful feasibility validations for further utilization of pre-trained transformers in multi-agent tasks, which we believe could guide practical implementations. While we do not foresee major negative social impacts due to the theoretical nature of this work, we would like to acknowledge the need for responsible usage of the practical implementation of the proposed game-playing transformers due to their high capability in various environments.
>
> - [Limitations] Appendix B.3 has discussed that it would be interesting to future investigate different game forms (as also suggested by the reviewer), pre-training dataset construction, and large-scale empirical evaluations. In addition, from the theoretical perspective, it would be valuable to investigate how to extend the current study on the tabular setting to incorporate function approximation, while another attractive question is how to learn from a dataset collected by multiple context algorithms. From the practical perspective, a future study on the impact of the practical training recipe (e.g., model structure, training hyperparameters, etc.) would be desirable to bring additional insights.
>
> ---

---

> > ### Comment · Reviewer_yLUv · 2024-08-12
> >
> > Thanks for your responses. I believe my concerns can be easily addressed in the revision. I’m happy to increase my score.

---

> > > ### Author Response · Authors · 2024-08-12
> > >
> > > Thank you for recognizing the contributions of this work! We will carefully incorporate your helpful suggestions into the revised paper.

---

### Official Review · Reviewer_beky · 2024-07-03

**Soundness:** 3
**Presentation:** 3
**Contribution:** 3
**Rating:** 7
**Confidence:** 2

**Summary:**

This paper explores the in-context learning capabilities of pre-trained transformer models in two-player zero-sum games. The authors provide the theoretical guarantees that pre-trained transformers can learn the approximate Nash equilibrium in an in-context manner, both in decentralized and centralized settings.

**Strengths:**

1. Considering multi-agent settings (two-player zero-sum games in this work) is well-motivated compared to previous works that focused on single-agent settings.

2. The theoretical results in this work provide a deeper understanding of how the pre-trained transformers can approximate NE in an in-context manner.

3. The paper is well-written, easy to follow, and makes non-trivial contributions to the relevant communities. However, because I am not an expert in theory, so it is quite difficult for me to check all the details of the proofs of all the theoretical results. Therefore, I am positive for acceptance but with relatively low confidence.

**Weaknesses:**

There are some questions:

1. In this work, the game instances are assumed to have the same length of time horizon H, which may not be always the case. For example, when playing SMAC, each game-play could terminate at any time. In this case, will the results of this work still be applicable?

2. In the current setup, the results seem only suitable for games with small state spaces because the augmented component (Line 163) needs to enumerate all the states. This could be impractical for more realistic and complex games.

3. The experiments are only conducted using normal-form games, which is simple. The performance of the proposed framework for more complex Markov games is unclear. I would have guessed that it is not very practical for the current framework as one needs to sample actions over the state space which could be extremely large.

**Questions:**

See Weaknesses.

**Limitations:**

The authors have discussed the limitations and future directions of the paper.

---

> ### Author Rebuttal · Authors · 2024-08-07
>
> We would like to first express our appreciation to the reviewer for reviewing this work. A point-by-point response is provided in the following, which hopefully can answer and clarify the raised questions and concerns.
>
> ---
>
> >**Weakness 1.** In this work, the game instances are assumed to have the same length of time horizon H, which may not be always the case. For example, when playing SMAC, each game-play could terminate at any time. In this case, will the results of this work still be applicable?
>
> **Response 1.** Yes, the results of this work are general and still applicable in the mentioned scenario. The same horizon is adopted only for theoretical convenience. A standard treatment in RL theory is to define one termination state in the state space, which provides no rewards and can only transit to itself once entered (i.e., marking that the game has ended). Then, the different plays of games can be padded with the termination state to be the same length.
>
> ---
> >**Weakness 2.** In the current setup, the results seem only suitable for games with small state spaces because the augmented component (Line 163) needs to enumerate all the states. This could be impractical for more realistic and complex games.
>
> **Response 2.** We would like to provide the following discussions on the augmented component.
>
> - We first note that, as mentioned in Lines 164-166, this augmentation is purely computational, i.e., to sample actions from the context algorithm. It requires no real interactions with the game environment. As a result, it is relatively simple to perform this operation during empirical implementations.
>
> - The reason behind the concerned enumeration over all states is essentially that the current setup is a tabular RL one, i.e., no relationships are assumed among state-action pairs. In other words, to provide a diverse dataset, we have to cover all states since information about one state cannot be provided by other states, as required by the tabular RL setup.
>
>   - In more complex games and real-world applications, function approximations via features are typically utilized to share information among state-action pairs. Thus, covering information of certain representative states should be conceivably sufficient (e.g., a well-coverage of the feature space). Similar evidences have been observed comparing the coverage requirement of tabular [R1] and function approximation [R2] setups in offline learning for Markov games. This investigation is out of the scope of this paper, and we will include these discussions in the revised paper to encourage future investigations.
>
> [R1] Cui, Q., and Du, S. S. (2022). When are offline two-player zero-sum Markov games solvable?.
>
> [R2] Zhong, H., Xiong, W., et al. (2022). Pessimistic minimax value iteration: Provably efficient equilibrium learning from offline datasets.
>
>
> ---
> >**Weakness 3.** The experiments are only conducted using normal-form games, which is simple. The performance of the proposed framework for more complex Markov games is unclear. I would have guessed that it is not very practical for the current framework as one needs to sample actions over the state space which could be extremely large.
>
> **Response 3.** Please refer to Response 2 for discussions on the action sampling, which, as noted there, requires no real interactions with the environment. In terms of empirical performance, during the rebuttal period, we have performed further experiments on a more complex environment with state transitions. The setup details and the results are provided in the uploaded PDF. From the figure, we can observe that in this more complex scenario, the pre-trained transformer can still learn to approximate NE in an in-context manner and have a performance similar to that of the context algorithm, corroborating the theoretical and empirical results in the paper. We believe this empirical evidence can alleviate the reviewer's concerns, and as mentioned in Lines 581-585, it is our hope that this work can encourage further experiments on this direction.
>
> ---

---

> > ### Comment · Reviewer_beky · 2024-08-12
> >
> > Thanks for the authors' responses, and my concerns have been addressed. So, I am happy to increase my score.

---

> > > ### Author Response · Authors · 2024-08-12
> > >
> > > We are glad to hear that our response addressed your concerns. Thank you for recognizing the contributions of this work!

---

### Official Review · Reviewer_kZH2 · 2024-07-12

**Soundness:** 4
**Presentation:** 4
**Contribution:** 4
**Rating:** 8
**Confidence:** 2

**Summary:**

The authors built on the recent work of Lin et. al 2023, extending their ICLR framework so that instead of being for one agent, it is for multi-agent systems; at the same time, they also analyze and provide evidence of "ICGP" (in-context game-playing) capabilities in transformers.

The paper analyzes zero-sum Markov games for two players and explores learning the Nash Equilibrium. In these games, two setups are studied: one of decentralized learning (where the Nash Equilibrium must be reached without each player observing the opponent's actions) and another of centralized learning (where a transformer controls the actions of both players).

The authors find important theoretical results related to transformers in decision-making, e.g., that pre-trained transformers can realize both model-based and model-free designs, and the in-context learning capability of transformers in playing normal-form games

**Strengths:**

1. **Soundness of the Claims**

   a) **Theoretical Grounding**

   The paper does a great job of formalizing and mathematically demonstrating decentralized and centralized training of transformers to achieve ICGP, and I think it is a high-impact contribution. One could argue that it is also necessary to provide more empirical evidence (which is scarce in this paper) of the dynamics being modeled, but results like Theorem 4.1 are, in my opinion, a sufficient contribution. Among other things, this work shows the capability of transformers to not only implement existing algorithms (not new in the literature) but also to adapt them to the needs of model-free designs (as far as I know, new), demonstrated by the transformer’s ability to perform exact and approximate implementations, which is important as we move to real-world applications.

   b) **Empirical Evaluation**

   The methodology followed by the authors ("J. Details of the Experiments") seems to be solid. They use the EXP3 algorithm to collect pre-training data to ensure that the data is reflective of competitive strategies and behaviors. They also customize GPT-2, with only 2 layers and 4 attention heads, tailored to match the action dimensionality in its output layer, which directly supports computing policies from the model outputs (Lines 1002-1008). The Nash Equilibrium (NE) gap is calculated by comparing the expected rewards of the max player's policy against the min player's policy over time. The gap measures how close the transformer’s induced policy is to an ideal Nash Equilibrium.

   Since the area of applied game theory for LLMs/multi-agent LLMs is still largely unexplored, providing such detailed information about the experiments and the code is likely going to be useful and impactful for the community.

2. **Significance**

   In general, I think that ICL work has a lot of impact; at this point, it is folk knowledge that ICL > fine-tuning, so papers like this one have a lot of significance for the advancement of the state of the art. In particular, Theorems 3.4 and 4.1 can prove to be of great impact.

3. **Novelty**

   This paper is timely and tackles a timely and interesting topic. To the best of my knowledge, it is the first paper providing a theoretical analysis of this level on game theory applied to LLMs.

**Weaknesses:**

1. **Soudness of the claims**

   My concerns about the paper relate to whether this theoretical framework might struggle with generalization and performance in more complex games. In this section, I provide some examples of limitations that suggest this possibility. I'm concerned that it may excel in controlled environments but not in more complex scenarios, which are typical in the multi-agent literature.

   Also, please note that the improved performance of transformers trained with 20 games compared to those trained with 10 (Lines 336-341) suggests a heavy dependence on the volume of training data. Thus, I'm also highlighting some potential problems related to this aspect.

   a) **Theoretical grounding**

      a1) **Decentralized learning scenario**

   In the decentralized setting, my primary concerns revolve around the diversity and representativeness of the training data, overly optimistic assumptions about model approximation capabilities, and potential issues with the suitability of transformer outputs for generating valid action probabilities.

* (Lines 157-170) If the data collected through $Alg+,0$ and $Alg−,0$ only captures a limited range of strategic interactions (e.g., typical or frequent scenarios but not edge cases or unusual strategies) the training data might lack the necessary diversity, leading to overfitting/lack of robustness. If the data does not sufficiently cover the state-action space, the expected log probability of correct actions given the state, as modeled by the transformer, may not reflect true gameplay dynamics and $\mathbb{E}\left[\log P(\text{actual action} \mid \text{state})\right]$ may overestimate actual performance.
* (Lines 200-202) Take into account that with Assumption "3.2. Decentralized Approximate Realizability" you assume that the true data-generating distribution ($Alg_0^+$) can be closely approximated by the learned model ($Alg_\theta^+$). This assumption may not hold if the real complexity of strategies in the environment cannot be captured by the model's architecture or training data -- causing the models have generalization issues.
* (Lines 195-190) Potential convergence/generalization problems with the learning algorithm if the covering number $\Theta$ is underestimated. The clipping operation (Lines 936-938), which is designed to maintain the norm of activations within bounds, can significantly impact the parameter sensitivity and the gradient flow during backpropagation. So, if $\Theta$ is underestimated, it might not sufficiently account for the reduced effective parameter space available for optimization, possibly leading to convergence on local minima that do not generalize well
* (Lines 182-185) The use of linear extraction mappings followed by a projection to the probability simplex to determine action probabilities assumes that the transformer’s outputs can be linearly mapped to form valid probability distributions. Buuut, if the transformer outputs are not suitable for this kind of linear transformation due to scale, bias, or other distributional issues, the resulting action probabilities may not be valid or optimal --> leading to suboptimal decisions and erratic behavior, especially in complex games.

   a2) **Centralized learning scenario**

   For the centralized learning scenario, I want to provide feedback on some specific concerns related to computational demands, dependency on precise model and algorithm alignment, and assumptions about uniform sampling that may not adequately capture the complexity of strategic behaviors across varied game states

* (Lines 307-309) The required dimensions and mappings such as $d⪅HS^2AB,L⪅GHSd⪅HS^2AB,L⪅GHS$ etc., indicate a high computational complexity and dependency on specific game parameters (like $S,A,B,G,HS,A,B,G,H$). As a result, this could make the model computationally expensive and potentially overfit to specific types of game environments.
* (Lines 307-309) This performance requirement: $Algθ(⋅,⋅∣Dt−1,s)=AlgVI−ULCB(⋅,⋅∣Dt−1,s)Algθ​(⋅,⋅∣Dt−1​,s)=AlgVI−ULCB​(⋅,⋅∣Dt−1​,s)$ assumes an ideal alignment between the transformer’s output and the algorithm’s requirements; note that any deviation in this alignment due to model approximation errors or misestimations in the transformer's training could lead to significant performance drops...
* (Lines 321-322) Note that the assumption that $\hat{\mu}\$ and $\hat{\nu}\$ are uniformly sampled and provide an effective representation of strategic options in every game state might not hold if some strategies are inherently more complex or contextually sensitive than others.

**Questions:**

* Given that the decentralized setup involves collecting offline trajectories using different algorithms for max- and min-players (Section "3.1 Supervised Pre-training Results") -- how do you ensure that the diversity and distributional properties of the training data do not adversely affect the learning outcomes?
* The clipping operation $clip_R$ is defined to constrain the norm of each vector (Lines 936 - 938). How does this norm constraint affect the learning dynamics, particularly in terms of gradient propagation and stability during backpropagation? This seems to be important as it directly affects the model's ability to learn and fine-tune strategies based on gameplay experience.
* In Appendix G.3, you detail the use of embedding and extraction mappings within the transformer architecture for implementing V-learning (Lines 810-819). These mappings essentially allow the transformer to understand and manipulate game states and decisions effectively. Are there specific state spaces where these mappings prove particularly beneficial/limited?

Just out of curiosity:
* Have you observed any sudden shifts in learning effectiveness or strategy adaptation in your decentralized learning models, based on the results from "The Mechanistic Basis of Data Dependence and Abrupt Learning in an In-Context Classification Task" by Gautam Reddy (2023)?

**Limitations:**

Yes.

---

> ### Author Rebuttal · Authors · 2024-08-07
>
> Thank you for reviewing this paper! The following responses are provided with the reviews compressed due to the length limit.
>
> ---
> >**W1.** (Lines 157-170) If the data collected ... a limited range of strategic interactions ... might lack the necessary diversity ...
>
> >**Q1.** Given ... the diversity and distributional properties of the training data do not adversely ...
>
> **R1.** Theorems 3.3 and C.3 are theoretical supports that better data coverage leads to better pre-trained models. The intuition that more data provides a better characterization of the context algorithm is rigorously captured by the error's $\sqrt{1/N}$ dependency on the data volume $N$.
>
> Also, Theorems 3.3 and C.3 are *high-probability* guarantees. It means that the performance can be abnormal under small-probability events, e.g., a badly sampled dataset. Nevertheless, these undesirable events are statistically rare occurrences within the data collection process.
>
> ---
>
> >**W2.** (Lines 200-202) Take into account ... This assumption may not hold ...
>
> **R2.** Assumption 3.2 introduced the realization error, i.e., $\varepsilon_{real,+}$, to model the capability of the learning model $Alg_{\theta_+}$ in approximating $Alg_{0,+}$. We first clarify that its core part is a definition: any $Alg_{\theta_+}$ satisfies the condition with its own scale of $\varepsilon_{real,+}$.
>
> Through this notation, Theorem 3.3 showed a pre-training guarantee depending on $\sqrt{\varepsilon_{real,+}}$, i.e., sublinearly. Thus, a reasonable $\varepsilon_{real,+}$ (i.e., a suitably strong model) ensures a good pre-training model.
>
> Theorem 3.4 and 4.1 further demonstrated that the transformer architecture is indeed strong enough to realize V-learning and VI-ULCB, two representative game-playing algorithms, i.e., $\varepsilon_{real,+} = 0$. These are the first results on the strong capability of transformers in competitive games.
>
> ---
>
> > **W3.** (Lines 195-190) Potential convergence/generalization problems ... if the covering number $\Theta$ is underestimated ...
>
> >**Q2.** The clipping operation $\text{clip}_R$ ... affect the learning dynamics ...
>
> **R3.**  The covering number of $\Theta$ is a definition (i.e., Definition 3.1) to theoretically capture the size of the parameterization space. Due to its entirely theoretical purpose, it is not used/required empirically.
>
> The clipping operation is to bound the covering number (as in Appendix I) for theoretical completeness. As in Lines 972-974, setting $R$ on the order of $\tilde{O}(1)$ is sufficient to not impact the theoretically required transformer operations. With its entirely theoretical propose, this clipping is also not used/required empirically.
>
> ---
>
> >**W4.** (Lines 182-185) The use of linear extraction mappings followed by a projection to the probability simplex ... not suitable ...
>
> **R4.** This work focuses on linear extraction mappings followed by a projection to probability simplex for theoretical analyses. It is *always a valid approach* to convert vector outputs from transformers to action distributions. Theorems 3.4 and 4.1 rigorously illustrate its sufficiency, as combining it with other operations *exactly* realizes the complex game-playing algorithms. Of course, in practice, we can leverage other operations, e.g., a non-linear mapping, which would provide higher flexibilities.
>
> ---
>
> >**W5.** (Lines 307-309) The required dimensions ... indicate a high computational complexity and dependency on specific game parameters ...
>
> **R5.** The dimensional specifications in Theorem 4.1 are curial results, showing that such a model size is already sufficient to realize VI-ULCB. The polynomial dependency on the game parameters is a very mild requirement, especially considering that practical transformers with millions to billions of parameters. We will encourage future investigations on lowering these requirements.
>
> ---
> >**W6.** (Lines 307-309) This performance requirement ... assumes an ideal alignment ...  any deviation ...
>
> **R6.** Theorem 4.1 demonstrates that the transformer is capable of *exactly* realizing VI-ULCB, i.e., ensuring $Alg_{\theta}(\cdot, \cdot|D_{t-1},s) = Alg_{\text{VI-ULCB}}(\cdot, \cdot|D_{t-1},s)$. It is not an assumption, but a novel theoretical guarantee of the strong capability of transformers, which applies to *any* two-player zero-sum game. Hopefully this alleviates the concern on model capabilities.
>
> The mentioned training error is independent of this theorem (which focuses on the existence). We can handle it via pre-training results in Sections 3.1, 4.1 and Appendix C by modeling it as one part of the realization error, which is further discussed R2.
>
> ---
> >**W7.** (Lines 321-322) Note that the assumption ... uniformly sampled ... might not hold ...
>
> **R7.** The uniformly sampled $\hat{\mu}$ and $\hat{\nu}$ is the standard online-to-batch conversion, to theoretically translate the cumulative regret to a sample complexity guarantee. It is a relatively simple process, i.e., sample a random episode from $1, \cdots, G$ and use the policy of that episode.
>
> ---
> >**Q3.** In Appendix G.3, ... embedding and extraction mappings ... specific state spaces where these mappings prove particularly beneficial/limited?
>
> **R8.** The mappings designed in theoretical proofs are indeed to encode sufficient information (i.e., step, state, action, reward, etc.) as the input. Although we did not notice states particularly special in terms of this mapping, it is generally a good mapping strategy to contain sufficient information to facilitate decision-making.
>
> ---
> >**Q4.** Have you observed any sudden shifts ... based on ...
>
> **R9.** We will cite and discuss this interesting reference! During the previous experiments, we have not examined the training process as delicately as it. We are currently following its reported procedures to conduct further examinations, which have not yet completed due to the limited time but will be reported in the revised paper.
>
> ---

---

> ### Comment · Reviewer_kZH2 · 2024-08-13
>
> Thanks for your detailed rebuttal. I am happy with the clarifications the authors provided. As Reviewer h1zf mentioned, I want to highlight that this work is not only important for the MARL community but also for the broader research community focused on advancing our understanding of ICL. Congratulations on this great work.

---

> > ### Author Response · Authors · 2024-08-13
> >
> > It is a great pleasure to hear the recognition that this work brings valuable contributions to communities of both MARL and ICL. Thank you for reviewing this work and providing the detailed comments! We will carefully incorporate your suggestions into the revised paper.

---

### Official Review · Reviewer_h1zf · 2024-07-13

**Soundness:** 3
**Presentation:** 3
**Contribution:** 3
**Rating:** 6
**Confidence:** 3

**Summary:**

The paper proposes a framework for pre-trained transformers to approximate Nash equilibrium in two-player zero-sum normal-form games and provides theoretical guarantees to show that these pre-trained transformers can learn to approximate Nash equilibrium in an in-context manner. This is shown for both the decentralized and centralized learning settings. The decentralized case follows the idea that each agent will have it's own model, and uses V-learning. The centralized case uses a centralized joint model to control both players, and uses VI-ULCB. The paper shows that there is a provable upper bound to the approximation error of the Nash equilibrium which demonstrates the in-context game player capability of transformers. Further more, empirical results show that given a sufficiently well pre-trained transformer that it can approximate Nash equilibrium in an in context manner, similar to that of decentralized EXP3 in two-player zero-sum normal-form games, that is it has a similar profile of the gradually decaying Nash equilibrium gap.

**Strengths:**

Originality:
This work offers new insights into the the theoretical analyses and empirical evidence for the in context game player capabilities of transformers.

Clarity and Quality:
The paper is well cited and explains the concepts it is conveying quite well, with the appendix showing the details of the proofs and supporting code demonstrating the reproducibility of the empirical work.

Significance:
The paper shows the theoretical capabilities of transformers adapted to V-learning (and thus aiding in the scalability of MARL) as well as approximating Nash equilibrium are important to MARL settings.

**Weaknesses:**

I have concerns that since the pre-training dataset is collected from a context algorithm that the transformers are simply learning to mimic this algorithm.

The fact that empirical results are only collected from bandit settings, I have concerns that this may not hold in a setting where there are state transitions. I would have hoped that there would have been a more complex domain than just matrix games.

**Questions:**

1) Do you believe that this implementation might have different results given a different pre-training dataset (produced from a different algorithm)?

2) Do you believe that in more complex domains, with state transitions, we will still observe similar results? Or is this a factor that might break the current state of this work?

**Limitations:**

The authors have identified the limitations and have stated that there should be no negative societal impact due to the theoretical nature of their work.

---

> ### Author Rebuttal · Authors · 2024-08-07
>
> Thank you for reviewing this work! We are happy to hear your recognition of the new insights and contributions of this work. The following point-by-point response is provided, which hopefully can address the raised questions and concerns.
>
> ---
>
> >**Weakness 1.** I have concerns that since the pre-training dataset is collected from a context algorithm that the transformers are simply learning to mimic this algorithm.
>
> >**Question 1:** Do you believe that this implementation might have different results given a different pre-training dataset (produced from a different algorithm)?
>
> **Response 1.** As weakness 1 and question 1 are both related to pre-training the transformer with data from the context algorithms, we would like to discuss them together in the following.
>
> - Before further discussions, we note that the utilization of a context algorithm to collect pre-training data is a standard approach in in-context RL studies, pioneered by [R1]. This work extends this well-established framework to the multi-agent scenario and provides corresponding theoretical/empirical studies.
>
> - The main goal of this paper is to show that the transformer, pre-trained from pre-collected game-playing data, can behave as a game-playing algorithm in an in-context manner. The context algorithms are introduced to provide distributional properties of the pre-training dataset. Thus, if the context algorithm changes, then the distribution of pre-training data changes, which intuitively results in that the pre-trained transformer and its induced game-playing algorithm also change.
>
>   The main result of our paper is the theoretical characterization of the transformer-powered game-playing algorithm extracted from the pre-training data. We would like to highlight that this is a highly nontrivial task in the game-playing setting that has not been investigated in previous works, with its major challenges emphasized in the following.
>
>   1. The game-playing data to be learned from contain complicated strategic interactions between the context algorithms and the environment. In the decentralized setting, there are also interactions between two separate context algorithms, further complicating the pre-training data distributions. It is a highly challenging task to extract the game-playing algorithm from such complex data distributions.
>   2. Given the complicated data, even imitation learning in game-theoretical environments are rarely studied. In this work, we take an even bigger step as the to-be-trained game-playing algorithm must be confined to the practically-adopted transformer architecture instead of any other neural nets. As an illustration, the theoretical proofs of Theorems 3.4 and 4.1 are performed strictly under the transformer architecture described in Section 2.3, while the empirical experiments also use a transformer-based minGPT model (see details in Appendix J.3).
>
> [R1] Laskin, M., Wang, L., et al. (2022). In-context reinforcement learning with algorithm distillation.
>
> ---
>
> > **Weakness 2.** The fact that empirical results are only collected from bandit settings, I have concerns that this may not hold in a setting where there are state transitions. I would have hoped that there would have been a more complex domain than just matrix games.}
>
> >**Question 2.** Do you believe that in more complex domains, with state transitions, we will still observe similar results? Or is this a factor that might break the current state of this work?}
>
> **Response 2.** Thank you for your comments! During the rebuttal period, we have performed further experiments on a more complex environment with state transitions. The setup details and the results are provided in the uploaded PDF. From the figure, we can observe that in this more complex environment, the pre-trained transformer can still learn to approximate Nash equilibrium in an in-context manner and achieve a performance similar to that of the context algorithm, which corroborates the theoretical and empirical results in the paper. This and additional results will be added to the revised paper. We hope this empirical evidence can alleviate the reviewer's concerns.
>
> ---

---

> > ### Comment · Reviewer_h1zf · 2024-08-12
> > **I am happy with the author responses, and they have addressed my concerns.**
> >
> > I am happy with the author responses and I believe the authors have addressed my concerns.
> > I would like to see the details of the new experiment expanded upon in the revised paper.
> > Overall I believe this work will be valuable to the greater MARL community.

---

> > > ### Author Response · Authors · 2024-08-12
> > >
> > > It is our pleasure to hear that the responses addressed your concerns. The new experiment will be described with full details in the revised paper. Thank you for recognizing the contributions of this work!

---

### Author Rebuttal · Authors · 2024-08-07

Dear Reviewers,

We would like to express our gratitude for all your time and effort in reviewing this work. Point-by-point responses have been provided, which hopefully can address the raised questions and comments. It would be our pleasure to engage in further discussions and incorporate any suggestions that you may have.

Along with this response, one PDF is uploaded, containing additional experimental results on the in-context game-playing capabilities of pre-trained transformers in a more complex environment (with state transitions). These results will be added to the revised paper. We believe they provide further evidences of the obtained theoretical and empirical results.

Our appreciations again!

Best regards,
Authors of Submission 13525

---

### Decision · Program_Chairs · 2024-09-25

**Decision:**

Accept (poster)

**Comment:**

This paper extends our theoretical understanding of transformers’ in-context learning results from single agent to two-player, zero-sum settings. In particular, it establishes precise bounds on the sizes of the transformer architectures needed to represent two convergent, no-regret game playing algorithms: decentralized V-learning and centralized VI-ULCB. The paper can be improved by adding:

*References to background literature and context for readers*:
- In-context game playing has been previously explored empirically in complex settings, for example, extensive-form [1].
- Transformers have been proven to be universal function approximators of sequence-to-sequence mappings [2]. Applying this to the game setting here, it is expected that transformers can map action histories to distributions over next actions given enough training data and a large enough network.

*More Extensive Experiments on Markov Games*:
- The authors initially included only results averaged over 10 randomly generated zero-sum, bi-matrix games, providing additional results on randomly generated Markov games in the rebuttal which was appreciated. Given the setting of the theoretical results, it would be beneficial to see 1) additional experiments on Markov games with longer horizons, 2) more trials, and 3) a normalized NE-gap to account for randomly generated games with small payoff ranges. If longer horizons (1) are not practically possible, please discuss this as a limitation in the paper.

The general consensus among the reviewers is that the generalization from single agent to two player zero sum games is a non-trivial task, especially in the decentralized version, therefore, we recommend accepting this paper.

- [1] Gandhi, Kanishk, Dorsa Sadigh, and Noah Goodman. "Strategic Reasoning with Language Models." NeurIPS 2023 Foundation Models for Decision Making Workshop. https://arxiv.org/pdf/2305.19165
- [2] Yun, Chulhee, et al. "Are Transformers universal approximators of sequence-to-sequence functions?." International Conference on Learning Representations. https://arxiv.org/abs/1912.10077